


Evaluating the accuracy of downwind methods for quantifying point source emissions
Mercy Mbua*,[1], Stuart N. Riddick[1], Elijah Kiplimo[1] and Daniel Zimmerle[1]
*The Energy Institute, Colorado State University, CO, 80524, Fort Collins, USA*
*Correspondence to*: Mercy Mbua (Mercy.Mbua@colostate.edu)
**Abstract.** The accurate reporting of methane ($CH_4$) emissions from point sources, such as fugitive leaks from oil and
gas infrastructure, is important for evaluating climate change impacts, assessing $CH_4$ fees for regulatory programs,
and validating methane intensity in differentiated gas programs. Currently, there are disagreements between emissions
reported by different quantification techniques for the same sources. It has been suggested that downwind $CH_4$
quantification methods using $CH_4$ measurements on the fence-line of production facilities could be used to generate
emission estimates from oil and gas operations at the site level, but it is currently unclear how accurate the quantified
emissions are. To investigate model accuracy, this study uses fence-line simulated data collected during controlled
release experiments as input for eddy covariance, aerodynamic flux gradient and the Gaussian plume inverse methods
in a range of atmospheric conditions. The results show that both the eddy covariance and aerodynamic flux gradient
methods underestimated emissions in all experiments.  Although calculated emissions had significant uncertainty, the
Gaussian plume inversion method performed better.  The uncertainty was found to have no significant correlation
with most measurement variables (i.e. downwind measurement distance, wind speed, atmospheric stability, or
emission height), which indicates that the Gaussian method can randomly either underestimate or overestimate
emissions. For eddy covariance, downwind measurement distance and percent error had negative correlation
indicating that far away emissions sources were likely underestimated or be undetected. The study concludes that
using fence-line measurement data as input to eddy covariance, aerodynamic flux gradient or Gaussian plume inverse
method to quantify $CH_4$ emissions from an oil and gas production site is unlikely to generate representative emission
estimates.
**1 Introduction**
Methane ($CH_4$), the primary component of natural gas (NG), is a potent greenhouse gas with a global warming
potential of 27 carbon dioxide ($CO_2$) equivalent over 100 years (US EPA, 2016). Methane emissions reduction is a
key part of global initiatives to reduce climate change (Chung, 2021). The 2021 Global Methane Assessment by the
Climate and Clean Air Coalitions (CCAC, 2024) and the United Nations Environment Programme (UN Environment
Programme, 2024) state that reducing $CH_4$ emissions from anthropogenic sources by 45% in 2030 would result in
avoiding a global atmospheric temperature increase of 0.3°C in 2045 (Chung, 2021). Such measures would align with
the Paris Agreement goal of limiting global temperature rise to 1.5˚C by 2030 (United Nations Climate Change, 2015).
The US is one of the countries that reports its total greenhouse gas emissions to the Intergovernmental Panel on
Climate Change as part of the Paris Agreement (United Nations Climate Change, 2015).
Currently, the amount of $CH_4$ emitted from US oil and gas production is calculated by the US Environmental
Protection Agency (EPA) using a bottom-up inventory approach. The inventory approach multiplies emission factors



(CH$_4$ emissions per equipment e.g., separator or emissions per event e.g., liquid unloading) by activity factors (total
number of pieces of equipment or events (OAR US EPA, 2023)). This quantification approach has several
shortcomings, including: 1. It separately calculates CH$_4$ emissions from natural gas and petroleum systems, which
practically are not independent systems, and can result in bias based on changes in gas to oil ratios throughout a basin
(Riddick et al., 2024a); 2. Some emission factors used are outdated (Riddick et al., 2024b) and others do not account
for the temporal and spatial variation in emissions (Riddick and Mauzerall, 2023); and 3. Emission factors do not
account for the long-tail distributions in emissions distributions (Riddick et al., 2024b). Recently, mechanistic models,
such as the Colorado State University's Mechanistic Air Emissions Simulator (MAES), have been developed to
address shortcomings in bottom-up CH$_4$ reporting (Colorado State University, 2021) but these still depend on direct
measurements to inform emission factors.
Top-down methods, including using aircraft and satellites, can also be used to infer emissions. For example, Carbon
Mapper satellites can locate and quantify CH$_4$ emissions using absorption spectra taken from space (Carbon Mapper,
2024). However, these survey methods only quantify emissions over a very short period of time (< 10 s) and
observations are typically made during the day which can often coincide with maintenance activities that can bias
emissions and result in overestimation (Riddick et al., 2024a; Zimmerle et al., 2024). Additionally, different top-down
technologies measuring the same source have disagreed in their reported emissions which has called into question the
credibility of these methods (Brown et al., 2023; Conrad et al., 2023). As a result, ensuring accuracy in models and
technologies used in CH$_4$ emissions quantification has been a complex issue.
The accurate reporting of CH$_4$ from fugitive emissions at oil and gas production sites is important for evaluating
potential effects on climate change, correctly assessing CH$_4$ fees on companies as part of the Methane Emissions
Reduction Program created under the 2022 Inflation Reduction Act (OA US EPA, 2023), and validating CH$_4$ content
of reported differentiated gas composition where NG companies differentiate their market products based on the
environmental impact (CO2EFFICIENT, 2022). Direct measurements have been recommended to augment/update
emissions factors used in bottom-up inventories and for better understanding temporal/spatial variability of emissions
(Riddick et al., 2024). Downwind methods are widely used to directly measure CH$_4$ emissions from area and point
sources at site/basin levels due to their low cost and wide coverage within a short time (Caulton et al., 2018;
Heimburger et al., 2017; Riddick et al., 2020, 2022a; Sonderfeld et al., 2017). Commonly used downwind
quantification methods include the Gaussian plume inversion method, eddy covariance, backward Lagrangian
stochastic models, aerodynamic flux gradient, mass balance method, the EPA Other Test Method (OTM 33) and the
Gaussian puff modelling approach (Denmead, 2008; Edie et al., 2020; Foster-Wittig et al., 2015; Jia et al., 2023; Kamp
et al., 2020; Nemitz et al., 2018; Shaw et al., 2021).
Currently, fence-line methods are used to detect, localize and quantify emissions. This approach uses point sensors
fixed to the fence-line of the production site and emissions detected when the measured concentration exceeds a
threshold, localized by triangulating multiple detections and quantified using a simple dispersion modelling
framework, usually based on a Gaussian plume approach (Bell et al., 2023; Day et al., 2024; Jia et al., 2023; Riddick
et al., 2022a). The detection and localization of simulated fugitive emission have been successful, with controlled
release testing against point sensors and scanning/imaging solutions reporting a 90% probability of detection for



emission of between 3.9 and 18.2 kg CH$_4$ h$^{-1}$ (Ilonze et al., 2024). Major shortcomings have been identified using a
fence-line approach with quantified emissions reported at between a factor of 0.2 to 42 times for emissions between
0.1 and 1 kg CH$_4$ h$^{-1}$, and between 0.08 and 18 times for emissions greater than 1 kg CH$_4$ h$^{-1}$ (Ilonze et al., 2024). As
a result, questions have arisen if other approaches, such as the eddy covariance or aerodynamic flux gradient would
generate more accurate results. These methods have been suggested as they have been used to quantify emissions
from other sectors, i.e. agriculture (Denmead, 2008; Morin, 2019) and landfills (Xu et al., 2014), have been used to
quantify emissions in large downwind areas (Vogel et al., 2024), and quantification does not require assumptions
made on downwind dispersion coefficients or micrometeorology that are often required for dispersion modelling
(Denmead, 2008).
Due to interest in using a subset of these methods to quantify emissions from oil and production sites, this study will
evaluate the quantification accuracy of the eddy covariance, aerodynamic flux gradient, and Gaussian plume inverse
methods. Eddy covariance is a vertical flux gradient measurement that measures CH$_4$ emissions based on the
covariance between CH$_4$ concentrations measured using a fast-response analyzer (> 10 Hz) and vertical wind vector
measured by a fast-response sonic anemometer (>10 Hz) (Figure 1A; Morin, 2019). It is typically implemented over
long homogenous fetches where eddy mixing scale is a small fraction of the distance from the site providing more
predictable vertical transport. The aerodynamic flux gradient method quantifies CH$_4$ emissions from a source by
comparing CH$_4$ concentrations at two heights (Figure 1B; Querino et al., 2011). The Gaussian Plume Inverse method
calculates CH$_4$ mole fraction at a point in space (x, y, z) as a function of the downwind distance, perpendicular distance
(crosswind), mean wind speed and atmospheric stability (Jia et al., 2023; Riddick et al., 2022b). These approaches
were developed to quantify emissions from single-point or area emission sources and have not been tested against a
controlled release to evaluate their quantification performance. The aerodynamic flux gradient and eddy covariance,
for example, have been used to measure trace gas, e.g., nitrogen oxide and carbon dioxide, fluxes from large croplands
(Kamp et al., 2020).

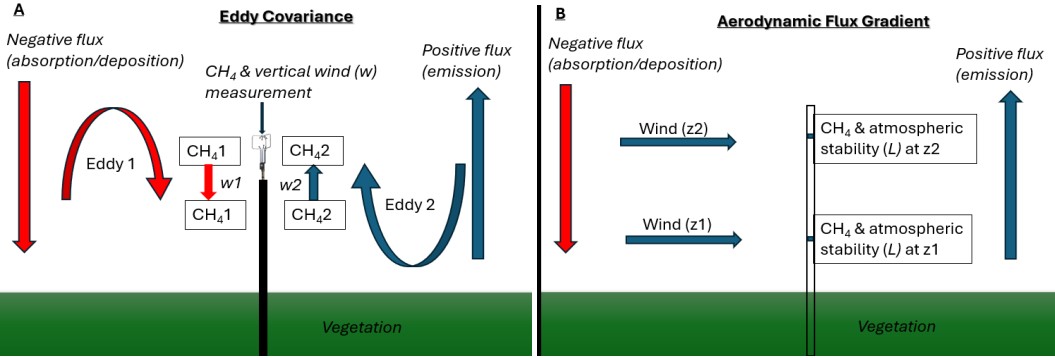


Figure 1: Illustrations of eddy covariance (A) and flux gradient measurements (B) where CH$_4$ is methane
concentrations, *w* is the vertical wind speed, *L* is the Monin-Obukhov length (measure of atmospheric stability), and
z is the measurement height.
The Gaussian plume inversion method has been used to quantify emissions from oil and gas production sites (Caulton
et al., 2014; Riddick et al., 2022b) but it assumes a homogenous, steady state flow, uniform dispersion of gas in an



open area free of obstructions (Hutchinson et al., 2017). Oil and gas emissions are characterized by intermittent, non-
uniform, single or multiple point source emissions, varying in leak size, location, height and distance between the
source and sensor, and are typically in complex aerodynamic environments (i.e. not flat). The need for accurate $CH_4$
quantification and reporting necessitates evaluating the performance of these downwind quantification approaches in
different controlled release and characterized meteorological conditions, to ensure credibility.
This study aims to investigate the performance of these methods in quantifying emissions for known gas release rates
and evaluating uncertainties that could result in incorrect $CH_4$ reporting. Specifically, the study will (1) evaluate the
overall quantification accuracy of eddy covariance, aerodynamic flux gradient, and the Gaussian plume inverse
method in quantifying single-point and multi-point emissions that simulate oil and gas emissions, (2) evaluate the
probability of these models quantifying within a defined range (i.e. ±30%), and (3) investigate which variables have
the largest effect on quantification uncertainty.
**2 Methods**
**2.1 Experimental Setup**
Controlled release experiments were conducted at the Colorado State University's Methane Emissions Technology
Evaluation Center (METEC) in Fort Collins, CO, USA, between February 8, and March 27, 2024. The weather
conditions during the test period were mostly sunny but precipitation was also observed (32 sunny, 7 snowy, 12 rainy,
7 cloudy and 1 foggy day; Supplementary Information Section 1). Wind speeds were between 0 and 25 m s$^{-1}$ and
temperatures ranged between -15 and +19 °C (Supplementary Information Section 1). Two stationary masts holding
the instrumentation were setup on the North-West corner of METEC to take advantage of the predominant wind
direction, avoid the largest aerodynamic obstructions and to simulate the likely placement of a fence line instrument
(Figure 2; Day et al., 2024; Riddick et al., 2022a). Fenceline sensors are typically placed within the oil and gas
perimeter (~30 m) (Riddick et al., 2022a). This study collected data for both close and far away releases, distances
between 9 and 94 m.

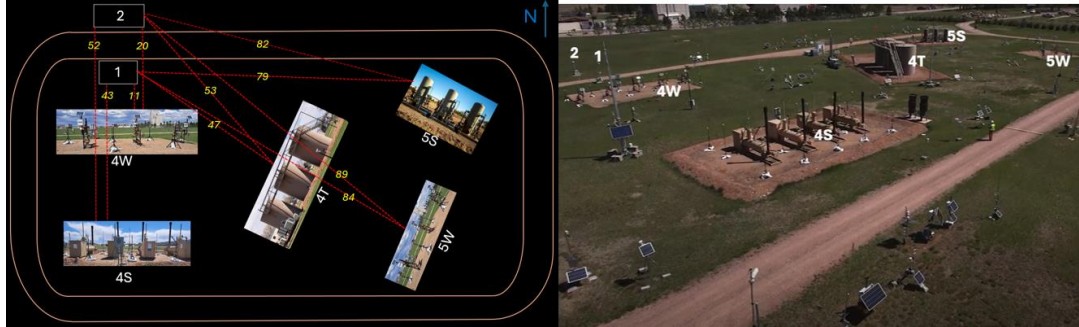

Figure 2: Left pane: Map illustration of major pieces of equipment and the measurements points at Colorado State
University's Methane Emissions Technology Evaluation Center (METEC) in Fort Collins, CO, USA. 4S denotes the
location of horizontal separators, 4W are well heads, 4T are tanks, 5S are vertical separators and 5W are well heads.
1 is the measurement point for the Microportable Greenhouse Gas Analyzer and 2 is the measurement point for the
Aeris analyzers. The red dotted lines with yellow numbers show the average distances (meters) between emission
equipment and measurement points. Right pane: Image of METEC showing relative heights of equipment ("METEC
| Colorado State University," 2024).
To calculate emissions using the aerodynamic flux gradient approach, two sampling inlets were mounted at 2 and 4
m heights on mast 2 and connected to the inlets of two Aeris (Hayward, CA, USA) MIRA Ultra Series analyzers
(Figure 3A). The analyzers were housed in a temperature-controlled unit and sampled at 5 Hz. Data from the 2 m
analyzer were also used as input for the Gaussian Plume Inverse method analysis. To collect $CH_4$ concentration data
for the eddy covariance method, the inlet tubing of the ABB (Zurich, Switzerland) GLA131 Series Microportable
Greenhouse Gas Analyzer (MGGA) sampling at 10 Hz was collocated with an R. M. Young (Traverse City, MI, USA)
81000 sonic anemometer (R.M. Young Company, 2023) which measured micrometeorology at 10 Hz, 3 m height
above ground level on mast 1 (Figure 3B).




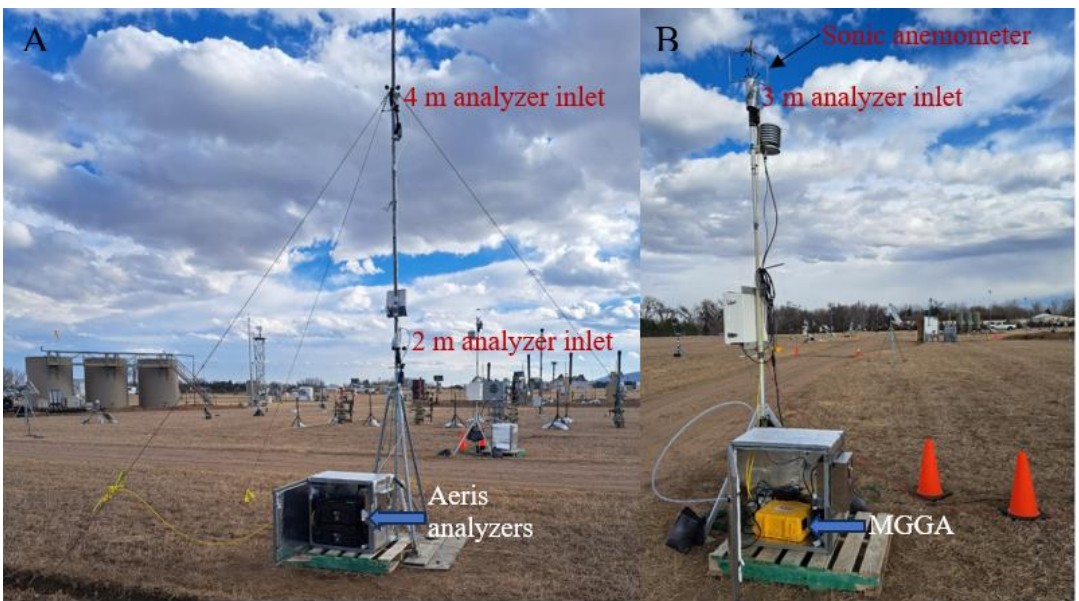


Figure 3: A is the aerodynamic flux gradient and Gaussian plume inverse sampling points and B is the eddy covariance
sampling point. The two sampling points are 9.4 m apart.
**2.2 Controlled Methane Releases**
At METEC, natural gas of known $CH_4$ content was released from above-ground emission points attached to equipment
typically present in an oil and gas facility (tanks, separators and well pads). The gas release rates ranged between
0.005 kg h$^{-1}$ and 8.5 kg h$^{-1}$, and the release durations ranged from 10 seconds to 8 hours, simulating both fugitive and
large emission events. The releases were run both during the day and night. The distance from the release points to
the measurement points ranged between 9 and 94 m, and emission heights were between 0.4 and 6.9 m (Figure 2).
Emission points simulate the realistic size and locations of typical emission from components such as the thief hatches,
pressure relief valves, flanges, bradenheads, pressure transducers, Kimray valves and vents. The releases included
both single point emissions (single releases) and multi-point emission events (multiple simultaneous releases).
**2.3 Data Processing**
Methane concentrations data from the analyzers were aggregated with the meteorological data from the sonic
anemometer. For aerodynamic flux gradient and Gaussian plume inverse method data were averaged to 1 Hz, for the
eddy covariance the raw $CH_4$ 10 Hz data was used. The aggregated meteorological-concentration data were then
merged with METEC's release data and metadata, and event tables created. The meteorological-concentration-release
event data were then separated into single-point and multi-point events. The event tables were split into 20-minute
emission events for aerodynamic flux gradient and Gaussian plume inverse method as they are dependent on
atmospheric stability that is typically determined in time durations of 15 to 30 minutes. Shorter duration measurements
(i.e. <15 minutes) may not represent the mean atmospheric state, while longer periods (> 30 minutes) may cause errors
especially during rapid transitions in weather conditions (Crenna, 2006). 30-minute events were used for eddy




covariance processing following published typical averaging times of eddy covariance measurements (Nemitz et al.,
2018), and its quantification is assumed to be independent of atmospheric stability (Denmead, 2008).
For eddy covariance and aerodynamic flux gradient, Monin-Obukhov length ($L$) was calculated as the measure of
atmospheric stability for every 20 or 30-minute time period, depending on the method, using output from the sonic
anemometer. $L$ was calculated from the surface friction velocity ($u_*$, m s$^{-1}$), mean potential temperature ($\Theta$, K), von
Kármán's constant ($k$, 0.41), gravitational acceleration ($g$, 9.8 m s$^{-1}$) and the surface (kinematic) turbulent flux of
sensible heat w'$\Theta$' (Eq. 1 and 2) (Kljun et al., 2015; Stull, 1988).

$$L = -\frac{u_*^3 \Theta}{k_v g \overline{w'\Theta'}}$$

(1)


$$u_* = \left[\left(\overline{u'w'}\right)^2 + \left(\overline{v'w'}\right)^2\right]^{1/4}$$

(2)

For the Gaussian method, atmospheric stability was calculated based on the EPA standard operating procedure for
point source Gaussian method (US EPA, 2013). The average local wind stability class ($pgi$) was calculated as the
average of atmospheric stability determined using the standard deviation of the wind direction, and the stability
calculated from turbulent intensity (ratio of the standard deviation of the wind speed to the average wind speed). The
dispersion coefficients used for Gaussian quantification were extracted from the EPA operating procedure that
provided coefficients for distances ranging from 1 to 200 m from source (US EPA, 2013).
The wind direction ($WD$) and speed ($WS$) were calculated from the wind vectors u and v, based on the manufacturer's
configuration: +$u$ values = wind from the east, +$v$ values = wind from the north, and +$w$ = updraft (Eq. 3 and 4).

$$WD = \mathrm{mod}(90 - \mathrm{atan2d}(v, u), 360)$$

(3)

$$WS = \sqrt{u^2 + v^2}$$

(4)

The bearing of each release point to the masts' location was calculated using the latitudes and longitudes of the release
points provided in the METEC metadata. This bearing was used to determine when the masts were downwind of the
release points during the 20/30-minute period. The models' quantification accuracies were tested in three downwind
ranges: ±10°, ±20°, and ±30°. A mast was considered downwind when the wind direction was within the specified
range for 30% of the 20/30-minute duration. Results for the 20-degree range are presented in the Results section, while
the 10- and 30-degree results are included in the Supplementary Material. The 30% threshold was chosen to ensure
sufficient data points for evaluating the models. The data were categorized into single release single emission (single
emission at the site and the mast was downwind of the release point), multi release single emission (multiple emissions
at the site level, but the mast was downwind of a single release point), and multi release multi emission (multiple
emissions at the site level, but the mast was downwind of more than one release point).





**2.4 Methane Emissions Quantification**
**2.4.1 Background Concentration**
Background concentration was determined for each of the sensors to calculate $CH_4$ enhancement. Due to inherent
variation in sensors that were used in this study, $CH_4$ background was calculated for each sensor separately. $CH_4$
background was calculated as the average of the lowest 5th percentile of all continuous concentration readings (US
EPA, 2013). Methane enhancement was determined as $CH_4$ concentration measurement minus the background
concentration measurement.
**2.4.2 Eddy Covariance**
Emissions were quantified using the eddy covariance method for all three emissions scenarios (single release single
emission, multi release single emission and multi release multi emission). Methane flux ($F$, kg m$^{-2}$ s$^{-1}$) was calculated
as the covariance between the vertical wind speed ($w$, m s$^{-1}$) and $CH_4$ enhancement ($c$, g m$^{-3}$) over 30 minutes (Eq. 5;
Denmead, 2008).

$$F = \overline{w'c'} \tag{5}$$

**2.4.3 Aerodynamic Flux Gradient**
Aerodynamic flux gradient quantification was also tested in all three cases. Methane flux ($F$, kg m$^{-2}$ s$^{-1}$) was calculated
based on surface friction velocity ($u_*$, m s$^{-1}$), von Kármán's constant ($k_v$, 0.41), the difference in the average $CH_4$
enhancement between the higher and lower height ($g$, m$^{-3}$), natural log of the higher and lower height, and stability
correction factors $\Psi$ (Eq. 6; Denmead, 2008; Kamp et al., 2020).

$$F = \frac{u_* k_v (c_{2-} c_1)}{In\left(\frac{z_2}{z_1}\right) - \Psi_{c,2} + \Psi_{c,1}} \tag{6}$$

**2.4.4 Determining the Area of Vertical Flux Contribution**
Eddy covariance and aerodynamic flux gradient measurements at a point (0, 0, z) generate vertical fluxes in kg m$^{-2}$ s$^{-1}$
. In this study, these fluxes represent emissions from single-point or multi-point sources distributed over an area (m$^2$).
The Kljun et al. (2015) footprint model, was used to calculate footprint, and determine the area that contributed 80%
($r = 80$, $10 \le r \le 90$) of the vertical flux measured by the eddy covariance and aerodynamic flux gradient systems. In
previous studies, 80% footprints have been used due to the difficulty of reproducing 90% of the sources under neutral
and stable conditions, where footprints tend to be long. The difference between the 80% and 90% contours is typically
excessively large, despite minimal flux contributions in that area (Rey-Sanchez et al., 2022). The Kljun et al. (2015)
model calculates footprint as a function of effective height ($z_m$ = sensor height ($z$) – displacement height ($m$)),
roughness length ($z_o$, m) / mean wind speed ($u_{mean}$, m s$^{-1}$ - used in this study), height of the boundary layer ($h$, m),
Obukhov length ($L$, m), standard deviation of the lateral velocity ($\sigma_v$, m s$^{-1}$), and friction velocity ($u^*$, m s$^{-1}$) (Kljun et
al., 2015). The roughness sublayer in the model was set to 1 (footprint is calculated even if $z_m$ is within the roughness
layer). The area of vertical flux contribution was calculated as the polygon area covered by the contour. Due to the
limitations of the flux footprint model for the measurement height and stability (Kljun et al., 2015), 20/30-minute files
flagged by the footprint model when $z_m/L < -15.5$, were excluded from further analysis.



**2.4.5 Gaussian Plume Inverse Method**
The Gaussian plume inverse method was used to quantify single release single emission and multi release single
emission. The quantified emission ($Q$, kg h$^{-1}$) was calculated from the CH$_4$ enhancement ($X$, g m$^{-3}$), wind speed ($u$, m
s$^{-1}$), horizontal dispersion coefficient ($\sigma_y$, m), vertical dispersion coefficient ($\sigma_z$, m), crosswind distance ($y$, m),
sampling height ($z$, m), emission height ($h_s$, m), and the height of the boundary layer (Equation 7; Riddick et al.,
2022b).

$$X(x,y,z) = \frac{Q}{2\pi u \sigma_y \sigma_z} e^{-\frac{y^2}{2\sigma_y^2}} (e^{\frac{-(z-hs)^2}{2\sigma_z^2}} + e^{\frac{-(z+hs)^2}{2\sigma_z^2}} + e^{\frac{-(z-2h+hs)^2}{2\sigma_z^2}} + e^{\frac{-(z+2h-hs)^2}{2\sigma_z^2}} + e^{\frac{-(z-2h-hs)^2}{2\sigma_z^2}}) \tag{7}$$

**3 Results**
**3.1 Methane Emission Quantification**
**3.1.1 Eddy Covariance**
For stable, continuous 30-minute release events, emissions calculated using the eddy covariance method were an
underestimate for single release single emission, multi release single emission and multi release multi emission events
(Figure 4). All data points were below the 1:1 line. A plot of the quantified emission versus controlled release (kg h$^{-1}$
$^{1}$) did not show a linear correlation (R$^2$ between 0.03 and 0.36), as all emissions were largely underestimated. The
eddy covariance method reported emissions of between 0 and 0.5 kg h$^{-1}$ overall, despite actual emissions being
between 0 and about 7 kg h$^{-1}$ (Figure 4). The underestimation was consistent across all downwind ranges, 10, 20 and
30 degrees (Supplementary Material Section 2.1).

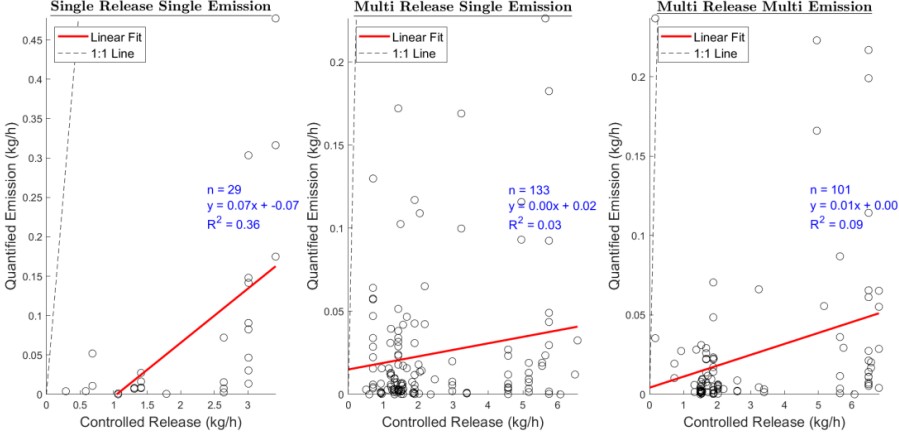


Figure 4: Quantified emission calculated using the eddy covariance method. Left pane shows a scatter plot of
quantified emission versus total controlled release for a single release at the site level and the mast was downwind of
the release point. Center pane shows a scatter plot of quantified emission versus total controlled release for multiple
releases at the site level, but the mast was downwind of a single release point. Right pane shows a scatter plot of
quantified emission versus total controlled release for multiple releases at the site level and the mast was downwind
of more than one release point. The dashed line represents the 1:1 line (points below the line were underestimated),
the red line is the linear regression fit of the data, and n is the number of data points.

### 3.1.2 Aerodynamic Flux Gradient

The aerodynamic flux gradient method also largely underestimated emissions for single release single emission, multi release single emission and multi release multi emission (Figure 5). A plot of quantified emission versus actual release did not show a linear relationship ($R^2$ between 0.01 and 0.39), and most data points were below the 1:1 line (Figure 5). The aerodynamic flux gradient quantified emissions were between 0 and about 1.6 kg h$^{-1}$ despite actual emissions being between 0 and about 7 kg h$^{-1}$ (Figure 5). The underestimation was also consistent across all downwind ranges, 10, 20 and 30 degrees (Supplementary Material Section 2.2).

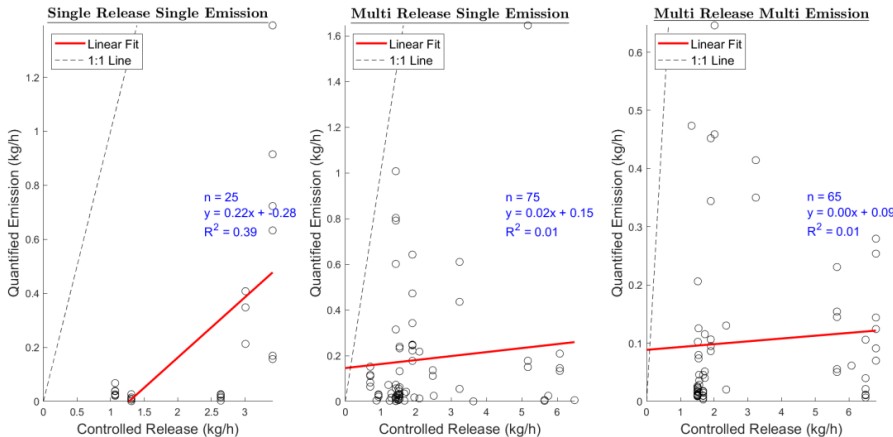

Figure 5: Quantified emission calculated using the aerodynamic flux gradient method. Left pane shows a scatter plot of quantified emission versus total controlled release for a single release at the site level and the mast was downwind of the release point. Center pane shows a scatter plot of quantified emission versus total controlled release for multiple releases at the site level, but the mast was downwind of a single release point. Right pane shows a scatter plot of quantified emission versus total controlled release for multiple releases at the site level and the mast was downwind of more than one release point. The dashed line represents the 1:1 line (points below the line were underestimated), the red line is the linear regression fit of the data, and n is the number of data points.

### 3.1.3 Gaussian Plume Inverse Method

The Gaussian plume inverse method was tested for single release single emission and multi release single emission as the method is only used for single-point sources and preliminary results showed the method provided reasonable results within 20 degrees downwind range (Figure 6; Supplementary Material Section 1.3). For single release single emission, the method quantified emissions within a factor of 1.5 (Figure 6) and showed reasonably linear relationship ($R^2$ of 0.65) (Figure 6). For multi release single emission, the gradient (m) of the linear regression was 0.95 and $R^2$ of 0.21. This suggests that the linear relationship cannot be well explained due to a random scatter of calculated emissions.





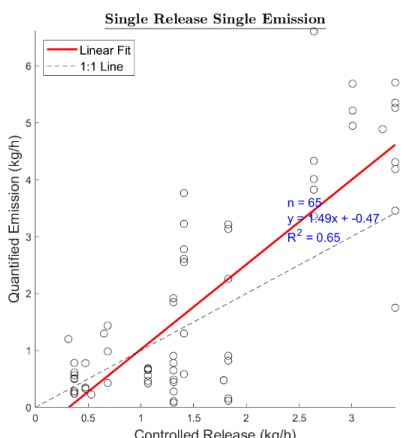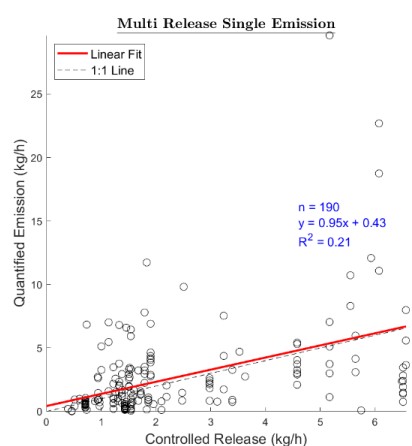


Figure 6: Quantified emission calculated using the Gaussian plume inverse method. Left pane shows a scatter plot of
quantified emission versus total controlled release for a single release at the site level and the mast was downwind of
the release point. Right pane shows a scatter plot of quantified emission versus total controlled release for multiple
releases at the site level, but the mast was downwind of a single release point. The dashed line represents the 1:1 line
(points below the line were underestimated), the red line is the linear regression fit of the data, and n is the number
of data points.
**3.2 Quantification within 30% Uncertainty**
**3.2.1 Eddy Covariance**
The eddy covariance method showed a very low probability of quantifying emissions within 30% uncertainty ($\pm$ 30%)
(Figure 7). Only a single measurement in the multi release multi emission category showed an approximately 0.01
probability of quantifying within 30% (Figure 7). The errors for eddy covariance were between -100 and -86% for
single release single emission, between -100 and -82% for multi release single emission, and between -100 and about
+30% for multi release multi emission (Figure 7). This shows that using eddy covariance to quantify single-point and
multi-point emissions will largely underestimate emissions.

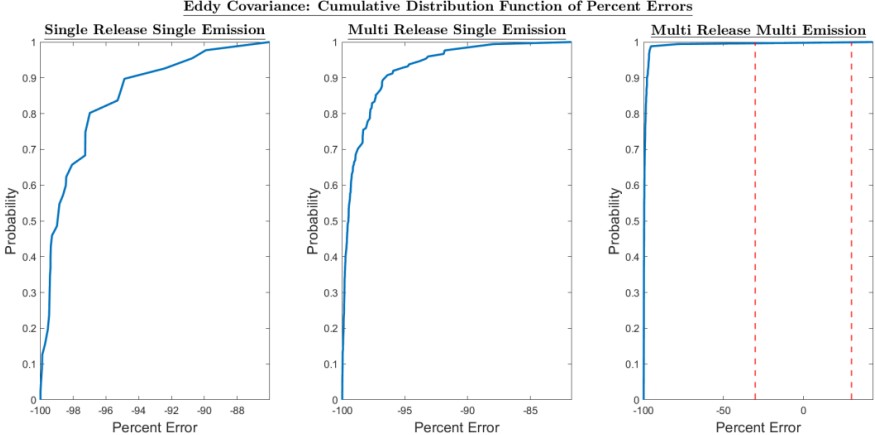


Figure 7: Cumulative distribution function (cdf) of percent errors for eddy covariance. Left pane shows a cdf plot for
a single release at the site level and the mast was downwind of the release point. Center pane shows a cdf for
multiple releases at the site level, but the mast was downwind of a single release point. Right pane shows a cdf for
multiple releases at the site level and the mast was downwind of more than one release point. The area bounded by
the red dotted line shows the region within ±30 uncertainty.

### 3.2.2 Aerodynamic Flux Gradient

The aerodynamic flux gradient also showed a very low probability of quantifying within 30% uncertainty (Figure 8).

In the multi release single emission category results indicate a 0.02 probability of quantifying within 30% (Figure 8)

of the true value. The errors for aerodynamic flux gradient were between -100 and -60% for single release single

emission, between -100 and 0% for multi release single emission, and between -100 and -70% for multi release multi

emission (Figure 8). These data show that the aerodynamic flux gradient will underestimate a point emission. Similar

to eddy covariance, quantifying an emission within 30% uncertainty using aerodynamic flux gradient for point sources

is highly unlikely.

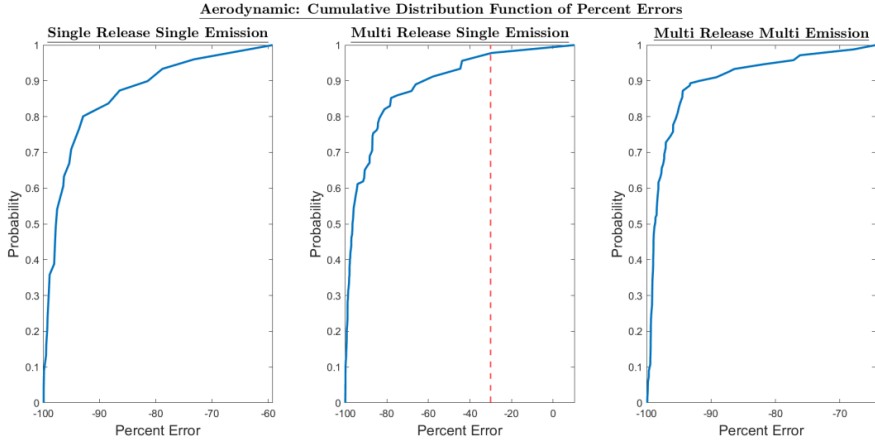


Figure 8: Cumulative distribution function (cdf) of percent errors for aerodynamic flux gradient method. Left pane
shows a cdf plot for a single release at the site level and the mast was downwind of the release point. Center pane





shows a cdf for multiple releases at the site level, but the mast was downwind of a single release point. Right pane
shows a cdf for multiple releases at the site level and the mast was downwind of more than one release point. The
area bounded by the red dotted line shows the region within ±30 uncertainty.
**3.2.3 Gaussian Plume Inverse Method**
The Gaussian plume inverse method showed a higher probability of quantifying an emission correctly within 30%
uncertainty than eddy covariance and aerodynamic flux gradient methods (Figure 9); ≈0.12 for the single release single
emission and ≈0.25 for the multi release single emission categories (Figure 9). Percent errors of the Gaussian method
calculated emissions are between -100 and +250% for single release single emission and between -100 and +800%
for multi release single emission (Figure 9). This shows that even though the Gaussian method is designed for point
sources, it is highly likely to miss, underestimate or overestimate an emission. Similar to eddy covariance and
aerodynamic flux gradient, it is a challenge to correctly quantify a single emission event (single release or multiple
release) using the Gaussian plume inverse method.

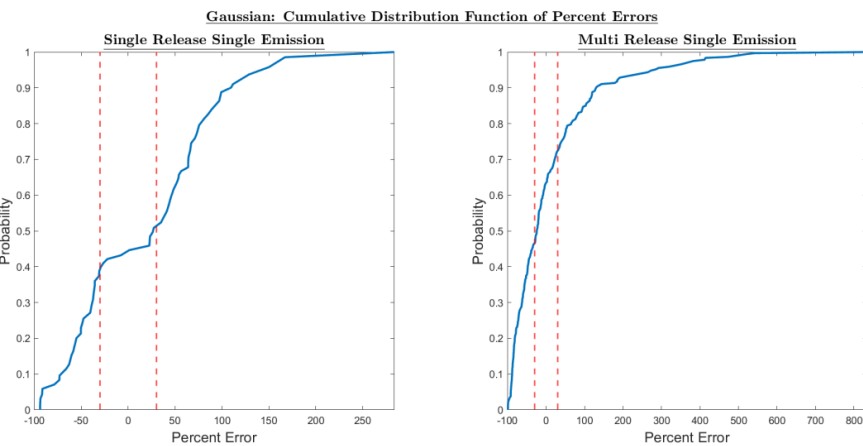


Figure 9: Cumulative distribution function (cdf) of percent errors for the Gaussian plume inverse method. Left pane
shows a cdf for a single release at the site level and the mast was downwind of the release point. Right pane shows a
cdf for multiple releases at the site level, but the mast was downwind of a single release point. The area bounded by
the red dotted line shows the region within ±30 uncertainty.
**3.3 Variables Affecting Quantification**
**3.3.1 Eddy Covariance**
A Spearman's rank correlation analysis of measurement and environmental variables (distance, controlled release,
emission height, mean wind speed ($WS$), Monin-Obukhov length ($L$) and contribution area) to percent error in
quantification as calculated by the eddy covariance method, showed that downwind distance had significant impact
on quantification for the single release single emission ($p = 4.73e-6$), multi release single emission ($p = 2.66e-4$), and
multi release multi emission ($p=2.00e-3$) categories for $p < 0.01$ significance level (Figure 10). The correlation
coefficients were -0.74 for single release single emission, -0.31 for multi release single emission, and -0.30 for multi
release multi emission. The negative correlation in all three categories suggests that the percent error became more
negative as distance increased i.e., far away emission sources were likely underestimated or undetected. Also,



controlled release and emission height had significant impact on quantification only in the multi release single emission category, p = 2.00e-3 and 9.42e-3 respectively, (Figure 10) but this correlation was inconsistent across the three categories. Due to inconsistent correlation, and the errors being close to -100%, the results show that generally, quantifying emissions using an eddy covariance approach will not work for emissions typically observed at oil and gas production sites.

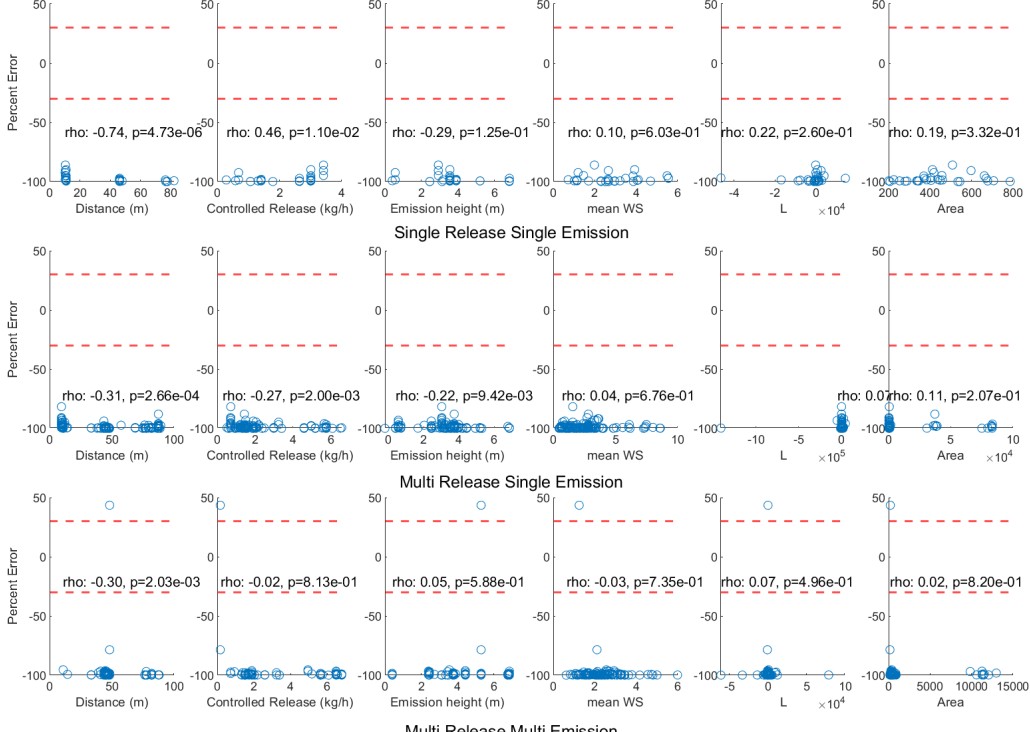

Figure 10: Correlation analysis for eddy covariance in the three release categories. The area bounded by the red dotted line shows the region within ±30 uncertainty.

**3.3.2 Aerodynamic Flux Gradient**

A Spearman's rank correlation analysis between the environmental and measurement variables and emissions calculated using the aerodynamic flux gradient method showed that only emission height in the single release single emission category had significant impact on model quantification (p = 1.79e-3) (Figure 11). The correlation between emission height and percent error in this category was -0.59 suggesting percent error became more negative as emission height increased. However, the correlation between emission height and percent error in the multi release single emission and multi release multi emission categories is approximately zero, meaning no correlation. Similar to eddy covariance, there is inconsistent correlation, and most errors are close to -100% (Figure 11). The results show that generally, quantifying emissions using an aerodynamic flux gradient approach will not work for emissions typically observed at oil and gas production sites.



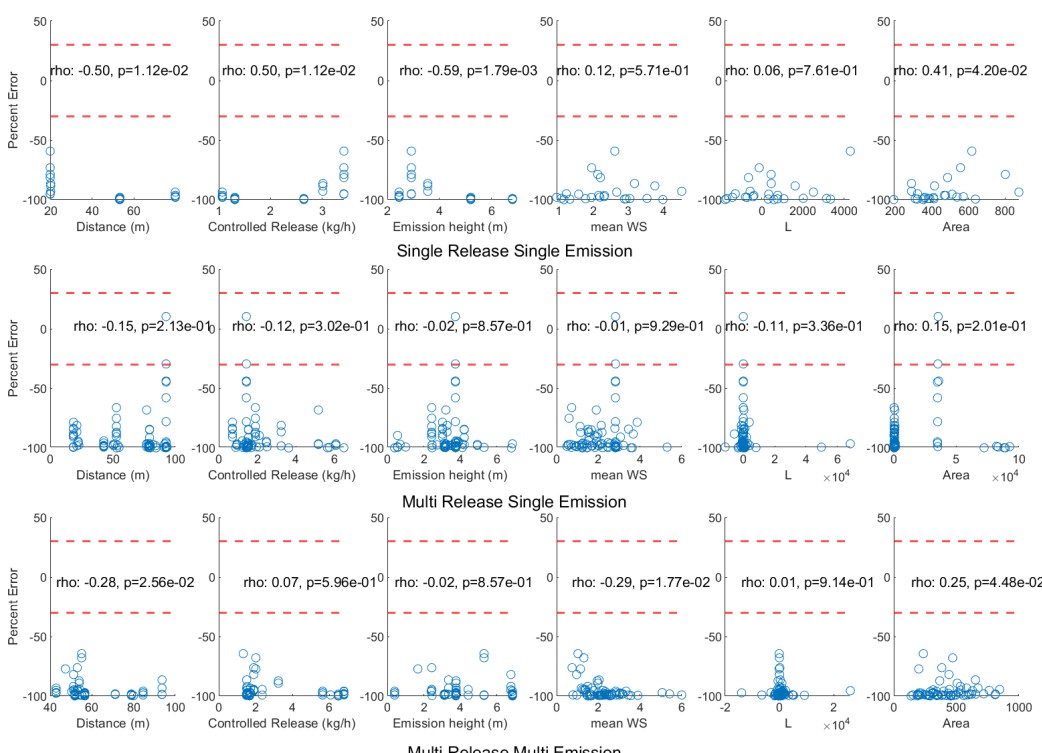


Figure 11: Correlation analysis for aerodynamic flux gradient in the three release categories. The area bounded by
the red dotted line shows the region within ±30 uncertainty.

### 3.3.3 Gaussian Plume Inverse Method

The Spearman's rank correlation analysis between the emissions calculated using the Gaussian plume inverse method
and measurement/environmental variables showed that only the mean wind speed and atmospheric stability had
significant impact on the model quantification (Figure 12). In the single release single emission category, mean wind
speed and percent error had a positive correlation (0.44, p = 2.74e-4) indicating that an increase in WS increased the
model's positive error. However, in the multi release single emission category, the correlation is opposite (a negative
correlation of -0.21, p = 3.71e-3) (Figure 12). Atmospheric stability had significant impact on model quantification
in the multi release single emission category (p = 9.15e-5) but not in the single release single emission category (Figure
12). The correlation analysis for the Gaussian plume inverse model was inconsistent suggesting random errors in
quantification. This shows that the model could either underestimate or overestimate an oil and gas emission at
random.



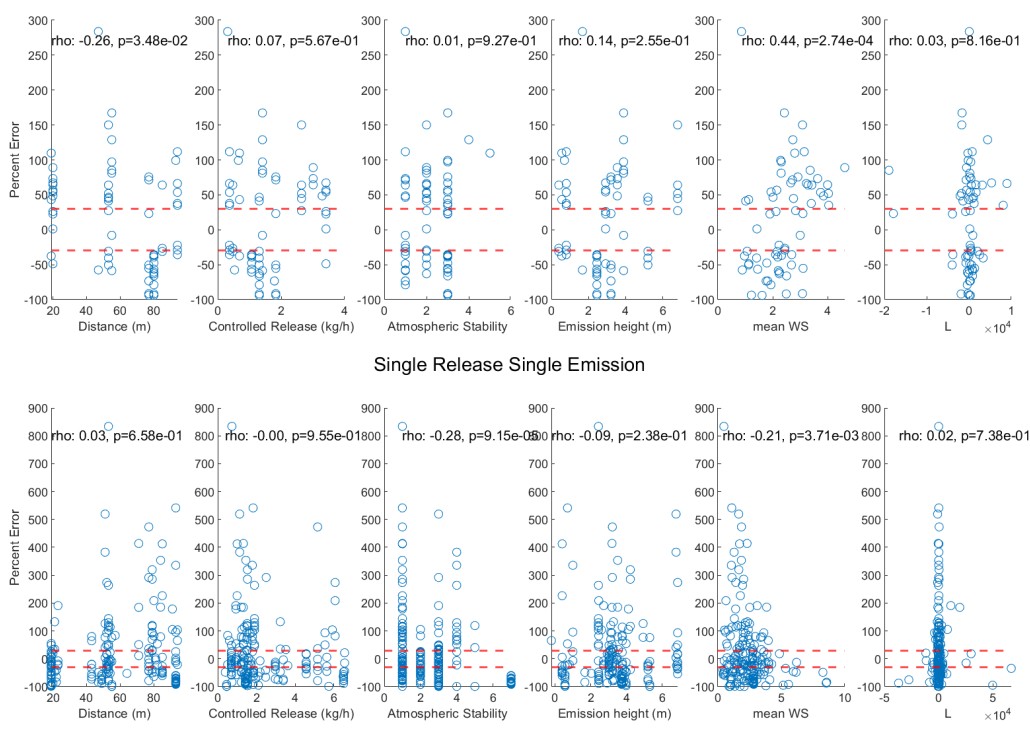

Figure 12: Correlation analysis for the Gaussian plume inverse method in the three release categories. The area
bounded by the red dotted line shows the region within ±30 uncertainty.

## 4 Discussion

Methane emissions quantification from oil and gas is a complex system comprising of gas emissions from different heights, different locations, encountering aerodynamic obstacles of different sizes, and of varying duration, amongst others. The ability to precisely quantify an emission using data collected by a point sensor, downwind of a source is directly influenced by plume dynamics. The $CH_4$ plume downwind of a source will change in size and shape in different atmospheric conditions, in open areas versus areas with obstacles, diurnally, and in different seasons (Casal, 2008). In this study, the precision to which downwind models (eddy covariance, aerodynamic flux gradient and Gaussian plume-based) could quantify the emission rate of point source(s) were tested in different atmospheric conditions (rain, sunny, snow, windy, calm etc.), and aerodynamic scenarios (emissions sources in open areas, behind obstacles, changing atmospheric stability, and day/night). As a result, testing the models' predicted emission rates to controlled release rates in different conditions introduced real-world scenarios that have not previously been tested, hence better understanding model uncertainty in the application of quantifying emissions from oil and gas production infrastructure.

### 4.1 Eddy Covariance

Eddy covariance underestimated or failed to observe almost all emissions released during this study (linear regression m between 0 and 0.07, and $R^2$ between 0.03 and 0.36) (Figure 4). The method measures $CH_4$ atmospheric fluxes for



area sources transferred as eddies of different sizes as caused by turbulence within the atmospheric boundary layer
(Babaeian and Tuller, 2023). Assumptions governing eddy covariance include: (1) the terrain is homogenous and
horizontal, (2) $CH_4$ fluxes are turbulent, (3) measurements at a point are from an upwind area, (4) measurements are
within the boundary layer and in the constant flux layer, (5) instruments can capture small fluctuations at high
frequency, (6) fluctuations in air density are negligible (Babaeian and Tuller, 2023), and (7) upward fluxes represent
emissions and downward fluxes represent depositions (Zinke et al., 2024). Nemitz et al., (2018) adds that eddy
covariance is frequently deployed to target large fluxes in high-emission ecosystems, which is not typical in oil and
gas, and that data where wind direction includes obstructed wind sectors should be flagged (Nemitz et al., 2018).
For oil and gas point sources, the measured gas concentration is dependent on plume dynamics as opposed to mass
transfer and eddy covariance methods using fence-line measurements are unlikely to work because:

- Oil and gas point sources violate assumptions (1), (2), and (4) as these sources are heterogenous and emissions are collimated plumes instead of turbulent fluxes.
- As the measurement by a point sensor is dependent on being inside the plume, which changes in different atmospheric conditions, placing the sensor high enough, and/or far enough downwind, to where the flux layer is constant, is impractical.
- Even though current eddy covariance application assumes the vertical flux at a point is independent of atmospheric stability (Denmead, 2008), atmospheric stability has impact on point source gas dispersion at fence line distances and hence needs to be accounted for even for eddy measurements.
- Footprint models are designed for area sources that require horizontal homogeneity of the flow (Kljun et al., 2015). As a result, the area of contribution generated by the models do not accurately represent the area between the point sources and the measurement location at fence line distances.

In summary, this study shows that eddy covariance is not applicable for oil and gas point source quantification.

**4.2 Aerodynamic Flux Gradient**

Overall, aerodynamic flux gradient method underestimated the emission rate of all controlled releases during this
experiment with high variability. The slope of the linear regression and $R^2$ were both very small (linear regression m
between 0 and 0.22, and $R^2$ between 0.01 and 0.39) (Figure 5). The aerodynamic flux gradient model quantification
is used to quantify emissions from area sources and relies on differences in $CH_4$ concentrations between the higher
and lower height, and stability correction factors. Assumptions of flux-gradient approach using Monin-Obukhov
similarity theory include: (1) measurements require steady state conditions of wind direction and speed, (2)
measurements should be done above the roughness sub-layer, (3) sufficiently large homogenous area for development
of an adequately equilibrated layer of air, and for constant equilibrium during measurement (Prueger and Kustas,
2015), and (4) positive fluxes represent emissions and downward fluxes represent absorptions (Kamp et al., 2020).
Similar to eddy covariance, aerodynamic flux gradient methods at fence-line distances are unlikely to work because
point sources typical of oil and gas emissions violate the following assumptions:

- Obstacles at an oil and gas facility affects wind direction and speed, and these impacts may also vary substantially with small changes in wind direction. Therefore, wind conditions are unlikely to attain steady state during the measurement period, as directed by assumption (1) above.





• The emission height of oil and gas sources in typical upstream field conditions can be as low as 0.4 m
and as high as 6.9 m and measurements are unlikely to be made by fence-line sensor above the roughness
sublayer (2 above), i.e. twice the height of the mean obstacle height for ~30 m downwind.
• Oil and gas sources are heterogeneous (i.e. varying source distance and height) and can last a short time
(e.g. a short maintenance event) or a long time ('normal' fugitive emissions) hence, achieving constant
equilibrium, as stated in (3) above, is unlikely.
• Footprint models used to generate the area of contribution between the source and the measurement
location are designed for area sources with horizontal flow homogeneity (Kljun et al., 2015). Thus, the
area of contribution generated for oil and gas point sources is likely inaccurate.

**4.3 Gaussian Plume Inverse Method**
In contrast to the other methods in this study, the Gaussian plume inverse model both underestimated and
overestimated emissions in this study. Linear regression gradient and coefficient of correlation (m between 0.95 and
1.49, and $R^2$ between 0.21 and 0.65; Figure 6) was better than either eddy covariance or aerodynamic flux gradient.
The main assumption of the Gaussian plume model is that $CH_4$ emitted from a point source enters the air flow,
disperses vertically and laterally, forming a conical plume (Riddick et al., 2022b; US EPA, 2013). However, the
formation of a conical plume is hindered at oil and gas facilities by obstacles (equipment) and is affected by
atmospheric stability. Atmospheric stability in the Gaussian plume inverse model is based on Pasquil-Gifford
classification system which accounts for daytime solar insolation (slight, moderate and strong), nighttime cloud cover
and surface wind speed at 10 m (Kahl and Chapman, 2018). Solar insolation and cloud cover are not typically
measured, and if measured, dispersion parameter models currently available do not use this data, therefore, it is
difficult to calculate for continuous fence-line measurements. The modified dispersion parameters developed by EPA
(US EPA, 2013) only account for wind conditions i.e., speed and deviation in direction. As a result, plume dynamics
during diverse atmospheric conditions such as during snow versus rain or sunny conditions are unaccounted for.
In this study, despite the Gaussian model having been developed for point sources, the model did not show consistent
correlation with the measurement and atmospheric variables. This showed that there are complexities in continuous
monitoring quantification compared to survey solutions where the model is widely applied, that introduce significant
uncertainties in quantification. It is suggested that one problem with the Gaussian plume model is that the dispersion
coefficients are simply not representative as they were developed for longer distances, in different climatological
conditions, and do not transfer well to current applications (Riddick et al., 2022a). We conclude that, while it is better
suited than eddy covariance or aerodynamic flux gradient, a Gaussian plume inverse approach will likely have
significant uncertainties when used to quantify emissions from oil and gas production sites using data collected at a
fence line (~ 30 m away).
**4.4 Implications**
In the recent years, there has been growing interest and need for accurate $CH_4$ quantification from oil and gas sites.
This is generally done through survey methods and continuous monitoring using fence-line sensors. Continuous
monitoring involves having stationary sensors measuring meteorology and $CH_4$ mixing ratios, which are then used to
infer emission rate. For point sources, downwind methods such as the Gaussian plume inverse method have been



widely used, especially for survey quantification. Continuous monitoring is relatively new but fast growing. This
study's design replicated a continuous monitoring setup.
Oil and gas point sources could either be single emissions or multiple emissions occurring concurrently. In cases of
multiple emissions with more than one release point being downwind, the Gaussian model is limited, as it can only
quantify one source at a time (dispersion coefficients are generated as a function of emission height and source
distance). As a result, models used in other applications such as eddy covariance and aerodynamic flux gradient have
been proposed as the solution. However, as this study has shown, eddy covariance and flux gradient approaches are
unlikely to quantify realistic emission estimates using fence-line measurements. Here, we strongly advise that
controlled tests under controlled environments are crucial to evaluate modelling approaches' precision and accuracy,
and associated uncertainties before applying them in the real world. Even though these modelling approaches have
been reported to work elsewhere (e.g., agricultural and landfill emissions), it does not necessary mean it could work
in the intended area of application.
**Author contributions**
Mercy Mbua: Conceptualization, Data curation, Methodology, Formal analysis, Investigation, Writing original draft,
Review, and Editing. Stuart N. Riddick: Conceptualization, Methodology, Review, Editing, Project Administration,
Supervision, and Funding Acquisition. Elijah Kiplimo: Investigation, Review, and Editing. Daniel J. Zimmerle:
Review, Editing, Funding acquisition, and Project administration.
**Declaration of competing interest**
The authors declare that they have no known competing financial interests or personal relationships that could have
appeared to influence the work reported in this paper.
**Acknowledgment**
This work is funded by the Office of Fossil Energy and Carbon Management within the Department of Energy as part
of the Site-Air-Basin Emissions Reconciliation (SABER) Project #DE-FE0032288. Any opinions, findings,
conclusions, or recommendations expressed herein are those of the authors and do not necessarily reflect the views of
those providing technical input or financial support. The trade names mentioned herein are merely for identification
purposes and do not constitute an endorsement by any entity involved in this study. The authors also thank Ryan
Brouwer, Daniel Fleischmann, Ryan Buenger, and Wendy Hartzell for their assistance.
**Data availability**
Data sets for this research are available in the in-text data citation reference: Mercy, Mbua; Riddick, Stuart N.;
Kiplimo, Elijah, and Zimmerle, Daniel J. Dataset for evaluating the accuracy of downwind methods for quantifying
point source emissions. [Dataset]. Dyrad.

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
