# Peer review of "Evaluating the feasibility of using downwind methods to quantify point source oil and gas emissions using continuous"

_EGUsphere, 2024_

## Community Comment (CC1)

Comments on Point Source emissions via Eddy Covariance etc

It is somewhat of a surprise that the eddy covariance and aerodynamic flux gradient methods were tested as fence line methods for point source emissions. Both of these micrometeorological methods are based on the atmospheric concentration conservation equation (atmospheric diffusion equation) applied to horizontally homogenous conditions. The diffusion equation can be written in the Reynolds averaging form as

$$\frac{\partial C}{\partial t} + U_j \frac{\partial C}{\partial x_j} + \frac{\partial \overline{u_j' c'}}{\partial x_j} = R + D + S$$

(1)   (2a,b)  (3a,b)      (4)    (5)   (6)

(1)     Change of pollutant concentration

(2a,b) Horizontal and vertical advection

(3a,b) Horizontal and vertical turbulent diffusion

(4)     Chemical reactions-sources/sinks

(5)    Deposition—sinks

(6)    Emissions--sources

The assumptions for eddy covariance and related methods include steady state and horizontally homogeneous source areas with no chemistry and no deposition. This yields

$$\frac{\partial \overline{w'c'}}{\partial z} = S$$

Or

$$\overline{w'c'} = \int S dz$$

However, for a steady constant point source, the horizontal advection and turbulent diffusion terms do not disappear so that the measured eddy covariance flux term (w'c') only represents a portion of the source and doesn't account for the horizontal advection or diffusion terms. The appropriate equation for a point source can be written as

$$U \frac{\partial C}{\partial x} + \left[ \frac{\partial v'c'}{\partial y} + \frac{\partial w'c'}{\partial z} \right] = S$$

This equation explicitly treats the transport due to turbulent diffusion as the plume spreads horizontally. As such, measuring only the vertical eddy covariance will miss this horizontal spread and underestimate the emission source. This is also true for the aerodynamic flux gradient method since it is based on the same set of equations.

In this paper, both methods are shown to underestimate the methane emission rate which is consistent with the fact that the micro-met procedures are ignoring horizontal plume transport and diffusion. This is also consistent with the fact that the inverse Gaussian method yields better results since it is based on the point source version of the concentration conservation equation.

The authors need to address the fact that the micro-met methods are not appropriate for point source emissions and/or develop a way to account for the effects of horizontal plume spread that can make up the amount of underestimation. One approach would be to use the measured turbulence data to improve the diffusion coefficients used in the Gaussian inverse method or use the measured turbulence data with a Lagrangian stochastic model as an alternative to the Gaussian plume method.

---

## Author Response (AR1)

Ms. Ref. No.: EGUSPHERE-2024-3161

Title: Evaluating the feasibility of using downwind methods to quantify point source oil and gas emissions using

continuous monitoring fence-line sensors

The Powerhouse Energy Campus

Colorado State University

430 North College Avenue

Fort Collins, CO 80524

E-mail: Mercy.Mbua@colostate.edu

March 18th, 2025

Dear Professor Presto,

We appreciate the time and effort you, the reviewers and community dedicated to providing feedback on our

manuscript. We have closely followed the suggestions and have revised the manuscript accordingly. We believe the

revisions have improved our manuscript and that it is ready for publication. We have listed the original comments and

our response in blue below for our reply to comments. All figure numbers, tables, and lines refer to the updated

for available manuscript. Data reviewers' is at Reviewer

URL: http://datadryad.org/stash/share/7s42bajRb2czaY9hr6NbTKDAolCWZ pNrty0o54qXBE. The authors would

also like to emphasis that the paper aims to inform real-world oil and gas fence-line sensors deployment on feasible

quantification approaches as oil and gas emissions can be complex (multiple sources could be emitting at the same

time). As a result, we have accordingly modified our manuscript and title to "Evaluating the feasibility of using

downwind methods to quantify point source oil and gas emissions using continuous monitoring fence-line sensors".

We look forward to hearing from you.

Please find our detailed responses below.

Yours sincerely,

Mercy Mbua (corresponding author)

and co-authors: Stuart N. Riddick, Elijah Kiplimo, Kira B. Shonkwiler, Anna Hodshire and Daniel Zimmerle

**Reviewer comments**

**Eben Thoma (RC1)**

**General comments**

The authors compare three approaches to quantify point source emissions (Gaussian, eddy covariance, and aerodynamic flux gradient methods). This manuscript is not recommended for publication in current form. Primary concerns center on the method application, transparency of the experiment conditions and data corrections, and quality assurance (QA). In addition to the comments provided here, the authors are encouraged to consider CC1.

**Author's response:**

Thank you for reviewing our paper. We have accordingly addressed the concerns in the specific comments below.

**Comments on Main Paper:**

**Comment 1**

It is assumed that this work was opportunistically executed as part of other controlled release activities performed by METEC to test, for example, fence line sensor network leak detection capability. If this is true, this point should be made clear to the reader as this may represent a study limitation. The controlled release profiles are complex and a primary issue with this work resides in the QA of the individual release trails in the context of this comparison. Even when the mean wind direction during a 20-minute or 30-minute observation is acceptable, a short duration release event (as low as 10 seconds, line 148) producing a spatially underdeveloped plume may yield zero methane signal enhancement on the instruments, as the release could have occurred while winds were off axis. If there is insufficient sampling of the plume for a given trial, the emission calculation performance is irrelevant. A similar argument holds for the 2m and 4 m vertical positions. The work would be improved by adding additional detail on the controlled release trials along with methane concentration statistics (including background corrections) by trial for each instrument. Ideally this could occur on a uniform 20-minute time base so direct comparisons across the techniques are possible.

**a) Complex controlled release profiles; QA of individual release trails**

**Author's response:**

Thank you for your comment. Measurements were conducted as part of the METEC Spring 2024 Advancing Development of Emissions Detection (ADED) Campaign, as this study aimed to investigate the accuracy of downwind quantification models in complex oil and gas settings for their application in continuous monitoring solutions using point sensors. We have reanalyzed our data to uniform 5 minutes, 10 minutes and 15-minute durations for all models where there were continuous durations. Even though there were short releases such as 10 s that represent intermittent emissions in oil and gas such as a venting of a gas pneumatic, our duration tables represent minute durations when there were continuous releases for that period. For example, a 5-minute event table is for a continuous release in a 5-minute duration. To clarify this, the following changes have been made to the manuscript.

**Changes to manuscript:**

Section 1

Lines 131-149:

Continuous monitoring of CH4 emissions using fence line sensors requires proper quantification of intermittent and persistent releases from oil and gas during all release (complex emission profiles) and atmospheric conditions (unstable, neutral and stable). Oil and gas emissions are characterized by intermittent, non-uniform, single or multiple point source emissions, varying in leak size, location, height and distance between the source and sensor, and are typically in complex aerodynamic environments (i.e. not flat). An ideal quantification model should always quantify emissions and should capture short and long-lasting emission events. Most models have been validated to work best during neutral conditions for single point sources. However, it is important to test and apply these models during non-neutral conditions as well as these are part of real-world conditions where continuous monitoring is applied. In this study, we evaluate if using a readily available CH4 cavity ring down analyzer for models' quantification such as the closed-path EC is a feasible solution to quantify point source emissions.

This study aims to inform the feasibility of downwind quantification models in oil and gas settings by investigating which models are likely to work most of the time with instrumentation that is typically available for fence-line deployment. Fence-line sensor deployments involve multiple sensors, continuously running in all conditions and providing emissions data. Using robust releases and environmental conditions, this study aims to investigate the performance of these methods in quantifying emissions for known gas release rates and evaluating uncertainties that could result in incorrect CH4 reporting. Specifically, the study will (1) evaluate the overall quantification accuracy of closed-path EC, AFG, bLs model, and the GPIM method in quantifying single-release single-point and multi-release single-point emissions that simulate oil and gas emissions, (2determine the mean relative factor (estimated emissions over actual emission) for these models.

Section 2.2

Lines 188-189: Controlled releases were part of the METEC Spring 2024 Advancing Development of Emissions Detection (ADED) Campaign conducted between February 6 and April 29, 2024 (Colorado State University, 2024). Section 2.4.1

Lines 221-223: The concentration-meteorological-release event tables were split into 5, 10 and 15-minute release event tables (i.e. there was a continuous release in the duration).

**Comment 2**

As it is currently written, there are issues with the eddy covariance data collection, pre-processing, flux calculations, post-processing, and footprint modeling that will cause erroneous results. These are detailed below: The authors use a non-standard system and instrument configuration for eddy covariance. These choices have significant consequences for flux quantification that need to be addressed and justified.

**Author's response:**

Thank you for your comment. We have accordingly reanalyzed our data using EddyPro software and clearly stated our system configuration and pre-processing of the data using our non-standard eddy covariance system.

Changes to manuscript:

Section 2.4.1

Lines 205-246:

**2.4.1 Eddy Covariance**

**2.4.1.1 Data pre-processing**

Evaluating the MGGA CH4 data showed that actual sampling was between 4 and 12 Hz (highest sampling at 6 Hz), even though it had been set to sample at 10 Hz (Supplementary Information Section 2b). To account for this sampling variability, data were filtered to when sampling was equal/greater than 8 Hz. Data where the frequency was greater than 8 Hz were down sampled to 8 Hz. The sonic anemometer meteorological data (horizontal wind vectors (u, v), vertical wind vector (w), temperature (T), and pressure (P)) actual sampling varied between 7 and 9 Hz with the most frequency at 8 Hz (Supplementary Information Section 2b). As the MGGA gas analyzer and sonic anemometer were not designed to clock synchronously, using the MGGA CH4 clock time as a reference, meteorological data from the sonic anemometer were matched to the MGGA CH4 data using linear interpolation to generate concentration-meteorological 8 Hz data.

The aggregated concentration-meteorological data were then merged with METEC's release data and metadata, and release event tables created. Release event tables were aggregated tables of concentration, meteorology and release (emission source location, duration and rate) information for all defined release events at METEC. The concentration-meteorological -release event data were then separated into single-release and multi-release events. Single-release events were when there was a single emission point at the site level, while multi-release events were when there was more than one emission point at the site level. The concentration-meteorological-release event tables were split into 5, 10 and 15-minute release event tables (i.e. there was a continuous release in the duration). Based on the bearing of the emission point to the measurement point and the average wind direction in the duration, the data was further filtered to downwind data,  $\pm 5^{\circ}$ ,  $\pm 10^{\circ}$ ,  $\pm 20^{\circ}$ , and  $\pm 45^{\circ}$ .

**2.4.1.2 Flux calculation**

Turbulent fluxes were calculated using the open software EddyPro® version 7 (LI-COR, Nebraska, USA, n.d.). Acquisition frequency was set at 8 Hz, while file duration and the flux interval were set at 5, 10, and 15 minutes, respectively, depending on the file being processed. Table 1 shows the instruments input to the software.

Table 1. Anemometer and Gas Analyzer Input into EddyPro

| Anemometer               |           | Gas                  | Analyzer |                     |
|--------------------------|-----------|----------------------|----------|---------------------|
| Information              |           | Information          |          |                     |
| Manufacturer             | Young     | Manufacturer         |          | Other               |
| Model                    | 81000     | Model                |          | Generic closed path |
| Height                   | 3 m       | Tube length          |          | 300 cm              |
| Wind data format         | u, v, w   | Tube inner diam      | eter     | 3.275 mm            |
| North alignment          |           | Nominal tube flo     | ow rate  | 3.2 l/m             |
| North off-set            | 0.0       | Northward separation |          | 0.00 cm             |
| Northward separation     | Reference | Eastward separation  |          | 0.00 cm             |
| Eastward separation      | Reference | Vertical separation  |          | -10.00 cm           |
| Vertical separation      | Reference | Longitudinal pat     | h length | 10.00 cm            |
| Longitudinal path length |           | Transversal path     | length   | 2.54 cm             |
| Transversal path length  |           | Time response        |          | 0.4 s               |

In raw data processing, axis rotations for tilt correction under wind speed measurement offsets was checked. Under turbulent fluctuations, double rotation and block average detrend methods were used. Covariance maximization with default was used for time lag detection; time lags detection was checked. Compensation for density fluctuations (Webb-Pearman-Leuning terms) was unchecked as the MGGA analyzer synchronously reported dry CH4 and water mole fractions, cell temperature and pressure. Mauder and Foken (2004) (0-1-2 system) were used for quality check. All statistical tests for raw data screening, Vickers and Mahrt (1997)— spike count/removal, amplitude resolution, drop-outs, absolute limits, skewness and kurtosis, discontinuities, time lags, angle of attack and steadiness of horizontal wind were checked. The default values for all these tests were used. Similarly, default settings for spectral analysis and corrections were used. Analytic correction of high-pass filtering effects (Moncrieff et al., 2005) for low frequency range; and correction of low-pass filtering effects (Fratini et al., 2012 - In situ analytic) and instruments separation (Horst and Lenschow, 2009 - only crosswind and vertical) in the high frequency range were used.

**2.4.1.3 Post-processing**

During post-processing, flux data were filtered based on (1) quality flags, Mauder and Foken (2004) (0-1-2 system), and (2) surface friction velocity ( $u^* > 0.13 \text{ m/s}$ ). Data that were flagged "2" were first filtered out as they were considered poor quality fluxes (LICOR, 2025), and the remaining dataset were filtered for high turbulence data. All data was filtered out as low quality and no further post-processing was done.

**Comment 3**

The gas analyzer (MGGA) used here and 3D sonic anemometer are not designed to be logged together with clocks synced. Properly synced clocks at 10 Hz is an essential step and possible large source of error in data acquisition, and the authors need to address how they achieved this with their non-standard system. For example, LICOR and Campbell Scientific instrument systems are designed to log all fast data (10 Hz) simultaneously on the same data logger or microcomputer with PIP LAN networking to ensure clocks remain synced.

**Author's response:**

Thank you for your comment. We have added this to our manuscript.

**Changes to manuscript:**

Section 2.4.1.1

Lines 212-215:

As the MGGA gas analyzer and sonic anemometer were not designed to clock synchronously, using the MGGA CH4 clock time as a reference, meteorological data from the sonic anemometer were matched to the MGGA CH4 data using linear interpolation to generate concentration-meteorological 8 Hz data.

**Comment 4**

The authors use a closed-path sampling system with a long (3 meter) tube. This choice creates significant errors that need to be addressed and corrected, if possible. Eddy covariance measurements require at least 10 Hz system response time in order to properly characterize turbulent eddies. Sampling at 10 Hz is not the same as a system response of 10 Hz. Closed path systems introduce a time-lag, laminar flow issues, and lack the response time of open-path systems. Slower system response is one of the main reasons the vast majority of methane eddy covariance is done using open-path sensors, and when closed-path sensors are used the system performance needs to be optimized and carefully justified. A well designed closed-path system with sufficient pump speed might effectively have a system response of 5 Hz while sampling at 10 Hz. This leads to an under estimation of fluxes, and in the best case scenarios can be corrected using transfer functions. At a minimum, the authors need to investigate the cospectra of their results to demonstrate they have a sufficient overall system response and properly apply transfer functions to correct their data. Related to this issue, when using a closed-path system the pump speed needs to be reported. With a longer tube such as this, a very fast pump speed, possibly also with a vacuum induced, is required in order to have a fast enough system response. Without a fast enough pump speed, the data cannot be used for eddy covariance methods. Pump speeds for closed path systems range from 8 l/m at normal pressure for short tubes, to over 100 l/m with partial vacuum for longer tubes.

**Author's response:**

Thank you for your comment. Our system has a relatively slow pump flow rate of ~ 3 lpm and we understand this could create laminar flow issues with a 3 m tubing. However, our study aimed to investigate if we can use data from instruments readily available to us to quantify using eddy covariance. Methane concentration readings from oil and gas emissions could be up to 300 ppm. The current readily available sensors have a range of up to 25 ppm. For example, the LI-COR LI-7700 open path CH4 analyzer has a measurement range of 0 to 25 ppm at -25 °C and 0 to 40 ppm at 25°C; PICARRO G2311-f, a closed-path analyzer has an operating range of 0 to 20 ppm. The instrument used in this study is a field instrument (ABB MGGA GLA131 Series has a measurement range of 0 to 100 ppm for CH4 but can be extended to 0 to 1%). We have attempted to account for the gas flow response time (0.4 seconds), pump flow rate (3 lpm) and the long tubing (3 m) using Eddy Pro corrections of time lag detection. Based on our reanalyzed results, our eddy covariance system quantified emissions with a mean relative factor (estimated emission over actual emission) of 0.7 to 1 for single-release single-point emissions, and within a mean relative of 1 and 2.4 for multi-release single-point emissions. Our system limitations have been clearly stated in the updated manuscript.

**Changes to manuscript:**

Section 2.4.1.2

**Lines 226-229**

Turbulent fluxes were calculated using the open software EddyPro® version 7 (LI-COR, Nebraska, USA, n.d.). Acquisition frequency was set at 8 Hz, while file duration and the flux interval were set at 5, 10, and 15 minutes, respectively, depending on the file being processed. Table 1 shows the instruments input to the software.

Table 1. Anemometer and Gas Analyzer Input into EddyPro

| Anemometer               |           | Gas Analyzer             | Gas Analyzer        |  |  |
|--------------------------|-----------|--------------------------|---------------------|--|--|
| Information              |           | Information              | Information         |  |  |
| Manufacturer             | Young     | Manufacturer             | Other               |  |  |
| Model                    | 81000     | Model                    | Generic closed path |  |  |
| Height                   | 3 m       | Tube length              | 300 cm              |  |  |
| Wind data format         | u, v, w   | Tube inner diameter      | 3.2 mm              |  |  |
| North alignment          |           | Nominal tube flow rate   | 3.2 l/m             |  |  |
| North off-set            | 0.0       | Northward separation     | 0.00 cm             |  |  |
| Northward separation     | Reference | Eastward separation      | 0.00 cm             |  |  |
| Eastward separation      | Reference | Vertical separation      | -10.00 cm           |  |  |
| Vertical separation      | Reference | Longitudinal path length | 10.00 cm            |  |  |
| Longitudinal path length |           | Transversal path length  | 2.54 cm             |  |  |
| Transversal path length  |           | Time response            | 0.4 s               |  |  |

Lines 244-245: Covariance maximization with default was used for time lag detection; time lags detection was checked.

**Section 4.1**

**Lines 464-486**

Eddy covariance was tested using a closed-path analyzer, cavity ring-down spectroscopy, with a 3.2 lpm pump flowrate and a 0.4 s gas flow response time. The closed-path EC estimated emissions between a factor of 0.67 and 0.97 for SRSP emissions, and between 1.02 and 2.43 for MRSP emissions at ±45° wind sector range (Section 3.1). This was a wider uncertainty in estimated emissions than one reported by Dumortier et al. (2019), who estimated emissions at between 90 and 113% of true emission (~1.5 kg day-1) with concentrations between 2 and 3 ppm. Our study tested closed-path EC at emission rates between 0.005 and 8.5 kg h-1.

Our study's results were when the data was filtered for frequencies greater than 8 Hz, hence, largely reducing the sampled emissions. The 10 Hz sampling frequency set for this instrument was not a true 10 Hz and this could have been due to the 0.4 gas flow response time that delayed analysis of the drawn air sample in the cavity, or the 3 lpm pump flow rate for a 3 m tubing that might have varied the effective sample turnover rate. The rest of the data were flagged as low quality by Mauder and Foken (2004) (0-1-2 system), which flags based on steady state and well-developed turbulence. This could have been due to low turbulence as experiments were carried out in winter, and instrumentation limitations (low pump flow rate and asynchronous configuration of the gas analyzer and meteorological instrument).

Continuous monitoring requires deployment of multiple sensors which create limitations of cost and requires instrumentation with a wide measurement range as concentrations for oil and gas emissions can range between 0 to 250 ppm, as in this study (Supplementary Information Section 1). The currently available EC instruments have a narrow measurement range (LI-COR LI-7700 open path CH4 analyzer has a measurement range of 0 to 25 ppm at -25 °C and 0 to 40 ppm at 25°C; PICARRO G2311-f, a closed-path analyzer has an operating range of 0 to 20 ppm). Also, the instrumentation should be environmentally robust and not lab-grade (be able to run smoothly in adverse weather conditions). Given these parameters, market available EC instruments that can currently be deployed in oil and gas are limited. The instrument used in this study is a field instrument (ABB MGGA GLA131 Series has a measurement range of 0 to 100 ppm for CH4 but can be extended to 0 to 1%).

**Comment 5**

The authors have not implemented numerous steps that are standard in the pre-processing of eddy covariance data. There are community standards for the QA/QC of the 10-20 Hz fast data before it is used for flux calculations. These include but are not limited to: Spike removal, absolute limits thresholding, corrections for skewness and kurtosis, dropouts removal, amplitude resolution, time lag corrections (absolutely critical for closed path systems such as this), steady-state tests and the mean removal through block averaging or detrending, and coordinate rotation or planar fit. Author's response:

Thank you for your comment. We have reprocessed our data through these statistical tests using EddyPro and has been included in the manuscript.

**Changes to manuscript:**

Lines 230-241: In raw data processing, axis rotations for tilt correction under wind speed measurement offsets was checked. Under turbulent fluctuations, double rotation and block average detrend methods were used. Covariance maximization with default was used for time lag detection; time lags detection was checked. Compensation for density fluctuations (Webb-Pearman-Leuning terms) was unchecked as the MGGA analyzer synchronously reported dry CH4 and water mole fractions, cell temperature and pressure. Mauder and Foken (2004) (0-1-2 system) were used for quality check. All statistical tests for raw data screening, Vickers and Mahrt (1997)—spike count/removal, amplitude resolution, drop-outs, absolute limits, skewness and kurtosis, discontinuities, time lags, angle of attack and steadiness of horizontal wind were checked. The default values for all these tests were used. Similarly, default settings for spectral analysis and corrections were used. Analytic correction of high-pass filtering effects (Moncrieff et al., 2005) for low frequency range; and correction of low-pass filtering effects (Fratini et al., 2012 - In situ analytic) and instruments separation (Horst and Lenschow, 2009 - only crosswind and vertical) in the high frequency range were used.

**Comment 6**

There are additional steps during the flux processing the authors do not implement. These include: high-pass and low-pass corrections, transfer functions (absolutely required given their system configuration), and WPL corrections. WPL corrections might not be necessary for a closed path system if the instrument is measuring methane, water, cell temperature, and cell pressure at synced 10 Hz rate. If this is the case, the reported dry mixing ratio of methane can

be used without WPL corrections. Since it can be the case that methane and water are measured at 10 Hz but cell temperature and pressure are not, the authors need to report how their instrumentation sampling was configured in this regard. Without synced 10 Hz cell temperature and pressure, the dry mixing ratio of methane is not truly at 10 Hz, and instead the authors should use the non-dry gas measurement and apply WPL corrections.

**Author's response:**

Thank you for your comment. We have reprocessed our data EddyPro and these corrections have been included. Our instrument was measuring methane, water, cell temperature and pressure synchronously hence, we used the reported dry methane measurement.

**Changes to manuscript:**

Lines 232-234: Compensation for density fluctuations (Webb-Pearman-Leuning terms) was unchecked as the MGGA analyzer synchronously reported dry  $CH_4$  and water mole fractions, cell temperature and pressure.

Lines 237-241: Similarly, default settings for spectral analysis and corrections were used. Analytic correction of high-pass filtering effects (Moncrieff et al., 2005) for low frequency range; and correction of low-pass filtering effects (Fratini et al., 2012 - In situ analytic) and instruments separation (Horst and Lenschow, 2009 - only crosswind and vertical) in the high frequency range were used.

**Comment 7**

There are post-processed QA/QC steps the authors have not implemented. These include:

- Quality tests for developed turbulence and stationarity. These are standard across the eddy covariance scientific community.
- Friction velocity thresholding. This eliminates low turbulence times and is assessed for each site/system.
- Energy balance closure. A good energy balance closure does not necessarily mean good fluxes, but a bad energy balance closure can immediately identify issues with the sonic anemometer or instrument configuration.

**Author's response:**

Thank you for your comment. We have reprocessed out data through Mauder and Foken (2004) (0-1-2 system) quality check and filtering for surface friction velocity > 0.13 m/s. All data were flagged by quality check and friction velocity thresholding hence no further post-processing was done.

**Changes to manuscript:**

**Lines 255 - 258**

**2.4.1.3 Post-processing**

Flux data were filtered based on (1) quality flags, Mauder and Foken (2004) (0-1-2 system), and (2) surface friction velocity ( $u^* > 0.13$  m/s). Data that were flagged 2 were first filtered out as were considered poor quality fluxes (LICOR, 2025) and the remaining dataset were filtered for high turbulence data. Almost all data were filtered out and no further calculations were done.

**Comment 8**

Given the experimental design, any controlled releases with durations shorter than the averaging period used in eddy covariance flux calculations should not be used because they will by definition be non-stationary. The authors should consider using a shorter averaging period. 30 minutes is standard for ecosystem fluxes which are spatially homogeneous and change relatively slowly in time. Eddy covariance controlled release experiments commonly use shorter averaging periods in order to minimize non-stationarity issues. The authors should process their fluxes using 5 minutes, 10 minutes, and 15 minutes for example. They should then investigate the ogives to determine how much if any of the fluxes are underestimated by using a shorter averaging window. Previous controlled release experiments with eddy covariance have shown that the under sampling of large eddies by using a shorter averaging window affected flux estimates by <10% and considered this a worthwhile trade off to ensure stationarity in the observations.

**Author's response**

Thank you for your comment. We have reanalyzed our data to uniform 5 minutes, 10 minutes and 15-minute durations for all models where there were continuous durations. All data were flagged by quality check and friction velocity thresholding hence no further calculations were done.

**Changes to Manuscript**

Section 2.4.1

Lines 221-223: The concentration-meteorological-release event tables were split into 5, 10 and 15-minute release event tables (i.e. there was a continuous release in the duration).

**Lines 243-246**

**2.4.1.3 Post-processing**

Flux data were filtered based on (1) quality flags, Mauder and Foken (2004) (0-1-2 system), and (2) surface friction velocity ( $u^* > 0.13$  m/s). Data that were flagged 2 were first filtered out as were considered poor quality fluxes (LICOR, 2025) and the remaining dataset were filtered for high turbulence data. Almost all data were filtered out and no further calculations were done.

**Comment 9**

The cospectra should be investigated and included in the SI. This is a standard QA/QC step that can illuminate issues such as slow system response times, aliasing, and interference near the sonic anemometer. A good cospectra will not necessarily mean good fluxes, but a bad cospectra will certainly mean there are issues that need to be corrected.

**Author's response**

Thank you for your comment. All data were flagged by quality check and friction velocity thresholding hence no further post-processing such as cospectra investigation was done.

**Comment 10**

Footprint modeling: This is an essential step in interpreting eddy covariance data in a system with point sources. However, the authors incorrectly parameterized the model and used the footprint area to normalize the fluxes to emissions incorrectly. Firstly, the choice of a roughness length of 1 is physically improbable for their site, and given

it is a key parameter in the model, the footprints are likely incorrect. The roughness length should not be estimated or chosen randomly. It can be derived from the authors' 3D sonic anemometer data under neutral conditions. In a scenario with perfectly homogeneous fluxes surrounding the eddy covariance tower, then the footprint area can be multiplied by the flux as the authors have done here. If there is any heterogeneity present (such as a point source) then this is incorrect. The footprint area is weighted with a small area of peak influence that asymptotically declines to zero influence in all directions away from it. The correct approach, given that the location of the point source is known, is to multiply the weight of the pixel representing the point source by the measured flux, and by the area of the pixel containing the point source. See Rey-Sanchez et al. 2022 for a description of how footprints should be used to calculate point source eddy covariance emissions and compared to controlled releases. Given that the authors can likely assume a zero flux for all non-point source pixels within the footprint, this is a straightforward calculation. Lastly, the authors chose to use a footprint model which previously studies have shown to perform the worst in controlled release experiments. Rey-Sanchez et al. 2022 compared three commonly used models and at a minimum, the authors here should do so as well. The exact footprint peak influence and location matters much more for correctly calculating emissions from point sources, so the variation in this result between footprint models is a source of uncertainty that should be quantified for a methods comparison paper such as this.

**Author's response**

Thank you for your comment. We have recalculated the surface roughness and footprint.

Comment 10 – Question 1 Calculation of surface roughness

Surface roughness was calculated using the high frequency sonic anemometer data during neutral conditions and was determined to be 0.1 m.

**Changes to Manuscript:**

Section 2.3

**2.3 Calculation of Roughness Length**

Surface roughness length (z0) was calculated from friction velocity (Supplementary Information Section 2a: Equations 1 and 2) by splitting the high frequency sonic anemometer data into 15-minute tables and filtering for those in neutral conditions, |L| > 500 (Supplementary Information Section 2a: Equation 3). The overall roughness length selected as the median of all the calculated z0 was 0.1 m (Rey-Sanchez et al., 2022).

**Comment 10-Question 2: Emission calculation from the footprint for point sources**

Point source emission was calculated based on the approach by Dumortier et al (2019) where emission per source = measured flux / value of the footprint at the cell. The source location was first determined if it was within the 90% footprint region, and the footprint value extracted.

**Changes to Manuscript:**

Section 2.4.3

**Lines 271-275**

Point source emissions of sources within this region were then calculated based on the approach by Dumortier et al. (2019). This approach assumes all measured flux is equal to flux resulting from a single point source. In case of

the mast being downwind of more than one source, more sonic anemometers are needed to solve the two unknown point source fluxes.

**Comment 10 – Question 3 Footprint Choice**

Our study was limited to a single sonic anemometer which limited site-based calculations of exponential wind velocity power law and the exponential eddy diffusivity power law that are inputs to the Kormann and Meixner (2001) footprint model. Our set up provided enough inputs for the Kljun footprint model. This limitation has been stated in our manuscript.

**Changes to Manuscript**

**Lines 254-259**

Eddy covariance and AFG footprints were calculated using the Kljun et al. (2015) footprint model. Even though Rey-Sanchez et al. (2022) reported the Kljun et al. (2015) footprint model to be less accurate compared to the Kormann and Meixner (2001), Kormann and Meixner (2001) was too complex for our study because it required multiple sonic anemometers or tracer release experiments to calculate the exponential wind velocity power law, and the exponential eddy diffusivity power law for site specific data. Our study was limited to a single sonic anemometer, and this provided enough inputs for the Kljun et al. (2015) footprint model.

**Comment 11**

It is unclear with the way the methods are written how the footprint normalization for the aerodynamic flux gradient method was done. It seems as though the area of the eddy covariance footprint was used to convert the AFG emissions into kg/hr. This is incorrect. The sensors are in a different location at different heights and will not have the same footprint as the eddy covariance measurements and will additionally have the same issues of normalizing to the whole area for a point source that was described above. In general, the AFG method should not be used to measure point source emissions. There will be a different footprint for each height along the profile. This means each sensor will see (or maybe not see at all) the point source differently. This will obviously lead to issues when calculating a flux from the profile gradient. The AFG method only works in homogeneous environments where the differing footprints at different heights don't matter.

**Author's response**

Thank you for your comment. We used the sonic anemometer collocated with the eddy covariance tower to calculate the aerodynamic footprint. Even though the anemometer was 9.4 m away at 3 m height (average of the two aerodynamic flux gradient sensors, 2 and 4 m), the inputs to the Kljun et al (2015) model are atmospheric stability, friction velocity, standard deviation of v wind speed and surface roughness. We do not anticipate these variables to change significantly between the two distances. The distance to the point source in footprint calculation was from the aerodynamic flux gradient locations. Further, emissions calculation using the corrected method for point source calculations provided promising results for aerodynamic flux gradient quantification in complex emissions profiles where there were multiple emitting sources. This limitation of non-collocated sonic anemometer and the flux gradient tower has been acknowledged and recommended for consideration in future studies.

**Changes to Manuscript**

For AFG, CH4 concentration data was collected at 2 and 4 m using two Aeris (Hayward, CA, USA) MIRA Ultra Series analyzers connected to tubing with a 3.275 mm inner diameter (Figure 3-2). As we had only one sonic anemometer, data from the sonic anemometer collocated with the MGGA were used for the AFG quantification.

**Comment 12**

Line 147 "The gas release rates ranged between 0.005 kg h-1 and 8.5 kg h-1, and the release durations ranged from 10 seconds to 8 hours, simulating both fugitive and large emission events. The releases were run both during the day and night". A table in SI that describes the release experiments and the meteorological conditions to make this transparent to the reader would be very helpful. Development of a QA metric that summarized the methane concentration fields observed during each trial would provide important supporting information. As currently written, the reader cannot understand if the under performance of the techniques is related to the method or to non-representative concentration fields or instrument factors.

**Author's response**

Thank you for comment. After reanalyzing our eddy covariance data, we realized the underestimation is because the data is not suitable for eddy covariance calculations as it failed the quality test. For aerodynamic flux gradient, the underestimation was due to using the area source for footprint calculation. Correctly calculating the footprint for a point source has improved our results. We have uploaded our data and the METEC releases as well. The Reviewer link is here: <a href="http://datadryad.org/stash/share/7s42bajRb2czaY9hr6NbTKDAolCWZ\_pNrty0o54qXBE">http://datadryad.org/stash/share/7s42bajRb2czaY9hr6NbTKDAolCWZ\_pNrty0o54qXBE</a>.

**Comment 13**

Extending comment 12, because the measurements were simultaneous and almost colocated, a direct comparison of the concentrations measured would be very useful. For example, at 2 m and 4 m on the same mast, what this the ratio of concentrations before and after background correction? For proximate releases at ground level, there may be little signal at 4 m potentially invalidating method assumptions for those cases. At the eddy covariance unit position at 3 m mast height there should be some agreement in measured concentrations (for like time periods) with the other mast (perhaps the average of 2 and 4 m values). Concentration correlation plots by trial would inform both the degree of colocation of the slightly separated masts and the factors of instrument performance, apart from inverse modeling complexity.

**Author's response**

Thank you for your comment. The data from the eddy covariance mast was used for eddy covariance, backward Lagrangian stochastic and the Gaussian plume inverse model. This ensured direct comparison of these three models. The aerodynamic flux gradient was the only one calculated based on measurements from the 2 and 4 m Aeris analyzers. This study acknowledges the limitation of the sonic anemometer being 9.4 m for aerodynamic measurements. The inputs to the Kljun et al (2015) model are atmospheric stability, friction velocity, standard deviation of v wind speed and surface roughness, and these variables should not change significantly at this distance. However, with the point

source footprint calculation, the aerodynamic flux gradient performed better that the Gaussian and backward Lagrangian models for point source quantification when multiple sources were happening. We have also provided the concentration plots in the Supplementary Information Section 1.

**Changes to Manuscript**

Supplementary Information Section 1

**Comment 14**

Line 125: Out of curiosity, why were the two masts not located together to facilitate direct comparison? By the photos, this seems that it would have been an easy thing to do.

**Author's response**

Thank you for your comment. The 2 and 4 m mast was set up before the 3 m one (which could only be set up at the placed location due to internet connectivity).

**Comment 15**

Basic information on the release trials should be included in SI. For example, how many releases of each type were conducted. At what levels, duration, time of day/night, etc.

**Author's response**

Thank you for your comment. The release data has been uploaded.

**Comment 16**

Line 159-165: It is unclear why uniform 20-minute event tables were not used for all measurements to facilitate direct comparisons between the techniques for discrete experiments. The authors make the argument that for two of the

approaches, 15 min is not long enough, and 30 min may capture excess atmospheric variability but then just choose the default 30 min homogenous source eddy covariance default period without explanation. For this custom use of EC, 20-minute averages temporally aligned with the other approaches would be the preferred starting position.

**Author's response**

Thank you for your comment. We have reanalyzed our data to uniform 5 minutes, 10 minutes and 15-minute durations for all models where there were continuous durations.

**Changes to Manuscript**

Lines 221-223: The concentration-meteorological-release event tables were split into 5, 10 and 15-minute release event tables (i.e. there was a continuous release in the duration).

**Comment 17**

Line 172: Why was this form of atmospheric stability estimate utilized for the inverse Gaussian approach instead of more robust and comparative estimates derived from high frequency sonic anemometry? This stability estimate was developed to support a specific mobile source assessment approach designed to provide approximate emissions estimates with the derived stability index values limited to daytime measurement. How were nighttime data treated?

**Author's response**

Thank you for your comment. We have reanalyzed data by generating our dispersion coefficients using the sonic anemometer data. This means that dispersion parameters can be generated at all times.

**Changes to Manuscript**

Supplementary Information Section 2a

 $\sigma_y$  (m) = horizontal dispersion coefficient calculated as  $\sigma_v(x/U)$ :  $\sigma_v$  is the standard deviation of v wind speed from the sonic anemometer, x is the downwind distance from the point source and U is the mean wind speed

 $\sigma_z(m)$  = vertical dispersion coefficient calculated as  $\sigma_w(x/U)$ :  $\sigma_w$  is the standard deviation of w wind speed from the sonic anemometer, x is the downwind distance from the point source and U is the mean wind speed

**Comment 18 and 19**

Line 192: The background subtraction procedure may represent a significant issue, or at minimum requires further clarification and supporting detail. The authors reference a background correction procedure utilized for a specific mobile source assessment approach designed to provide approximate emissions estimates. That background correction procedure is based on the lowest 5th percentile of a single 15-minute observation. The authors state "CH4 background was calculated as the average of the lowest 5th percentile of all continuous concentration readings (US EPA, 2013)." If the authors combined all data for the series of experiments for this calculation, there are many issues with that approach related to drift and other factors. If the background for each 20/30 minute trial was determined, this needs to be more clearly stated. These data (the corrections) should ideally be provided as SI so the reader can better understand the fidelity of the concentration measurement itself. Currently, the concentration measurements are assumed to be accurate and precise with operation in challenging field conditions.

Related to comment (18), even if the background correction technique utilized was applied to each trial, it may be inappropriate for the inverse forms utilized. This approach can be useful for single point measures of short duration that employ high precision instrumentation (e.g. CRDS). However, is the ambient precision envelope is somewhat broad and if the calibration offset exhibits trend (drift), the background compensation for the 2m vs 4m units for the Aerodynamic Flux Gradient, for example, can have a large impact on the calculation. Currently the reader has no sense for the levels of these corrections or the basic stability and comparability of these instruments over time.

**Author's response**

Thank you for your comment. We recalculated our background concentration as lowest 5th percentile, 5 minutes before each release started to capture concentrations from the residual methane in air during the preceding release especially during stable conditions. However, in cases where this background was higher than the mean concentration in the quantifying duration, the minimum concentration for that duration was used as background. This was done to ensure we capture the background noise due to the complexity of multiple emitting points, using a constant background from a long emission event, and to capture sensor drift. The following changes were made to the manuscript.

**Changes to Manuscript**

Section 2.4.3.1

Lines 269-277

For continuous monitoring sensors, background concentration can be determined from CH4 concentrations measured by a sensor upwind of the emission source, or by sampling when the wind is blowing away from the source. However, for continuous monitoring sensors, using an upwind sensor has the limitation of missing downwind background noise resulting from emissions in the preceding emission event where there is residual CH4 in air especially during stable conditions, and capturing sensors drift in the downwind sensor. In this study, background CH4 was calculated as the average of the lowest 5th percentile, 5 minutes before each release started. In cases where this background was greater than the mean CH4 concentration in the quantifying duration, the minimum CH4 concentration for that duration was used as the background. Methane enhancement was then calculated as CH4 concentration minus the background.

**Comment 20**

The introduction and discussion lack context and comparison to previous research. There are numerous controlled release experiments and other methods comparisons for these three approaches that the authors' results should be put in context with.

**Author's response**

Thank you for your comment. We have added previous controlled studies to our introduction. The discussion section has also been rewritten.

**Changes to Manuscript**

**4.1 Eddy Covariance**

Eddy covariance was tested using a closed-path analyzer, cavity ring-down spectroscopy, with a 3.2 lpm pump flowrate and a 0.4 s gas flow response time. The closed-path EC estimated emissions between a factor of 0.67 and 0.97 for SRSP emissions, and between 1.02 and 2.43 for MRSP emissions at ±45° wind sector range (Section 3.1). This was a wider uncertainty in estimated emissions than one reported by Dumortier et al. (2019), who estimated emissions at between 90 and 113% of true emission (~1.5 kg day-1) with concentrations between 2 and 3 ppm. Our study tested closed-path EC at emission rates between 0.005 and 8.5 kg h-1.

Our study's results were when the data was filtered for frequencies greater than 8 Hz, hence, largely reducing the sampled emissions. The 10 Hz sampling frequency set for this instrument was not a true 10 Hz and this could have been due to the 0.4 gas flow response time that delayed analysis of the drawn air sample in the cavity, or the 3 lpm pump flow rate for a 3 m tubing that might have varied the effective sample turnover rate. The rest of the data were flagged as low quality by Mauder and Foken (2004) (0-1-2 system), which flags based on steady state and well-developed turbulence. This could have been due to low turbulence as experiments were carried out in winter, and instrumentation limitations (low pump flow rate and asynchronous configuration of the gas analyzer and meteorological instrument).

Continuous monitoring requires deployment of multiple sensors which create limitations of cost and requires instrumentation with a wide measurement range as concentrations for oil and gas emissions can range between 0 to 250 ppm, as in this study (Supplementary Information Section 1). The currently available EC instruments have a narrow measurement range (LI-COR LI-7700 open path CH4 analyzer has a measurement range of 0 to 25 ppm at -25 °C and 0 to 40 ppm at 25°C; PICARRO G2311-f, a closed-path analyzer has an operating range of 0 to 20 ppm). Also, the instrumentation should be environmentally robust and not lab-grade (be able to run smoothly in adverse weather conditions). Given these parameters, market available EC instruments that can currently be deployed in oil and gas are limited. The instrument used in this study is a field instrument (ABB MGGA GLA131 Series has a measurement range of 0 to 100 ppm for CH4 but can be extended to 0 to 1%).

**4.2 Aerodynamic Flux Gradient**

Overall, the AFG method quantified emissions within an MRF of 1.3 to 1.7 for SRSP emissions and between an MRF of 2.4 and 3.3 for MRSP emissions (Section 3.2). The uncertainties in AFG were higher than EC especially for MRSP emissions but lower than the GPIM and bLs methods. The differences between AFG and EC could have been due to different instrumentation and analytical approaches that limited the exact comparison of the methods i.e. EC data was filtered for frequencies less than 8 Hz, and AFG instrumentations required both analyzers to be running, periods when one analyzer was down, were not tested. To our knowledge, this is the first study to test AFG for point source quantification and results are promising.

Compared to the EC method that requires a very fast analyzer which may be difficult to deploy in oil and gas, the AFG requires at least 2 analyzers sampling at 1 Hz frequency, which is currently possible with the range of sensors available in the market. The main advantage of flux gradient methods (EC and AFG) tested in this study is that they do not require background control as background CH4 concentration is a highly variable parameter that cannot be controlled in open air especially when multiple emissions are happening. The AFG method relies on

differences in CH4 concentrations between two heights and this study shows that in complex sites where there are multiple sources, the method quantifies better than the point-source GPIM and bLs methods. The main limitation of the AFG and EC methods for point-source quantification is that when the measurement point is downwind of more than one source in a wind sector range, more than one sonic anemometers are required to estimate the flux of each source for footprint calculation based on Dumortier et al. (2019)'s calculations.

**4.3 Gaussian Plume Inverse Method**

The GPIM method quantified emissions within an MRF of 2.4 and 2.6 for SRSP and between 15.7 and 25 for MRSP emissions (Section 3.3). The GPIM method is a point-source specific quantification approach and works best in open areas, free of obstacles, and when the background concentration is well defined. For multiple emissions, in aerodynamically complex environments, even though the sensor is downwind of a single source based on average wind direction, quantification is complexed by interference from other neighboring sources. The GPIM has previously been reported to quantify emissions within 40.7 and 60% error for a single point-source, (Riddick et al., 2022b). However, GPIM correct quantification has been suggested to be better for longer distances where the plume is well mixed. This is typically a challenge for fence-line sensors that have to be deployed within the facility boundaries where large downwind distances may not be practical.

**4.3 Backward Lagrangian Stochastic Model**

The bLs method quantified emissions within an MRF of 0.8 to 1.05 for SRSP emissions and between 3.85 and 11958.8 MRF for MRSP emissions (Section 3.4). Similarly to the GPIM method, the bLs method used in this study is a point-source specific quantification method that simulates transport of molecules in open area and where the background concentration is defined. In this case, as with the SRSP test scenario, the bLs approach quantified within 20% uncertainty. However, for MRSP emissions, the bLs largely overestimated emissions and this could have been due to the interference of neighboring sources, that even though the measurement point is downwind of a single source, actual plumes are not distinct and model-simulated plumes may not be representative. The point-source bLs approach in WindTrax is also not designed for more than one downwind source.

**Ali Lashgari (RC2)**

**General comments**

This manuscript compares three methods of point source emissions quantification (eddy covariance-EC, aerodynamic flux gradient method-AFGM, and the inverse Gaussian plume method-IGPM), investigating the performance of these methods in quantifying emissions for known gas release rates and evaluating uncertainties involved in the estimates. Although this is a timely topic, I recommend a revision to the current form of the manuscript before its publication. Here are my comments:

**Authors response**

Thank you for reviewing our paper. We have accordingly addressed the concerns in the specific comments below. We have also added the backward Lagrangian point-source quantification model.

**Comment 1**

Please provide a sufficient explanation of the three employed approaches. For instance, it is unclear how the inferred rates are estimated in Figures 4-6. Providing sufficient details about the different modeling choices can make significant improvements to the paper. For instance, the equation for the inverse Gaussian plume model (equation 7) only applies to a single source. It will be helpful to explain how this approach is used to model multiple release scenarios. Another example is the definition of the downwind distance (\$x\$) under such conditions.

**Authors response**

Thank you for your comment. We have revised our manuscript accordingly. In this study, we evaluated the models for single-release single-point emissions (single emission at the site level) and multi-release single-point emissions (multiple emissions at the site level, but the mast was downwind of one source. For the eddy covariance and aerodynamic flux gradients models, to estimate footprint of each source when there is more than one source in wind sector range, more than one sonic anemometer is needed to solve the footprint equation as calculated by Dumortier et al (2019). In this study, we had a single sonic anemometer. The backward Lagrangian stochastic model in WindTrax can only simulate one upwind source at a time. As a result, we limited our comparison to these two test scenarios.

**Changes to Manuscript**

**Lines 136-139**

Specifically, the study will (1) evaluate the overall quantification accuracy of closed-path EC, AFG, bLs model, and the GPIM method in quantifying single-release single-point and multi-release single-point emissions that simulate oil and gas emissions, (2) determine the mean relative factor (estimated emissions over actual emission) for these models.

 $(1) \ Eddy \ covariance \ and \ aerodynamic \ flux \ gradient \ footprint \ calculation$

**Lines 254-265**

Eddy covariance and AFG footprints were calculated using the Kljun et al. (2015) footprint model. Even though Rey-Sanchez et al. (2022) reported the Kljun et al. (2015) footprint model to be less accurate compared to the Kormann and Meixner (2001), Kormann and Meixner (2001) was too complex for our study because it required

multiple sonic anemometers or tracer release experiments to calculate the exponential wind velocity power law, and the exponential eddy diffusivity power law for site specific data. Our study was limited to a single sonic anemometer, and this provided enough inputs for the Kljun et al. (2015) footprint model. The default pixel size 2 \* 2 m was used in this study. This study first calculated the area that contributed 90% of the vertical flux; and based on the location of the point source, the source was determined if it was within the 90% footprint area. Point source emissions of sources within this region were then calculated based on the approach by Dumortier et al. (2019). This approach assumes all measured flux is equal to flux resulting from a single point source. In case of the mast being downwind of more than one source, more sonic anemometers are needed to solve the two unknown point source fluxes.

**(2) Backward Lagrangian stochastic model**

**Lines 288-293**

Pre-processed data from the GPIM method was used for bLs quantification. Quantification was done using the open-source software WindTrax 2.0 (Crenna, 2006; WindTrax 2.0, n.d.). For every 5-, 10- and 15-minute duration in the  $\pm 5^{\circ}$ ,  $\pm 10^{\circ}$ , and  $\pm 20^{\circ}$ , respectively, inputs included roughness length (z0), Monin-Obukhov length (*L*), mean (wind speed, wind direction, concentration, pressure, temperature), background concentration, source height, and distance from the emission point to sensor. WindTrax is also designed to quantify a single point source at a time, and hence, was only used to quantify single-point single emissions and multi-point single emissions.

**Comment 2**

As part of this study, critical choices are made related to the physics embedded in the covariance w'c' (equation 5) under different scenarios that an eddy covariance (EC) tower provides. More in-depth explanations of those choices will be helpful.

**Authors Response**

We have accordingly reanalyzed our data using EddyPro software and clearly stated our system configuration and pre-processing of the data using our non-standard eddy covariance system.

**Changes to manuscript:**

Section 2.4.1

**2.4.1 Eddy Covariance**

**2.4.1.1 Data pre-processing**

Evaluating the MGGA CH4 data showed that actual sampling was between 4 and 12 Hz (highest sampling at 6 Hz), even though it had been set to sample at 10 Hz (Supplementary Information Section 2b). To account for this sampling variability, data were filtered to when sampling was equal/greater than 8 Hz. Data where the frequency was greater than 8 Hz were down sampled to 8 Hz. The sonic anemometer meteorological data (horizontal wind vectors (u, v), vertical wind vector (w), temperature (T), and pressure (P)) actual sampling varied between 7 and 9 Hz with the most frequency at 8 Hz (Supplementary Information Section 2b). As the MGGA gas analyzer and sonic anemometer were not designed to clock synchronously, using the MGGA CH4 clock time as a reference, meteorological data from the sonic anemometer were matched to the MGGA CH4 data using linear interpolation to generate concentration-meteorological 8 Hz data.

The aggregated concentration-meteorological data were then merged with METEC's release data and metadata, and release event tables created. Release event tables were aggregated tables of concentration, meteorology and release (emission source location, duration and rate) information for all defined release events at METEC. The concentration-meteorological -release event data were then separated into single-release and multi-release events. Single-release events were when there was a single emission point at the site level, while multi-release events were when there was more than one emission point at the site level. The concentration-meteorological-release event tables were split into 5, 10 and 15-minute release event tables (i.e. there was a continuous release in the duration). Based on the bearing of the emission point to the measurement point and the average wind direction in the duration, the data was further filtered to downwind data,  $\pm 5^{\circ}$ ,  $\pm 10^{\circ}$ ,  $\pm 20^{\circ}$ , and  $\pm 45^{\circ}$ .

**2.4.1.2 Flux calculation**

Turbulent fluxes were calculated using the open software EddyPro® version 7 (LI-COR, Nebraska, USA, n.d.). Acquisition frequency was set at 8 Hz, while file duration and the flux interval were set at 5, 10, and 15 minutes, respectively, depending on the file being processed. Table 1 shows the instruments input to the software.

Table 1. Anemometer and Gas Analyzer Input into EddyPro

| Anemometer               |           | Gas Analyzer             |                     |  |
|--------------------------|-----------|--------------------------|---------------------|--|
| Information              |           | Information              |                     |  |
| Manufacturer             | Young     | Manufacturer             | Other               |  |
| Model                    | 81000     | Model                    | Generic closed path |  |
| Height                   | 3 m       | Tube length              | 300 cm              |  |
| Wind data format         | u, v, w   | Tube inner diameter      | 3.275 mm            |  |
| North alignment          |           | Nominal tube flow rate   | 3.2 l/m             |  |
| North off-set            | 0.0       | Northward separation     | 0.00 cm             |  |
| Northward separation     | Reference | Eastward separation      | 0.00 cm             |  |
| Eastward separation      | Reference | Vertical separation      | -10.00 cm           |  |
| Vertical separation      | Reference | Longitudinal path length | 10.00 cm            |  |
| Longitudinal path length |           | Transversal path length  | 2.54 cm             |  |
| Transversal path length  |           | Time response            | 0.4 s               |  |

In raw data processing, axis rotations for tilt correction under wind speed measurement offsets was checked. Under turbulent fluctuations, double rotation and block average detrend methods were used. Covariance maximization with default was used for time lag detection; time lags detection was checked. Compensation for density fluctuations (Webb-Pearman-Leuning terms) was unchecked as the MGGA analyzer synchronously reported dry CH4 and water mole fractions, cell temperature and pressure. Mauder and Foken (2004) (0-1-2 system) were used for quality check. All statistical tests for raw data screening, Vickers and Mahrt (1997)— spike count/removal, amplitude resolution, drop-outs, absolute limits, skewness and kurtosis, discontinuities, time lags, angle of attack and steadiness of horizontal wind were checked. The default values for all these tests were used. Similarly, default settings for spectral analysis and corrections were used. Analytic correction of high-pass filtering effects (Moncrieff et al., 2005) for low

frequency range; and correction of low-pass filtering effects (Fratini et al., 2012 - In situ analytic) and instruments separation (Horst and Lenschow, 2009 - only crosswind and vertical) in the high frequency range were used.

**2.4.1.3 Post-processing**

During post-processing, flux data were filtered based on (1) quality flags, Mauder and Foken (2004) (0-1-2 system), and (2) surface friction velocity ( $u^* > 0.13 \text{ m/s}$ ). Data that were flagged "2" were first filtered out as they were considered poor quality fluxes (LICOR, 2025), and the remaining dataset were filtered for high turbulence data. All data was filtered out as low quality and no further post-processing was done.

**Comment 3**

Equations 1-4 and their associated text can be moved to Supplemental Information, as they provide more introductory content. I also suggest moving the Gaussian Plume Model (equation 7) along with detailing the dispersion coefficients  $\sigma$  and other model parameters in the text to the SI.

**Authors response**

Thanks for your comment. All equations have been moved to Supplementary Information Section 2a.

**Comment 4**

While a detailed criticism of the assumptions embedded in the EC and other approaches is added in the discussion section by the authors, very little quantitative effort is made to confirm how several of these assumptions are being violated. For instance, when using the EC approach, no plots are provided, attempting to explain whether the local surface layer at the point of measurement is well mixed or not.

**Authors response**

Thank you for your comment. We have reanalyzed our data by accounting for these assumptions. For eddy covariance, EddyPro was used to account for factors such as time lags, stationarity and steady tests, and statistical tests that determined whether the data was usable for eddy covariance. Changes to manuscript are in comment 2 above. For the aerodynamic flux gradient, backward Lagrangian and the Gaussian plume inverse models, raw data are used by the assumption that by accounting for the distance, atmospheric stability, wind speed and height which are inputs to the model, we can estimate what the model quantifies in continuous measurements. The variability such as whether the plume is well mixed at the point of measurement is captured in the uncertainty. The point of the study is to check what uncertainties are associated with using continuous monitoring data as inputs for fence-line quantification as inputs to a model.

**Comment 5**

The decision to use an averaging time window of 30 minutes remains suspect as there are many METEC experiments with durations shorter than half an hour. Reynolds averaging is a critical piece behind the use of eddy covariance concepts and a choice of 30 mins based on discussion from a prior article on agricultural emissions isn't well justified.

**Authors response**

Thank you for your comment. We have reanalyzed our data to uniform 5 minutes, 10 minutes and 15-minute durations for all models where there were continuous durations. Even though there were short releases such as 10 s that represent intermittent emissions in oil and gas such as a venting of a gas pneumatic, our duration tables represent minute

durations when there were continuous releases for that period. For example, a 5-minute event table is for a continuous release in a 5-minute duration. To clarify this, the following changes have been made to the manuscript.

**Changes to Manuscript**

Section 2.4.1.1

Lines 211-213: The concentration-meteorological-release event tables were split into 5, 10 and 15-minute release event tables (i.e. there was a continuous release in the duration).

**Comment 6**

I recommend providing a reference dataset of the raw measurements along with a comprehensive validation exercise. This helps readers in the evaluation of the accuracy of the data reduction procedures and to appreciate the inverse modeling choices.

**Authors response**

Thank you for comment. A comprehensive dataset used to evaluate the models has been uploaded.

URL: http://datadryad.org/stash/share/7s42bajRb2czaY9hr6NbTKDAolCWZ\_pNrty0o54qXBE

**Comment 7**

I recommend a better flow of text and linkage between sections 4.1 and 4.2 (where authors refer to results) and section 3 to draw their conclusions.

**Authors response**

Thank you for your comment. The discussion section has been rewritten to reflect our results.

**Changes to Manuscript**

**4 Discussion**

Methane emissions quantification from oil and gas is a complex system comprising of gas emissions from different heights, different locations, encountering aerodynamic obstacles of different sizes, and of varying emissions duration, amongst others. The ability to precisely quantify emissions using data collected by a point sensor, downwind of a source is directly influenced by plume dynamics. The CH4 plume downwind of a source will change in size and shape in different atmospheric conditions, in open areas versus areas with obstacles, diurnally, and in different seasons (Casal, 2008). In this study, the precision to which downwind methods (closed-path EC, AFG, GPIM and bLs) could quantify the emission rate of point source(s) were tested in different atmospheric conditions (rain, sunny, snow, windy, calm etc.), and aerodynamic scenarios (emissions sources in open areas, behind obstacles, changing atmospheric stability, and day/night). As a result, testing the predicted emission rates to controlled release rates in different conditions introduced real-world scenarios that have not previously been tested, hence better understanding model uncertainty in the application of quantifying emissions from oil and gas production infrastructure.

**4.1 Eddy Covariance**

Eddy covariance was tested using a closed-path analyzer, cavity ring-down spectroscopy, with a 3.2 lpm pump flowrate and a 0.4 s gas flow response time. The closed-path EC estimated emissions between a factor of 0.67 and 0.97 for SRSP emissions, and between 1.02 and 2.43 for MRSP emissions at  $\pm 45^{\circ}$  wind sector range (Section 3.1). This was a wider uncertainty in estimated emissions than one reported by Dumortier et al. (2019), who estimated

emissions at between 90 and 113% of true emission (~1.5 kg day-1) with concentrations between 2 and 3 ppm. Our study tested closed-path EC at emission rates between 0.005 and 8.5 kg h-1.

Our study's results were when the data was filtered for frequencies greater than 8 Hz, hence, largely reducing the sampled emissions. The 10 Hz sampling frequency set for this instrument was not a true 10 Hz and this could have been due to the 0.4 gas flow response time that delayed analysis of the drawn air sample in the cavity, or the 3 lpm pump flow rate for a 3 m tubing that might have varied the effective sample turnover rate. The rest of the data were flagged as low quality by Mauder and Foken (2004) (0-1-2 system), which flags based on steady state and well-developed turbulence. This could have been due to low turbulence as experiments were carried out in winter, and instrumentation limitations (low pump flow rate and asynchronous configuration of the gas analyzer and meteorological instrument).

Continuous monitoring requires deployment of multiple sensors which create limitations of cost and requires instrumentation with a wide measurement range as concentrations for oil and gas emissions can range between 0 to 250 ppm, as in this study (Supplementary Information Section 1). The currently available EC instruments have a narrow measurement range (LI-COR LI-7700 open path CH4 analyzer has a measurement range of 0 to 25 ppm at -25 °C and 0 to 40 ppm at 25°C; PICARRO G2311-f, a closed-path analyzer has an operating range of 0 to 20 ppm). Also, the instrumentation should be environmentally robust and not lab-grade (be able to run smoothly in adverse weather conditions). Given these parameters, market available EC instruments that can currently be deployed in oil and gas are limited. The instrument used in this study is a field instrument (ABB MGGA GLA131 Series has a measurement range of 0 to 100 ppm for CH4 but can be extended to 0 to 1%).

**4.2 Aerodynamic Flux Gradient**

Overall, the AFG method quantified emissions within an MRF of 1.3 to 1.7 for SRSP emissions and between an MRF of 2.4 and 3.3 for MRSP emissions (Section 3.2). The uncertainties in AFG were higher than EC especially for MRSP emissions but lower than the GPIM and bLs methods. The differences between AFG and EC could have been due to different instrumentation and analytical approaches that limited the exact comparison of the methods i.e. EC data was filtered for frequencies less than 8 Hz, and AFG instrumentations required both analyzers to be running, periods when one analyzer was down, were not tested. To our knowledge, this is the first study to test AFG for point source quantification and results are promising.

Compared to the EC method that requires a very fast analyzer which may be difficult to deploy in oil and gas, the AFG requires at least 2 analyzers sampling at 1 Hz frequency, which is currently possible with the range of sensors available in the market. The main advantage of flux gradient methods (EC and AFG) tested in this study is that they do not require background control as background CH4 concentration is a highly variable parameter that cannot be controlled in open air especially when multiple emissions are happening. The AFG method relies on differences in CH4 concentrations between two heights and this study shows that in complex sites where there are multiple sources, the method quantifies better than the point-source GPIM and bLs methods. The main limitation of the AFG and EC methods for point-source quantification is that when the measurement point is downwind of more than one source in a wind sector range, more than one sonic anemometers are required to estimate the flux of each source for footprint calculation based on Dumortier et al. (2019)'s calculations.

**4.3 Gaussian Plume Inverse Method**

The GPIM method quantified emissions within an MRF of 2.4 and 2.6 for SRSP and between 15.7 and 25 for MRSP emissions (Section 3.3). The GPIM method is a point-source specific quantification approach and works best in open areas, free of obstacles, and when the background concentration is well defined. For multiple emissions, in aerodynamically complex environments, even though the sensor is downwind of a single source based on average wind direction, quantification is complexed by interference from other neighboring sources. The GPIM has previously been reported to quantify emissions within 40.7 and 60% error for a single point-source, (Riddick et al., 2022b). However, GPIM correct quantification has been suggested to be better for longer distances where the plume is well mixed. This is typically a challenge for fence-line sensors that have to be deployed within the facility boundaries where large downwind distances may not be practical.

**4.3 Backward Lagrangian Stochastic Model**

The bLs method quantified emissions within an MRF of 0.8 to 1.05 for SRSP emissions and between 3.85 and 11958.8 MRF for MRSP emissions (Section 3.4). Similarly to the GPIM method, the bLs method used in this study is a point-source specific quantification method that simulates transport of molecules in open area and where the background concentration is defined. In this case, as with the SRSP test scenario, the bLs approach quantified within 20% uncertainty. However, for MRSP emissions, the bLs largely overestimated emissions and this could have been due to the interference of neighboring sources, that even though the measurement point is downwind of a single source, actual plumes are not distinct and model-simulated plumes may not be representative. The point-source bLs approach in WindTrax is also not designed for more than one downwind source.

**Comment 8**

In dispersion modeling efforts, the modeler connects raw measurements with their understanding of the complex boundary layer physics and then uses this to judiciously fix the free parameters of the chosen model while acknowledging the underlying limitations of the constitutive equations all along. Additional explanations are appreciated to provide details on how authors tackled this challenge.

**Authors response**

Thank you for your comment. This study investigates if the four established downwind models can be applied in oil and gas continuous monitoring i.e., can the model quantify an emission every time there is a detection for a 5, 10 or 15-minute duration. As such, we have fixed the parameters that are inputs to the model based on measurement conditions and our site in our revised manuscript. For example, for the Gaussian plume inverse model, we have reanalyzed our data such that the dispersion parameters are calculated based on sonic anemometer data, as opposed to literature values. This way, we have used site and atmospheric specific data for our calculations ensuring we have correctly used the model.

**Comment 9**

Line 390: "Oil and gas point sources violate assumptions (1), (2) and (4)". Assumption 2 requires the CH4 fluxes to be turbulent. If they are not turbulent at the point of measurement, are they laminar? I'd recommend that the authors confirm this by plotting the power spectrum of the CH4 concentration at the point of measurement and investigating the absence of the characteristic turbulence cascade in response. Our experience with continuous monitoring methane

concentration data from METEC experiments referenced in the study unequivocally suggests that ambient concentration levels at measuring stations could be turbulent in nature.

**Authors response**

Thank you for your comment. After reprocessing our data using the EddyPro software, all data were flagged by quality check and friction velocity thresholding hence no further post-processing such as cospectra investigation was done. The Eddy covariance discussion was rewritten as below.

**Changes to Manuscript**

**4.1 Eddy Covariance**

Eddy covariance was tested using a closed-path analyzer, cavity ring-down spectroscopy, with a 3.2 lpm pump flowrate and a 0.4 s gas flow response time. The closed-path EC estimated emissions between a factor of 0.67 and 0.97 for SRSP emissions, and between 1.02 and 2.43 for MRSP emissions at ±45° wind sector range (Section 3.1). This was a wider uncertainty in estimated emissions than one reported by Dumortier et al. (2019), who estimated emissions at between 90 and 113% of true emission (~1.5 kg day-1) with concentrations between 2 and 3 ppm. Our study tested closed-path EC at emission rates between 0.005 and 8.5 kg h-1.

Our study's results were when the data was filtered for frequencies greater than 8 Hz, hence, largely reducing the sampled emissions. The 10 Hz sampling frequency set for this instrument was not a true 10 Hz and this could have been due to the 0.4 gas flow response time that delayed analysis of the drawn air sample in the cavity, or the 3 lpm pump flow rate for a 3 m tubing that might have varied the effective sample turnover rate. The rest of the data were flagged as low quality by Mauder and Foken (2004) (0-1-2 system), which flags based on steady state and well-developed turbulence. This could have been due to low turbulence as experiments were carried out in winter, and instrumentation limitations (low pump flow rate and asynchronous configuration of the gas analyzer and meteorological instrument).

Continuous monitoring requires deployment of multiple sensors which create limitations of cost and requires instrumentation with a wide measurement range as concentrations for oil and gas emissions can range between 0 to 250 ppm, as in this study (Supplementary Information Section 1). The currently available EC instruments have a narrow measurement range (LI-COR LI-7700 open path CH4 analyzer has a measurement range of 0 to 25 ppm at -25 °C and 0 to 40 ppm at 25°C; PICARRO G2311-f, a closed-path analyzer has an operating range of 0 to 20 ppm). Also, the instrumentation should be environmentally robust and not lab-grade (be able to run smoothly in adverse weather conditions). Given these parameters, market available EC instruments that can currently be deployed in oil and gas are limited. The instrument used in this study is a field instrument (ABB MGGA GLA131 Series has a measurement range of 0 to 100 ppm for CH4 but can be extended to 0 to 1%).

**Comment 10**

Assumption 4 states that measurements should be inside the boundary layer and in the constant stress layer. If the choice of RL from section 2.4.4 is to be taken at face value, then the constant stress layer extends up to 1m upward thus implying that measurements satisfy assumption 4.

**Authors response**

This statement has been withdrawn from our discussion as in the response to Comment 7 above. Also, we recalculated the roughness length from our sonic anemometer measurements in neutral conditions.

**Changes to Manuscript**

**2.3 Calculation of Roughness Length**

Surface roughness length (z0) was calculated from friction velocity (Supplementary Information Section 2a: Equations 1 and 2) by splitting the high frequency sonic anemometer data into 15-minute tables and filtering for those in neutral conditions, |L| > 500 (Supplementary Information Section 2a: Equation 3). The overall roughness length selected as the median of all the calculated z0 was 0.1 m (Rey-Sanchez et al., 2022).

**Comment 11**

Line 395-398: "Even though current eddy covariance application assumes the vertical flux at a point is independent of atmospheric stability (Denmead, 2008), atmospheric stability has an impact on point source gas ...". While Denmaead (2008) notes that EC "is independent of atmospheric stability", it merely implies that EC by virtue of being a more fundamental approach relies on direct measurement of turbulent fluxes like  $\langle u'w' \rangle$  and  $\langle u'\theta' \rangle$ , i.e. the information on atmospheric stability is implicitly built into the EC approaches. Apriori estimates of the Obhukov length scale that divide the surface layer into multiple regimes as required by the AFGM and Gaussian models are therefore not needed. The EC does not imply that the vertical flux is independent of atmospheric stability.

**Authors response**

Thank you for your comment. After reprocessing our data using the EddyPro software, all data were flagged by quality check and friction velocity thresholding hence no further post-processing. This statement has been withdrawn from our discussion as in the response to Comment 7 above.

**Comment 12**

Line 384-385 & 411: "Positive fluxes represent emissions and downward fluxes represent absorptions". This is not an assumption, but rather a statement of fact and an unnecessary one at that.

**Authors response**

This statement has been withdrawn from our discussion as in the response to Comment 7 above.

**Comment 13**

Line 391: "Emissions are collimated plumes instead of turbulent fluxes". It requires clarification as to what the authors mean when mentioning the phrase 'turbulent fluxes'. Collimated plumes are simply the smoothing effect of time averaging on an underlying turbulent structure. Please see the following references

- https://www.cambridge.org/highereducation/books/turbulentflows/C58EFF59AF9B81AE6CFAC9ED16486B3A#overview
- o https://epubs.siam.org/doi/pdf/10.1137/10080991X

**Authors response**

This statement has been withdrawn from our discussion as in the response to Comment 7 above.

**Comment 14**

Line 72: I wonder how are major shortcomings identified using fence-line approaches by Ilonze et al. (2024)?

**Authors response**

Ilonze et al. (2024) study compared continuous monitoring solutions and their reported quantification estimates based on what solutions reported. Details on specific factors that caused some solutions to overestimate are not provided. The study provides the overall performance of fence-line continuous monitoring sensors.

**Comment 15**

Line 219: "The roughness sublayer is set to 1" is not well justified. I suppose it means 1m? A roughness sublayer (RL) is a region immediately above the surface and below the inertial layer where horizontal in-homogeneity in flow variables persists. In other words, within the RL variables like <w'c'> (where c is the local methane concentration, w is the vertical wind speed, and 'represents the fluctuating field) can be assumed to be horizontally homogeneous. While there are well-established results in the literature on how to estimate the RL height, having access to direct point measurements at multiple heights offers a direct route to estimate the RL height. Please see the following reference.

o https://journals.aps.org/prfluids/abstract/10.1103/PhysRevFluids.3.114603

**Authors response**

We recalculated the roughness length from our sonic anemometer measurements in neutral conditions.

**Changes to Manuscript**

**2.3 Calculation of Roughness Length**

Surface roughness length (z0) was calculated from friction velocity (Supplementary Information Section 2a: Equations 1 and 2) by splitting the high frequency sonic anemometer data into 15-minute tables and filtering for those in neutral conditions, |L| > 500 (Supplementary Information Section 2a: Equation 3). The overall roughness length selected as the median of all the calculated z0 was 0.1 m (Rey-Sanchez et al., 2022).

**Comment 16 and 17**

Lines 430-431: "The main assumption of the Gaussian plume model is that CH4 emitted from a point source enters the airflow, disperses vertically and laterally, forming a conical plume (Riddick et al., 2022b; US EPA, 2013)". This is not an assumption but instead the standard response of the tracer being advected horizontally and vertically by the air.

Line 431-433: "The formation of a conical plume is hindered at oil and gas facilities by obstacles (equipment) and is affected by atmospheric stability." While strictly speaking sizable obstacles may hinder the plume, whether this is indeed the case for the METEC site is unclear. The manuscript does not offer proper evidence to support this argument. Given the sparsity of the few obstacles on the METEC site and the focus of the current study on distant, fenceline monitoring, it seems to reason that obstacles in fact do not hinder plume features.

**Authors response**

This statement has been withdrawn from our Gaussian plume discussion has been corrected as below.

**Changes to manuscript**

**4.3 Gaussian Plume Inverse Method**

The GPIM method quantified emissions within an MRF of 2.4 and 2.6 for SRSP and between 15.7 and 25 for MRSP emissions (Section 3.3). The GPIM method is a point-source specific quantification approach and works best in open areas, free of obstacles, and when the background concentration is well defined. For multiple emissions, in aerodynamically complex environments, even though the sensor is downwind of a single source based on average wind direction, quantification is complexed by interference from other neighboring sources. The GPIM has previously been reported to quantify emissions within 40.7 and 60% error for a single point-source, (Riddick et al., 2022b). However, GPIM correct quantification has been suggested to be better for longer distances where the plume is well mixed. This is typically a challenge for fence-line sensors that have to be deployed within the facility boundaries where large downwind distances may not be practical.

**Comment 18**

Line 463-465: "Even though these modeling approaches have been reported to work elsewhere (e.g., agricultural and landfill emissions), it does not necessarily mean it could work in the intended area of application." This blanket statement is unwarranted. Compared to many complex scenarios observed in agricultural and landfill emission cases, the current controlled-release tests present a much-simplified dataset both in terms of emission characteristics (discrete constant emissions rates from only 5 equipment groups all starting and stopping simultaneously) and site features (flat terrain with few sparsely spaced obstacles).

**Authors response**

This statement has been withdrawn and the implications section has been corrected as below.

**Changes to manuscript**

**Lines 522-535**

Oil and gas point sources could either be single emissions or multiple emissions occurring concurrently. In cases of multiple emissions with more than one release point being upwind, the Gaussian model and the backward Lagrangian stochastic models are limited, as they can only quantify one source at a time; and interference from neighboring emissions affects the underlying principles of dispersion on which these models were developed. As a result, flux quantification models used in other applications such as eddy covariance and aerodynamic flux gradient have been proposed as the solution. This study's results show that generally reasonable quantification estimates are achieved with flux approaches (eddy covariance and aerodynamic flux gradient), but these methods require more instrumentation effort (fast sampling analyzer for eddy covariance, and multiple collocated sensors for aerodynamic flux gradient). Even though the widely applied Gaussian plume inverse method and the backward Lagrangian stochastic models are widely used for single-point emissions, this study shows aerodynamic complexities, the difficulty in defining the background, and interference from neighboring sources challenge the application of these models for fence-line continuous monitoring. This study recommends more testing of flux quantification models for

oil and gas quantification as they could improve emissions quantification for leak repair prioritization and methane reporting.

**Anonymous Referee #3 (RC3)**

This paper makes a valuable contribution towards quantifying emissions from point sources. The problem is important, and the work is thoughtful. Mbua et al. use controlled release experiments to test three widely used methodologies and find all three methods are inaccurate and two methods seriously underestimate emissions. In Mbua et al's controlled release experiments, using fence line measurements, both eddy covariance and flux gradient methods were found to underestimate emissions. Gaussian plume methods performed better but with very wide scatter. I note the points already raised by RC1 and RC2, particularly RC1's point 4 on closed path measurement leading to underestimation. More generally, this paper is tackling a major problem. Accurate quantification of emissions (especially methane) from point sources such as natural gas production facilities is a tough nut to crack, more so if the "point: source is somewhat disseminated (e.g. a gas production facility with pipes, pumps and valves scattered over areas up to a hectare, or similarly, a farm complex over a similar area with manure lagoons, biodigesters and animals. Very large sources (eg >100 kg/hr methane) can be studied from space, and small sources (e.g. single cows) can be isolated, but quantifying complex aggregated sources in the 1-50 kg/hr range (like gas production facilities) is not easy. Use of methods similar to Mbua et al's systems is widespread, and thus these experiments are potentially important in devising better protocols and better regulation and mitigation of methane emissions from fossil fuel installations (gas production facilities, venting oil wells, coal mine vents, etc). The work also has wide applicability for tracking biogenic emissions from landfill and waste, around biodigesters, and on farms with large animal populations. The scientific methodology and experimental procedures are clearly explained and the work is well presented. Overall the paper is well written. Controlled release experiments like this are likely to be influential in designing better quantification methods and in devising regulation protocols. Thus, the paper should be accepted. That said, I note the comments made by other referees and have some additional minor comments.

**Authors response**

Thank you for reviewing our paper. We have accordingly addressed the concerns in the specific comments below.

**Comment 1**

Abstract. Lines 1-11 are really introduction and might be omitted here, and covered in Section 1 instead. Line 20-21 'input to.....method' – minor English problem? Maybe methods?

**Authors response**

Thank you for your comment. The first sentences aim at providing context for our study, and why we have to evaluate quantification accuracy for methane reporting.

**Changes to Manuscript**

**Abstract.** The dependable reporting of methane (CH4) emissions from point sources, such as fugitive leaks from oil and gas infrastructure, is important for profit maximization (retaining more hydrocarbons), evaluating climate impacts, assessing CH4 fees for regulatory programs, and validating CH4 intensity in differentiated gas programs. Currently,

there are disagreements between emissions reported by different quantification techniques for the same sources. It has been suggested that downwind CH4 quantification methods using CH4 measurements on the fence-line of production facilities could be used to generate emission estimates from oil and gas operations at the site level, but it is currently unclear how accurate the quantified emissions are. To investigate downwind methods' accuracy, this study uses fenceline simulated data collected during controlled release experiments as input for closed-path eddy covariance, aerodynamic flux gradient, the Gaussian plume inverse method, and the backward Lagrangian stochastic model in a range of atmospheric conditions. Generally, results show that flux quantification methods provide more reasonable estimates compared to point-source specific models especially when multiple releases are happening at the facility level. The closed-path eddy covariance quantified emissions with a mean relative factor (estimated emission over actual emission) of 0.7 to 1 for single-release single-point emissions, and within a mean relative of 1 and 2.4 for multirelease single-point emissions. The aerodynamic flux gradient method quantified emissions within a mean relative factor of 1.3 to 1.7 for single-release single-point emissions, and between 2.4 and 3.3 for multi-release single-point emissions. The Gaussian plume inverse model quantified emissions within a mean relative factor of between 2.4 and 2.6 for single-release single-point emissions, but largely overestimated emissions when multiple releases were happening; mean relative factor between 16 and 25. Similarly to the Gaussian plume inverse method, the backward Lagrangian stochastic model for point sources using WindTrax quantified within a mean relative factor of between 0.8 to 1 for single-release single-point emissions, but largely overestimated emissions for multi-release single-point emissions; mean relative factor of 3.9 and 11958. As continuous monitoring of oil and gas sites can involve complex emissions where plumes are not defined due to multiple sources, this study shows that common downwind point source dispersion models could largely overestimate emissions. This study recommends more testing of flux quantification models for oil and gas continuous monitoring quantification.

**Comment 2**

Lines 1-32 could be rewritten somewhat, maybe to mention the Global Methane Pledge and to introduce the tension between bottom-up and top-down methodologies.

**Authors response**

Thank you for your comment. We have rewritten the abstract section based on our results as in Comment 2 above. The discrepancy between bottom up and top down technologies is included in the introduction.

**Comment 3**

Line 50 – maybe give some details about very rough emission ranges (e.g. satellite detection of point sources >100 kg/h, low flying aircraft (say 10-100 kg, SUV-mounted instruments (say 1-20 kg/hr). Also maybe mention the scale of 'disseminated' 'point' sources – e.g. a complex gas facility on a 100mx100m pad, or a farm barn/lagoon complex of the same size.

**Authors response**

Thank you for your comment. Detection limits have been added to the manuscript.

**Changes to Manuscript**

Lines 52-60

Top-down methods, including using aircraft such as Bridger Photonics LiDAR (Light Detection and Ranging; 90% detection limit of ~ 2 kg h-1) (Johnson et al., 2021) and satellites such as Carbon Mapper (predicted 90% detection limit of about 100 kg h-1) ("Carbon Mapper - Science & Technology," n.d.) can also be used to infer emissions. However, these survey methods only quantify emissions over a very short period of time (< 10 s) and observations are typically made during the day which can often coincide with maintenance activities that can bias emissions and result in overestimation (Riddick et al., 2024a; Zimmerle et al., 2024). Additionally, different top-down technologies measuring the same source have disagreed in their reported emissions which has called into question the credibility of these methods (Brown et al., 2023; Conrad et al., 2023). As a result, ensuring accuracy in models and technologies used in CH4 emissions quantification has been a complex issue.

**Comment 4**

Line 56 – 'NG acronym may not be understood in a global journal. Better spell it out.

**Authors response**

Thanks for your comment. NG has been spelled out.

**Changes to Manuscript**

**Comment 5**

Line 96 – Fig 1 is good. Maybe add a Gaussian plume panel also?

Thanks for your comment. The figures for the Gaussian plume inverse model and backward Lagrangian models have been added.

**Changes to Manuscript**

Figure 2. A: An illustration of a plume that follows a Gaussian plume inverse model where emission rate can be inferred from concentrations at different downwind distances and crosswind distances. B: An illustration of how the backward Lagrangian stochastic model traces particles to the source.

**Comment 6**

Line 115 – maybe say how big the METEC facility is and give more details for non-US readers.

**Authors response**

Thank you for your comment. A description of METEC has been added.

**Changes to Manuscript**

Section 2.1

**Lines 142-146**

Controlled release experiments were conducted at the Colorado State University's Methane Emissions Technology Evaluation Center (METEC) in Fort Collins, CO (USA, 65 miles north of Denver) between February 8, and March 20, 2024. The METEC center is a simulated oil and gas facility that does controlled testing for emissions leak detection and quantification technology development, field demonstration, leak detection protocol and best practices development (METEC, 2025).

**Comment 7**

Line 181 – masts' locations? Plural masts on many locations? Text as it is now has many mast on one location.

**Authors response**

Thank you for your comment. We had two masts 9.4 m apart, one for eddy covariance, backward Lagrangian stochastic model and the Gaussian plume measurements, and the other for the aerodynamic flux gradient quantification.

**Changes to Manuscript**

**Lines 163-173**

Methane concentration data for closed-path EC, GPIM and bLs methods were collected through an inlet tubing (3.275 mm inner diameter) at 3 m height, connected to the ABB (Zurich, Switzerland) GLA131 Series Microportable Greenhouse Gas Analyzer (MGGA) set to sample at 10 Hz. The MGGA is a closed-path greenhouse gas analyzer with a ~3.2 lpm pump flowrate, 10 cm cell length, 1 inch cell diameter (~0.23 standard cubic centimeters per minute (sccm) effective volume), and 0.4 s gas flow response time. The inlet tubing was collocated with an R. M. Young (Traverse City, MI, USA) 81000 sonic anemometer (R.M. Young Company, 2023) which measured micrometeorology at 10 Hz (Figure 3-1). The northward, eastward and vertical separation of the inlet tubing from the sonic anemometer was 0, 0, -10 cm, respectively. For AFG, CH4 concentration data was collected at 2 and 4 m using two Aeris (Hayward, CA, USA) MIRA Ultra Series analyzers connected to tubing with a 3.275 mm inner diameter (Figure 3-2). As we had only one sonic anemometer, data from the sonic anemometer collocated with the MGGA were used for the AFG quantification. The two sampling points are 9.4 m apart.

**Comment 8**

Line 194 – background. 5th percentile - A steady time-invariant small leak could thus be included in 'background' Are there any measurements of upwind background?

**Authors response**

Thanks for your comment. We did not have an upwind mast. Background concentrations were recalculated to account for time variation and residual methane from previous releases.

**Changes to Manuscript**

**Lines 269-277**

For continuous monitoring sensors, background concentration can be determined from CH4 concentrations measured by a sensor upwind of the emission source, or by sampling when the wind is blowing away from the source. However, for continuous monitoring sensors, using an upwind sensor has the limitation of missing downwind background noise resulting from emissions in the preceding emission event where there is residual CH4 in air especially during stable conditions, and capturing sensors drift in the downwind sensor. In this study, background CH4 was calculated as the average of the lowest 5th percentile, 5 minutes before each release started. In cases where this background was greater than the mean CH4 concentration in the quantifying duration, the minimum CH4 concentration for that duration was used as the background. Methane enhancement was then calculated as CH4 concentration minus the background.

**Comment 9**

Line 215 – footprint 80%-90% contour contrast. This is interesting: space for a slightly longer comment? Authors response

After evaluating the approach for footprint calculation for point sources, we used the 90% footprint area. We identified if the point source was located in the 90% footprint, and using the footprint value, we calculated the emission.

**Changes to Manuscript**

**Lines 254-265**

Eddy covariance and AFG footprints were calculated using the Kljun et al. (2015) footprint model. Even though Rey-Sanchez et al. (2022) reported the Kljun et al. (2015) footprint model to be less accurate compared to the Kormann and Meixner (2001), Kormann and Meixner (2001) was too complex for our study because it required multiple sonic anemometers or tracer release experiments to calculate the exponential wind velocity power law, and the exponential eddy diffusivity power law for site specific data. Our study was limited to a single sonic anemometer, and this provided enough inputs for the Kljun et al. (2015) footprint model. The default pixel size 2 \* 2 m was used in this study. This study first calculated the area that contributed 90% of the vertical flux; and based on the location of the point source, the source was determined if it was within the 90% footprint area. Point source emissions of sources within this region were then calculated based on the approach by Dumortier et al. (2019). This approach assumes all measured flux is equal to flux resulting from a single point source. In case of the mast being downwind of more than one source, more sonic anemometers are needed to solve the two unknown point source fluxes.

**Comment 10**

Line 235 – Line 346 'largely underestimated' – see also Fig 4. I note the comment posted earlier by Brian Lamb. Maybe that question could be picked up further? Mbua et al's finding is surprising and immediately makes the reader worry about eddy covariance quantification in wetlands also!

**Authors response**

Thanks for your comment. Point source emission was recalculated based on the approach by Dumortier et al (2019) where emission per source = measured flux / value of the footprint at the cell. The source location was first determined if it was within the 90% footprint region, and the footprint value extracted.

**Changes to Manuscript:**

Section 2.4.3

**Lines 261-265**

Point source emissions of sources within this region were then calculated based on the approach by Dumortier et al. (2019). This approach assumes all measured flux is equal to flux resulting from a single point source. In case of the mast being downwind of more than one source, more sonic anemometers are needed to solve the two unknown point source fluxes.

**Comment 11**

Lines 388- 401. This is important and perhaps needs also later to be put in the wetland context where atmospheric stability can also be a problem.

**Authors response**

Thanks for your comment. This statement was withdrawn and eddy covariance discussed as follows.

**Changes to Manuscript**

**Section 4.1**

Eddy covariance was tested using a closed-path analyzer, cavity ring-down spectroscopy, with a 3.2 lpm pump flowrate and a 0.4 s gas flow response time. The closed-path EC estimated emissions between a factor of 0.67 and 0.97 for SRSP emissions, and between 1.02 and 2.43 for MRSP emissions at ±45° wind sector range (Section 3.1). This was a wider uncertainty in estimated emissions than one reported by Dumortier et al. (2019), who estimated emissions at between 90 and 113% of true emission (~1.5 kg day-1) with concentrations between 2 and 3 ppm. Our study tested closed-path EC at emission rates between 0.005 and 8.5 kg h-1.

Our study's results were when the data was filtered for frequencies greater than 8 Hz, hence, largely reducing the sampled emissions. The 10 Hz sampling frequency set for this instrument was not a true 10 Hz and this could have been due to the 0.4 gas flow response time that delayed analysis of the drawn air sample in the cavity, or the 3 lpm pump flow rate for a 3 m tubing that might have varied the effective sample turnover rate. The rest of the data were flagged as low quality by Mauder and Foken (2004) (0-1-2 system), which flags based on steady state and well-developed turbulence. This could have been due to low turbulence as experiments were carried out in winter, and instrumentation limitations (low pump flow rate and asynchronous configuration of the gas analyzer and meteorological instrument).

Continuous monitoring requires deployment of multiple sensors which create limitations of cost and requires instrumentation with a wide measurement range as concentrations for oil and gas emissions can range between 0 to 250 ppm, as in this study (Supplementary Information Section 1). The currently available EC instruments have a narrow measurement range (LI-COR LI-7700 open path CH4 analyzer has a measurement range of 0 to 25 ppm at -25 °C and 0 to 40 ppm at 25°C; PICARRO G2311-f, a closed-path analyzer has an operating range of 0 to 20 ppm). Also,

the instrumentation should be environmentally robust and not lab-grade (be able to run smoothly in adverse weather conditions). Given these parameters, market available EC instruments that can currently be deployed in oil and gas are limited. The instrument used in this study is a field instrument (ABB MGGA GLA131 Series has a measurement range of 0 to 100 ppm for CH4 but can be extended to 0 to 1%).

**Comment 12**

Line 434 – daytime insolation - in facilities working all night as well as all day, nocturnal factors like low inversion height and fogs may also be a factor, especially if the facility is installing automated systems.

**Authors response**

Thanks for your comment. We recalculated the Gaussian plume inverse model based on sonic measurements as opposed to Pasquil Gifford classification system hence, the daytime insolation is no longer considered. This means that dispersion parameters can be generated at all times.

**Changes to Manuscript**

Supplementary Information Sectio 2a

 $\sigma y$  (m) = horizontal dispersion coefficient calculated as  $\sigma v(x/U)$ :  $\sigma v$  is the standard deviation of v wind speed from the sonic anemometer, x is the downwind distance from the point source and U is the mean wind speed

 $\sigma z$  (m) = vertical dispersion coefficient calculated as  $\sigma w(x/U)$ :  $\sigma w$  is the standard deviation of w wind speed from the sonic anemometer, x is the downwind distance from the point source and U is the mean wind speed

**Comment 13**

Line 465 – maybe a paragraph here in 'future fixes' – how to make EC work better perhaps, and especially how to make Gaussian methods better. For example, by using drones it may be possible to map plumes much more accurately. Also perhaps a digression somewhere to mention wetlands (e.g. L386) in more detail.

**Authors response**

Based on our results, the discussion and conclusion was changed as below.

**Changes to Manuscript**

**4.1 Eddy Covariance**

Eddy covariance was tested using a closed-path analyzer, cavity ring-down spectroscopy, with a 3.2 lpm pump flowrate and a 0.4 s gas flow response time. The closed-path EC estimated emissions between a factor of 0.67 and 0.97 for SRSP emissions, and between 1.02 and 2.43 for MRSP emissions at ±45° wind sector range (Section 3.1). This was a wider uncertainty in estimated emissions than one reported by Dumortier et al. (2019), who estimated emissions at between 90 and 113% of true emission (~1.5 kg day-1) with concentrations between 2 and 3 ppm. Our study tested closed-path EC at emission rates between 0.005 and 8.5 kg h-1.

Our study's results were when the data was filtered for frequencies greater than 8 Hz, hence, largely reducing the sampled emissions. The 10 Hz sampling frequency set for this instrument was not a true 10 Hz and this could have been due to the 0.4 gas flow response time that delayed analysis of the drawn air sample in the cavity, or the 3 lpm pump flow rate for a 3 m tubing that might have varied the effective sample turnover rate. The rest of the data were flagged as low quality by Mauder and Foken (2004) (0-1-2 system), which flags based on steady state and welldeveloped turbulence. This could have been due to low turbulence as experiments were carried out in winter, and instrumentation limitations (low pump flow rate and asynchronous configuration of the gas analyzer and meteorological instrument).

Continuous monitoring requires deployment of multiple sensors which create limitations of cost and requires instrumentation with a wide measurement range as concentrations for oil and gas emissions can range between 0 to 250 ppm, as in this study (Supplementary Information Section 1). The currently available EC instruments have a narrow measurement range (LI-COR LI-7700 open path CH4 analyzer has a measurement range of 0 to 25 ppm at -25 °C and 0 to 40 ppm at 25°C; PICARRO G2311-f, a closed-path analyzer has an operating range of 0 to 20 ppm). Also, the instrumentation should be environmentally robust and not lab-grade (be able to run smoothly in adverse weather conditions). Given these parameters, market available EC instruments that can currently be deployed in oil and gas are limited. The instrument used in this study is a field instrument (ABB MGGA GLA131 Series has a measurement range of 0 to 100 ppm for CH4 but can be extended to 0 to 1%).

**4.2** Aerodynamic Flux Gradient**

Overall, the AFG method quantified emissions within an MRF of 1.3 to 1.7 for SRSP emissions and between an MRF of 2.4 and 3.3 for MRSP emissions (Section 3.2). The uncertainties in AFG were higher than EC especially for MRSP emissions but lower than the GPIM and bLs methods. The differences between AFG and EC could have been due to different instrumentation and analytical approaches that limited the exact comparison of the methods i.e. EC data was filtered for frequencies less than 8 Hz, and AFG instrumentations required both analyzers to be running, periods when one analyzer was down, were not tested. To our knowledge, this is the first study to test AFG for point source quantification and results are promising.

Compared to the EC method that requires a very fast analyzer which may be difficult to deploy in oil and gas, the AFG requires at least 2 analyzers sampling at 1 Hz frequency, which is currently possible with the range of sensors available in the market. The main advantage of flux gradient methods (EC and AFG) tested in this study is that they do not require background control as background CH4 concentration is a highly variable parameter that cannot be controlled in open air especially when multiple emissions are happening. The AFG method relies on differences in CH4 concentrations between two heights and this study shows that in complex sites where there are multiple sources, the method quantifies better than the point-source GPIM and bLs methods. The main limitation of the AFG and EC methods for point-source quantification is that when the measurement point is downwind of more than one source in a wind sector range, more than one sonic anemometers are required to estimate the flux of each source for footprint calculation based on Dumortier et al. (2019)'s calculations.

**4.3 Gaussian Plume Inverse Method**

The GPIM method quantified emissions within an MRF of 2.4 and 2.6 for SRSP and between 15.7 and 25 for MRSP emissions (Section 3.3). The GPIM method is a point-source specific quantification approach and works best in open areas, free of obstacles, and when the background concentration is well defined. For multiple emissions, in aerodynamically complex environments, even though the sensor is downwind of a single source based on average wind direction, quantification is complexed by interference from other neighboring sources. The GPIM has previously been reported to quantify emissions within 40.7 and 60% error for a single point-source, (Riddick et al., 2022b).

However, GPIM correct quantification has been suggested to be better for longer distances where the plume is well mixed. This is typically a challenge for fence-line sensors that have to be deployed within the facility boundaries where large downwind distances may not be practical.

**4.3 Backward Lagrangian Stochastic Model**

The bLs method quantified emissions within an MRF of 0.8 to 1.05 for SRSP emissions and between 3.85 and 11958.8 MRF for MRSP emissions (Section 3.4). Similarly to the GPIM method, the bLs method used in this study is a point-source specific quantification method that simulates transport of molecules in open area and where the background concentration is defined. In this case, as with the SRSP test scenario, the bLs approach quantified within 20% uncertainty. However, for MRSP emissions, the bLs largely overestimated emissions and this could have been due to the interference of neighboring sources, that even though the measurement point is downwind of a single source, actual plumes are not distinct and model-simulated plumes may not be representative. The point-source bLs approach in WindTrax is also not designed for more than one downwind source.

---

## Referee Report (RR1)

Review of revision 2 egusphere-2024-3161 "Evaluating the accuracy of downwind methods for quantifying point source emissions", Mbua et al. (Reviewer Comments by E. Thoma and S. Ludwig)

The authors were responsive to reviewer comments, offering a significantly revised manuscript with improved results visibility and supporting detail. The authors added critical QA/QC information on eddy covariance (EC) and decided to remove the aerodynamic flux gradient analysis. The authors modified the Gaussian plume inverse method (GPIM) approach, and backwards Lagrangian stochastic (bLs) analysis (based on WindTrax). Additional details were added for these emission calculation approaches.

There remains one major set of concerns with the revised manuscript relating to the EC analysis. As a first point, the EC results changed significantly from manuscript version 2. The reason for this is not clear and the authors are encouraged to double check the analysis and identify the root cause for this difference.

Assuming the current version of the EC analysis is final, the major concern centers on the strength of conclusions on the performance of the EC approach that can be drawn from this study. The authors have progressed their EC footprint and QA/QC analysis significantly from the original manuscript. However, new supporting information on ogive and cospectra departs significantly from expected form. These results indicate that EC analysis is likely not possible with these data and in fact illustrate "textbook examples" of issues illuminated by these QA/QC checks. This general issue with the EC analysis is further evidenced by the presence of large negative fluxes (which indicate issues in the EC data collection).

When examining both ogives and cospectra as a part of the QA/QC process, there is both a qualitative shape expected and quantitative metrics of slopes (for portions of the cospectra) and sigmoidal parameters (for the ogives) when good EC data are collected. Some deviations from the ideal form are expected. For example, especially in closed-path eddy covariance, there is often tube attenuation or increased lag that results in poorer performance with data at the highest frequencies. This is seen as a slightly steeper slope than ideal in the cospectra shape at high frequencies and is compensated for during the transfer function calculations when processing data into fluxes. However, even in closed-path EC or EC sampled at 5 Hz rather than 10 Hz, the cospectra still closely follow the ideal shape (especially when examining the cospectra of sonic temperature, which should not suffer any of the issues of the closed path gas sampling system), and slope changes at high frequencies are well modeled by the transfer functions. When the shape of cospectra deviate significantly from the ideal curve (as is the case here), it is an indication the data were not collected properly in a way that can be used for eddy covariance, with causes that include obstructions, mis-aligned time series, too slow system response time, among other issues with the instrumentation as seen here.

Similarly, the ogive analysis should follow a characteristic shape, a sigmoid curve plateauing at the y-axis at 1 and also at 0. The ogive analysis is used to indicate if an appropriate averaging interval was used, as those that are too short will not sufficiently plateau at 1. Furthermore, those ogives which do not follow a sigmoid shape at all indicate issues in data collection. Even accounting for the log-scale y-axis in the authors' ogive figures, they do not follow an acceptable shape, and all ogives here would indicate issues leading to removing the data during QA/QC. I am including examples of the appropriate expected shapes of cospectra and ogives as described in the textbook "Eddy Covariance Method" by George Burba, section 5.1 "Quality Control of Eddy Covariance Flux Data". This chapter provides several examples of how the shapes of cospectra can be used to diagnose issues with the instrumentation and data collection (such as is the case here) that invalidate the EC method.

From the book "Eddy Covariance Method" by George Burba, section 5.1 Quality Control of EC data; Cospectra Analysis. This figure depicts both the ideal cospectra and expected slope deviations at high frequencies for certain gases and systems.

From the book "Eddy Covariance Method" by George Burba, section 5.1 Quality Control of EC data; Ogive Optimization. This figure depicts the expected ogive shape for observations at a site with an optimized averaging interval of 60-minutes.

The authors now acknowledge the limitations in design of the EC data acquisition system for this study and attempt caveat in numerous places. They also point to non-stationarity in the data as part of the issue with the EC measurement.

However, if the EC results are deemed invalid, then these caveats are insufficient and conclusions around EC performance are not supported. The authors should either remove the EC analysis or suitably modify description to further clarify the severity of the issues for the reader. With little further work, the authors may choose to take this opportunity to illustrate some basic aspects of QA/QC assessment of EC data for this application. The information would be beneficial to the oil and gas/leak detection community (that largely consists of non-EC experts) and would assist future efforts to assess EC for this application.

Here is one example of unsupported conclusions from the abstract.

Ln 17 "Generally, the closed-path EC system used in this study proved generally unreliable and largely underestimated emissions, primarily due to non-stationarity and study limitations associated with using a non-standard setup. In comparison, the Gaussian Plume Inverse Method (GPIM) consistently outperformed the EC system for both single-release and multi-release single-point emissions."

This is an inappropriate indictment of the EC methodology. If your primary QA/QC data indicate that the attempt at the application of the EC method was not successful (for whatever reason), then it is not possible to draw this conclusion. If the EC analysis is to remain in this manuscript, the description needs to be recast as an attempt at EC that failed. This would render the presented comparisons to other methods invalid. The issues with the method application were detected and reasons for these issues are presented here as lessons learned. Future attempts at exploring EC for this application will benefit from the information in this paper.

---

## Author Response (AR2)

Ms. Ref. No.: EGUSPHERE-2024-3161

Title: Evaluating the feasibility of using downwind methods to quantify point source oil and gas emissions using

continuous monitoring fence-line sensors

The Powerhouse Energy Campus

Colorado State University

430 North College Avenue

Fort Collins, CO 80524

E-mail: Mercy.Mbua@colostate.edu

June 10th, 2025

Dear Professor Presto,

We appreciate the time and effort you, the reviewers and community dedicated to providing feedback on our

manuscript. We have closely followed the suggestions and have revised the manuscript accordingly. We believe the

revisions have improved our manuscript and that it is ready for publication. We have listed the original comments and

our response in blue below for our reply to comments. All figure numbers, tables, and lines refer to the updated

manuscript. Data for reviewers' is available Reviewer at

URL: http://datadryad.org/stash/share/7s42bajRb2czaY9hr6NbTKDAolCWZ\_pNrty0o54qXBE.

We look forward to hearing from you.

Please find our detailed responses below.

Yours sincerely,

Mercy Mbua (corresponding author)

and co-authors: Stuart N. Riddick, Elijah Kiplimo, Kira B. Shonkwiler, Anna Hodshire and Daniel Zimmerle

**Reviewer comments**

Eben Thoma (RC1)

**General comments**

The authors added additional detail on eddy covariance (EC) and modified the analysis approach for EC, aerodynamic

flux (AF), and the Gaussian plume inverse method (GPIM). A backwards Lagrangian stochastic (bLs) analysis based

on WindTrax was added. The authors also provided links to supporting information on the specifics of the controlled

release trials and other data. The revised manuscript carries new discussion and conclusions. There are two major

concerns with the revised manuscript:

(1) Although an improved EC data analysis approach is presented, there remains concerns with the configuration and

some aspects of method quality assurance (detailed below).

(2) The GPIM analysis changed from the original manuscript now showing large overestimates, especially for the

multi-release single-point scenarios (MSRP). A similar result is found for bLs. The GPIM equation (SI Equation 7)

differs from the original form, and it is unclear if ground reflection is considered (see identical terms in denominator).

Please check for possible calculation error that could drive bias.

Author's response:

Thank you for your comment. The GPIM equation had a typo, and we have corrected it. We have redone our analysis

using the original submission equation that accounts for the height of the boundary, and using the standard Gaussian

equation (Supplementary Information Equations 7 & 8). Based on your concern about using the bootstrapped mean

as a metric for evaluating the models in the comment below, we have now presented our data using a linear regression

and this has ensured correct models evaluation in alignment with our original submission.

For the Gaussian Plume Inverse model, we tested the model in six different scenarios to evaluate both the equations

and generation of dispersion coefficients using single-release single-point emissions to test parameters where the

model works best, and this was used for multi-release single-point emissions quantification.

For the backward Lagrangian stochastic model, data where the measurement point is on the edge of touchdown or

outside of the touchdown is flagged by the model as (-9999) and this was filtered out.

Further, models comparison using a data subset, 15-minutes 10 degrees wind sector range has been added to the

manuscript.

Changes to Manuscript

Section 2.4.2.2

Lines 290-295

The GPIM was evaluated under six scenarios (two equations and three different dispersion coefficients generations)

using single-release single-point emissions to test when the model works best (Supplementary Information Section

2a: Equation 7 and 8). Dispersion coefficients were generated based on (1) high frequency sonic anemometer data at

~ 10 Hz, (2) EPA point-source dispersion coefficients (US EPA, 2013), and (3) 1 Hz sonic anemometer data. The

scenario with the slope closest to 1, and highest  $R^2$  across averaging durations, and wind sector ranges was selected and used for multi-release single-point emissions quantification.

From the controlled release data file, there are 713 individual releases. This reviewer counts N=69 clearly defined single point single source trials with a mean (min, max) emission rate of 1.3 (0.2, 6.6) kg/hr and a release duration of 2 hrs and 41 minutes (0:19, 7:10). This represents a very reasonable dataset that can be understood. The remainder of the dataset consists of multiple release points grouped as trials, some very complex in design with many source locations and mixtures of long and short release durations. It is not clear how the MSRP set is formed and screened since almost all multipoint release trials involve multiple release locations. Even with use of mean wind direction as an upwind source screening tool, the potential influence of off-axis sources under meandering winds over the duration of a trial is difficult to eliminate. The authors acknowledge this possibility in Section 4.3(4.4) for both the GPIM and bLs result. However, this has as a direct bearing on the conclusions drawn as the "interfering sources" may dominate the MSRP result for GPIM and bLs. Additionally, the bLs point source analysis by WindTrax can strongly diverge when source is located near the edge of the touchdown cone (similar to Figure 2B). This is nonphysical effect that is the cause of the extreme outlier results (e.g., MRF = 11958) and should be evaluated.

**Author's response:**

Thank you for your comment. We agree that using the average wind direction as an upwind screening tool is effective for clearly isolated, single-source emissions, but more problematic in multi-source trials where wind meandering and overlapping plumes introduce significant uncertainty. We acknowledge that in such cases, plume interference can compromise the accuracy of quantification, particularly for point-source dispersion models such as GPIM and bLs. As noted, these complexities are not limitations of the modeling approaches per se, but rather reflect the constraints of our experimental setup. Our single-sensor configuration and the decision to include multi-source trials were intentional, aiming to evaluate model performance under realistic but challenging conditions—such as those encountered in continuous emissions monitoring. In response to Reviewer 2's suggestion, we have revised the manuscript to more explicitly clarify the influence of experimental design on source interference and model performance. We now emphasize that, in practical applications, improved source localization and screening can often be achieved using multiple sensors, which were beyond the scope of this study. Regarding the extreme outliers in the bLs point source analysis (e.g., MRF = 11958, in the revised plot, extreme estimates of up to 2\*10^5 kg h-1), we agree this divergence can occur when the estimated source lies near the edge of the touchdown cone—a known artifact of the model's geometry and assumptions. This can also occur due to unsteady emissions due to plume meandering, or inaccurate atmospheric stability (L) over such a short time. This could be the case as the estimated emissions begin to stabilize at 15 and 30 minutes duration shown in the revised plot.

**Changes to Manuscript**

Section 4.2

Lines 518-527

The MSRP emission profiles tested in this study were complex challenging the GPIM application as the method is a point-source specific quantification approach and works best in open areas, free of obstacles, and when the background

concentration is well defined. For multiple emissions, even when the sensor is nominally downwind of a single source based on the average wind direction, quantification can be complicated by interference from neighboring sources. However, it is important to emphasize that such complexity is not a fundamental limitation of quantification itself, but rather a function of the experimental design and study objectives. For example, plume interference can often be minimized through strategic localization and optimization using multiple sensors—an approach that differs from the single-instrument setup used in this study. This study's design involves defining plumes based on wind sector ranges, as opposed to using multiple sensors to localize sources, and therefore does not replicate how various continuous monitoring solutions typically operate.

Section 4.4

Lines 560-566

Oil and gas point sources could either be single emissions or multiple emissions occurring concurrently. In this study's design, cases involving multiple emissions with more than one release point located upwind posed challenges for the specific Gaussian and backward Lagrangian stochastic (bLs) model implementations, which were applied assuming a single active source at a time. While these models can be extended to handle multi-source scenarios, the assumptions used here limited their ability to distinguish individual contributions when plumes overlapped. As a result, interference from neighboring emissions introduced ambiguity in model-observation alignment, particularly under complex wind conditions.

**Comments on Main Paper:**

**Comment 1**

Specific comments regarding EC.

Despite being improved from the original submission, there seems to be a fundamental lack of understanding of the eddy covariance method that undermines the experimental design, QAQC, and analysis in this study.

1. Figure 1A has an incorrect depiction of the eddy covariance method. It seems to imply downward eddies are negative fluxes and upward eddies are positive. In fact there are always both downward and upward eddies in contexts with either negative and positive fluxes.

**Author's response:**

Thank you for your comment. We have accordingly modified the depiction for eddy covariance and aerodynamic flux gradient

**Eddy Covariance**

2. From the responses from the authors, it is clear the data were not collected in a manner suitable for eddy covariance. For example, both the CH4 and the wind data were not sampled at 10 Hz. Instead they were sampled at variable frequencies from 4 - 9 Hz. At best the authors could average down to the lowest common frequency and caveat their results to acknowledge they are missing all of the high frequency eddies. It is inappropriate to interpolate up to a higher frequency as the authors have done here.

**Author's response:**

Thank you for your comment. Yes, our system did not sample data at the recommended >10 Hz frequency (varied between 4 and 12 Hz) suitable for eddy covariance. However, we did not interpolate the data to the highest frequency, instead, we filtered out our dataset when the frequency was 8 Hz and above. We down sampled all 8Hz and greater data to an 8 Hz common frequency. To make this clear to our readers, we have made the following changes to the manuscript.

**Changes to Manuscript:**

**Section 2.4.1.1 Lines 195-201**

Evaluating the MGGA CH4 data showed that actual sampling was between 4 and 12 Hz (majority of the data collected at approximately 6 Hz), even though the analyzer had been configured to sample at 10 Hz (Supplementary Information Section 2b). To account for this sampling variability, data were filtered to when sampling was equal to or greater than 8 Hz. Data sampled at frequencies above 8 Hz were down sampled to 8 Hz. The 8 Hz frequency threshold was selected to ensure uniform sampling, enough data for model evaluation as most sampling was at lower frequencies, and to preserve as much temporal resolution as possible given the system limitations.

**Section 4.1 Lines 475-485**

Our results were derived from data filtered to include only periods with sampling frequencies  $\geq 8$  Hz, which significantly reduced the number of usable emission measurements. Although the instrument was configured to sample at 10 Hz, it did not consistently achieve this rate. This discrepancy may be attributed to instrument-related factors such

as the 0.4-second gas flow response time, which could delay analysis of the drawn air sample in the cavity, or the use of a 3 lpm pump with 3 meters of tubing, which reduced the effective turnover rate. The dataset used for eddy covariance evaluation was predominantly flagged as low quality (flag 2) according to the Mauder and Foken (2004) quality control test, which classifies flux data based on steady-state conditions and the presence of well-developed turbulence (flags 0 = high, 1 = intermediate, 2 = low quality). Many of the low-quality flags were likely driven by wide deviation in w/CH4 stationarity reflecting intermittent plume capture, where the EC system alternated between sampling emitting and non-emitting regions. The EC model produced negative emission rates associated with negative fluxes during periods of high non-stationarity (Supplementary Information, Section 2c. iv).

3. The authors did not sync the dataloggers between the sonic anemometer and CH4 gas analyzer, so they have no reliable way to match the CH4 series averaged down to 8 Hz with the sonic data. In an ideal circumstance, out of sync time series can be compensated for if there is a clear point of maximum covariance when determining the time lag. However, the authors used an insufficient pump speed in their closed path system with a long tube length, which would induce both attenuation and massive time lag.

**Author's response:**

We acknowledge that asynchronous CH4 and sonic data collection with no accurate way of matching time series due to insufficient pump speed and a long tube is a significant source of error. We acknowledge this in our manuscript with the following changes:

**Changes to Manuscript:**

Section 2.4.1.1 Lines 203-209

As the MGGA gas analyzer and sonic anemometer were not designed to clock synchronously, using the MGGA CH4 clock time as a reference, meteorological data from the sonic anemometer were matched to the MGGA CH4 data using linear interpolation to generate concentration-meteorological 8 Hz data. While in an ideal circumstance of a fast pump and short tube length a correct timeseries matching can be achieved through establishing a clear point of maximum covariance when determining the time lag, this is difficult for our system due to a 3 lpm pump flowrate and a 3 m tubing that caused both attenuation and time lag.

**Section 4.1 Lines 493-495**

We acknowledge that the system used in this study was not designed or configured for standard eddy covariance analysis, and that this limitation impacts the interpretation of our results in the context of EC-based flux quantification.

4. The authors report that their closed path intake was 10 cm below the location of their sonic anemometer. This level of vertical separation would not matter for an instrument height of 40 meters for example, but for short towers such as theirs, vertical separation should be avoided, and is an additional source of error.

**Author's response:**

Thank you for your comment. The intake for the closed-path system was positioned approximately 10 cm below the sonic anemometer to protect the inlet tubing from debris and precipitation by mounting it on an aluminum shield facing downward. We recognize that even this small vertical separation can introduce additional errors in flux

measurements when using short towers. This design choice was a compromise to ensure instrument protection while maintaining data collection in field conditions. We have now clarified this limitation in the manuscript and acknowledge its potential impact on our results.

**Changes to Manuscript:**

Section 4.1 Lines 489-493

The intake for the closed-path system was positioned approximately 10 cm below the sonic anemometer to protect the inlet tubing from debris and precipitation by mounting it on an aluminum shield facing downward. We recognize that even this small vertical separation can introduce additional errors in flux measurements when using short towers. This design choice was a compromise to ensure instrument protection while maintaining data collection in field conditions. We acknowledge that the system used in this study was not designed or configured for standard eddy covariance analysis, and that this limitation impacts the interpretation of our results in the context of EC-based flux quantification.

5. Given all of these issues with the data collection, the transfer function corrections are likely substantial and probably outside of acceptable bounds. The original review recommended an analysis of the cospectra, which the authors did not do.

**Author's response:**

We have added cospectra analysis to the supplementary information.

**Changes to Manuscript:**

Section 2.4.1.3 Lines 239-242

Cospectral analysis revealed that the EC system in this study smoothed out low-frequency eddies, as the cospectra lack the ideal shape characterized by a low-frequency rise, a peak region, and a high-frequency decay (Supplementary Information Section 2c.i). While the slope in the high-frequency region varies around the theoretical -4/3 slope, the cospectral data followed the 1:1 line, indicating consistent spectral shape across sampling periods.

6. The authors need to provide evidence for their u\* threshold choice based on the independence of fluxes from u\*. This is a routine analysis and figure included in supplements for any eddy covariance study.

**Author's response:**

We evaluated the relationship between CH4 flux and u\* to identify the point where the fluxes are independent of u\*. We have added the CH4 flux vs u\* plots under supplementary information.

**Changes to Manuscript:**

Section 2.4.1.3 Lines 243-248

We also examined the relationship between CH4 flux and friction velocity (u\*) to identify a u threshold below which flux estimates may be unreliable (Supplementary Information Section 2c. ii). However, no consistent relationship was observed across atmospheric stability classes (unstable, stable, and neutral). CH4 fluxes varied widely—including both positive and negative values—across the full range of u\* (~0 to 1 m s-1), with no discernible threshold beyond which

fluxes stabilized. This indicated that CH4 fluxes were effectively independent of u\*, and thus, data from all u\* values were retained.

7. The authors use time intervals shorter than the typical 30 minutes. While 15 minutes can sometimes be sufficient, smaller intervals than that almost always are neglecting low frequencies of eddies and result in incorrect fluxes. Anytime an eddy covariance study deviates from the 30 minute standard, it has to be justified with an ogive analysis, as was requested in the initial review.

**Author's response:**

We have reanalyzed our data at 5-, 15- and 30-minute averaging periods. For all these averaging durations, ogive plots have been added under supplementary information.

**Changes to Manuscript:**

**Section 2.4.1.3 Lines 248-257**

Ogive analysis was conducted to assess whether averaging durations of 5, 15, and 30 minutes were sufficient for capturing the full turbulent flux. The resulting ogive curves deviated from the ideal asymptotic shape, particularly at the highest and lowest frequencies. Notably, the curves did not exhibit a clear plateau near the low-frequency end, where cumulative flux should approach unity. This indicates incomplete flux capture. Furthermore, the similarity in ogive shapes across different frequencies—mirroring patterns seen in the cospectra—suggests a lack of significant turbulent contributions and the influence of non-turbulent, possibly advective, processes. These results imply that the EC system may not have fully resolved the flux due to either insufficient averaging time, non-stationarity, or instrument-related limitations (Supplementary Information Section 2c.iii). As positive fluxes are generally considered emissions, and negative fluxes depositions, data were further filtered for positive fluxes which were then quantified to emission rates.

8. The authors interpret the large amount of data flagged as poor quality as a consequence of low turbulence or bad system design. It is most likely that non-stationarity is the main problem, as they are including extremely brief emissions that are shorter than the averaging interval.

**Author's response:**

We used release-events where release durations were equal or longer than the averaging interval. Shorter releases were excluded from the analysis. We also acknowledge that even though the release might have been steady, changes in wind direction, flow distortion caused by presence of nearby equipment, and differences between the source and measurement height could contribute to non-stationarity as evidenced by presence of negative fluxes in some cases.

**Changes to Manuscript:**

**Section 4.1 Lines 479-485**

The dataset used for eddy covariance evaluation was predominantly flagged as low quality (flag 2) according to the Mauder and Foken (2004) quality control test, which classifies flux data based on steady-state conditions and the presence of well-developed turbulence (flags 0 = high, 1 = intermediate, 2 = low quality). Many of the low-quality flags were likely driven by wide deviation in w/CH4 stationarity reflecting intermittent plume capture, where the EC system alternated between sampling emitting and non-emitting regions. The EC model produced negative emission

rates associated with negative fluxes during periods of high non-stationarity (Supplementary Information, Section 2c. iv).

9. The authors used a footprint model that previous studies have found to perform poorly in controlled release experiments, against the recommendation of the original review. There are several other widely used footprint models, one or more of which should at least be used in conjunction with the model the authors have chosen to use. The authors claim they cannot implement the Kormann and Meixner footprint model without multiple sonic anemometers, but they are incorrect.

**Author's response:**

We have accordingly recalculated the footprint using both the Klujn, and Kormann and Meixner footprint models.

**Changes to Manuscript:**

Section 2.4.1.4 Lines 259-267

Eddy covariance footprints were calculated using the Kljun et al. (2015) and the Kormann and Meixner (2001) footprint models. For the Kljun et al. (2015), the freely online MATLAB code of the model was used, while the Kormann and Meixner (2001) was coded in MATLAB. To determine the point source footprint contribution, the study first calculated the area that contributed 90% of the vertical flux; and based on the location (x and y coordinates based on wind direction and distance from source) of the point source, the source was determined if it was within the 90% footprint area. Point source emissions of sources within this region were then calculated based on the approach by Dumortier et al. (2019). This approach assumes all measured flux is equal to flux resulting from a single point source. In case of the mast being downwind of more than one source, more sonic anemometers are needed to solve the two unknown point source fluxes.

10. The authors use footprints generated from 3 m height instruments to interpret aerodynamic flux gradients from 2 and 4 m heights. This is incorrect. As was pointed out in the first review, an entire meter of height difference especially at these short heights has a massive effect on the size, extent, and peak location of footprints, and the 3 m footprint cannot be used to interpret at 2 m and 4 m instrument heights. By this same logic, the aerodynamic flux gradient method cannot be used for point sources, since the footprint influence at the source location is necessarily different from the two measurement heights.

**Author's response:**

We appreciate the point and agree that using the 3 m footprint to interpret flux gradients between 2 m and 4 m is not valid, particularly for a point source. We have removed the aerodynamic flux gradient analysis from the manuscript and revised the relevant sections accordingly.

11. The authors only report a single metric of performance in their results: bootstrapped relative mean factor. This is not the most meaningful metric of method performance, and bootstrapping it obfuscates the actual performance. What the authors should do is report the slope, adjusted r-squared, and significance of a linear regression through the actual and estimated emissions. The authors demonstrate this fit in the top row of their results figures, but the choice to log-transform one axis obfuscates the results. Using a log-plot artificially reduces the variability in results (visually). It is clear despite this, that there is very little agreement between the estimate and actual emissions for any of their results.

**Author's response:**

We have accordingly reported our results based on metrics of a linear regression between the estimated and actual emissions, (the slope, adjusted r-squared, and p-value).

**Changes to Manuscript**

Section 3

12. The authors justify their choice of gas analyzer by pointing out the only open-path CH4 analyzer on the market has a more limited range (0-40 ppm). However, that LICOR instrument can be extended to 100 ppm which would put it on the same range as the analyzer used in this study. Both the LICOR and Picarro instruments the authors mention in their discussion are specifically designed for eddy covariance and to operate in adverse weather. For example, the closed path Picarro system they mention is designed with a faster pump speed, shorter tube length, and synchronized datalogger, such that the system has a known system response time that while short of the 10-Hz that can be achieved with open-path systems, is still sufficient in many circumstances. There are comparative studies of closed path and open path methane eddy covariance that can serve as examples of closed-path performance when set up well. The majority of studies use open path sensors for methane eddy covariance because of the extra effort involved in avoiding the time-lag, attenuation, clock syncing, etc issues that the authors encounter here. Closed path eddy covariance can be done well, but it was not implemented correctly in this case.

**Author's response:**

We acknowledge that the instrument used in this study was not suitable for EC and that limits the EC conclusions drawn in the study.

**Changes to Manuscript:**

Section 4.1 Lines 502-509

This study acknowledges the limitations of the eddy covariance (EC) setup used, particularly that the ABB MGGA GLA131 Series analyzer is not designed specifically for EC applications. As a result, the conclusions drawn from the EC data are constrained. The study recommends further EC testing with instruments specifically designed for EC, ideally featuring a wide measurement range (0 to ~500 ppm), faster pump speeds, shorter tubing, synchronized data logging, sampling frequencies above 10 Hz, and rugged designs suitable for field deployment. Additionally, the study recognizes that environmental factors—such as obstructions, intermittent emissions, and variable wind directions causing plume meandering—can degrade EC data quality and complicate its application in oil and gas field studies.

**Ali Lashgari**

**Comment 6:**

For transparency, it would be helpful to the reader to have a section on validation where at least a few cherry-picked cases from the dataset are discussed after being processed through the different modeling frameworks in addition to having a few figures on how the measurements made by the different sensors compare in an author-defined common context.

**Authors response**

We removed the aerodynamic flux gradient from our study which used the two separate analyzers as Reviewer 1 stated the model cannot be used for point source quantification. As a result, the models we are evaluating are eddy covariance, Gaussian Plume Inverse Method and the backward Lagrangian stochastic model all using one instrument to avoid instrument bias.

For the eddy covariance data, we filtered out cases where negative fluxes were estimated by EddyPro and used two footprint models, Klujn et al (2015) and Kormann and Meixner (2001) to validate the footprint models. However, emissions were largely underestimated, and this has been acknowledged as our study limitation that could have been caused by non-stationarity and instrument limitations with our non-standard eddy covariance system.

For the Gaussian Plume Inverse model, we tested the model in six different scenarios to evaluate both the equations and generation of dispersion coefficients using single-release single-point emissions to test parameters where the model works best, and this was used for multi-release single-point emissions quantification.

For the backward Lagrangian stochastic model, data where the measurement point is on the edge of touchdown or outside of the touchdown is flagged by the model as (-9999) and this was filtered out.

Further, models comparison using a data subset, 15-minutes 10 degrees wind sector range has been added to the manuscript.

**Changes to Manuscript**

Section 2.4.4.2

Lines 290-295

The GPIM was evaluated under six scenarios (two equations and three different dispersion coefficients generations) using single-release single-point emissions to test when the model works best (Supplementary Information Section 2a: Equation 7 and 8). Dispersion coefficients were generated based on (1) high frequency sonic anemometer data at ~ 10 Hz, (2) EPA point-source dispersion coefficients (US EPA, 2013), and (3) 1 Hz sonic anemometer data. The scenario with the slope closest to 1, and highest R2 across averaging durations, and wind sector ranges was selected and used for multi-release single-point emissions quantification.

**Section 3.4**

**3.4 Models Comparison - Subset Data**

Using a subset of the data (SRSP), filtered by 15-minute intervals within a 10-degree wind sector range where each model provided an emission estimate, the bLs model exhibited the best performance, with its linear regression closely aligned with the 1:1 line (Figure 10). The slope of the regression line for the GPIM was 1.6,

indicating an overestimation, while the bLs had a slope of 0.95, suggesting high accuracy. In contrast, the EC model produced slopes of 0.08 and 0.10 when using the Kormann and Meixner (2001) and Kljun et al. (2015) footprint parameterizations, respectively, indicating significant underestimation. When emission estimates were categorized by emission point, the GPIM notably overestimated emissions at locations 4W-22 and 4W-51 (identified in Figure 3), both situated approximately 10 m from the measurement location. The EC model consistently underestimated emissions across all sites, while the bLs model provided estimates closest to the expected values. The EC model produced negative emission rates associated with negative fluxes during periods of high non-stationarity (Supplementary Information, Section 2c. iv). These deviations from stationarity reflect intermittent plume capture, where the EC system alternated between sampling emitting and non-emitting regions. Overall, these findings indicate that for source-receptor distances ranging from approximately 10 to 90 meters, the bLs model demonstrated the highest accuracy in quantifying emissions.

**Comment 7:**

Regarding the following statement: "In this study, the precision to which downwind methods (closed-path EC, AFG, GPIM and bLs) could quantify the emission rate ... introduced real-world scenarios that have not previously been tested"; the claim that the effect of obstacles is included in these experiments may need a reconsideration. To my knowledge, the tank battery is the only noticeable on-site obstacle and even its effect on concentration measurements by the sampling devices is expected to be non-existent: the tanks with a length-scale of ~4-5m are easily around 50m+ from the location of the sampling devices (figure 2). In the current revision, no evidence is included to substantiate the claim that the influence of obstacles is indeed present.

Additionally, while it is true that the field experiments were agnostic to the local conditions during the selected time period, the individual experiments were not stratified based on these conditions to offer a nuanced and in-depth discussion. Thus to claim that "In this study, the precision to which downwind methods could quantify the emission rate of point source(s) were tested in different atmospheric conditions (rain, sunny, snow, windy, calm etc.), and aerodynamic scenarios (emissions sources in open areas, behind obstacles, changing atmospheric stability, and day/night" appears to generalize beyond the specific experimental design.

**Authors response**

Thank you for this comment. We agree that our earlier statement overgeneralized the influence of obstacles and atmospheric conditions. In the current study, although field measurements were collected during a range of weather conditions, these were not explicitly stratified or controlled in the analysis. Furthermore, while on-site structures such as tank batteries were present, their distance from the sampling systems (>50 m) makes any direct aerodynamic influence on measurements unlikely. We have revised the manuscript to remove these claims and have rephrased the discussion to more accurately reflect the scope and limitations of the study.

**Changes to Manuscript**

Section 4

Lines 457-464

In this study, we evaluated the ability of downwind methods—including a non-standard closed-path EC system, the GPIM, and the bLs model—to quantify emissions from single-release and multi-release point sources. While the field measurements took place under naturally varying meteorological conditions, these were not explicitly stratified or analyzed as experimental factors. Additionally, although on-site infrastructure such as storage tanks were present, their distance from the sampling instruments (~50 m) likely rendered their aerodynamic influence negligible. As such, the analysis focuses on quantification of performance under realistic but uncontrolled field scenarios, without attributing model behavior to specific atmospheric or obstacle-related conditions.

**Comment 9:**

Although appreciated, I believe the explanation does not fully address the original comment. The authors may reduce the discussion around hardware specifications and instead provide a context as to how the authors recover the parity plots (figures 4-11). Since this research is an investigation on assessing the pros/cons of the different modeling frameworks, this context can be very valuable for a reader. As noted earlier, a validation exercise for a few cherry-picked cases when a direct connection is drawn going from raw measurements to rate estimates for the different approaches would go a long way in making this work more transparent.

**Authors response**

I have added supplementary data 'MATLAB Code & Software Configuration' folder that contains MATLAB scripts used detailing how primary measurement was converted to quantified estimates, WindTrax and EddyPro configuration, and a subset data used to generate the models comparison' using a data subset, 15-minutes 10 degrees wind sector range has been added to the manuscript.

I have a added a section of Traceability Example

I have also added more details on the quantification for the GPIM and bLs.

**Changes to Manuscript**

**Section 3.5**

**3.5 Traceability Example**

To illustrate how raw data were converted into model-based emission estimates, we present one representative 15-minute interval used in Figure 10. During a controlled release at point 4W-22 (wellhead), located approximately 10.5 m from the mast, the ground-truth release rate was 3 kg CH4 h-1. Over this interval, the average CH4 concentration enhancement was 8.3 ppm above the background (determined using the 5th percentile method, see Section 2.4.2.1). The cross wiwind direction was 153° (0.3 m crosswind distance), with an average wind speed of 5.8 m s-1. The same interval was processed through the three modeling frameworks:

- The bLs model (WindTrax), using measured concentration, geometry, and meteorological data, estimated 3.5 kg h-1.
- The GPIM model, using Equation 7 and 8 (Supplementary Information) and dispersion coefficients, estimated 10.3 kg h-1.

• The EC method (using the (Kormann and Meixner, 2001) footprint) estimated –0.004 kg h-1 due to a negative flux under high non-stationarity conditions.

This example illustrates how the bLs model reproduced the true emission most closely, GPIM overestimated, and EC underestimated the emission. More examples of data presented in Figure 10 are available under supplementary data "MATLAB Code & Software Configuration – Validation.xlsx".

**Section 2.4.2.1**

**Lines 270-279**

Methane concentration data from the MGGA analyzer and meteorology data from the sonic anemometer were averaged to 1 Hz and aggregated. The aggregated concentration-meteorological data were merged with METEC's release data and metadata, and release event tables created. The concentration-meteorological-release event data were then separated into single-release and multi-release events. For single-release events, the concentration-meteorological-release event tables were split into 5, 15 and 30-minute release event tables. Based on the bearing of the emission point to the measurement point and the average wind direction in the duration, the data was further filtered to downwind data,  $\pm 5^{\circ}$ ,  $\pm 10^{\circ}$  and  $\pm 20^{\circ}$  wind sector ranges. Multi-release events were further classified into multi-release single-point emissions (i.e., there were multiple emissions at the site level, but the mast was downwind of a single source) and multi-release multi-point emissions (i.e. there were multiple emissions at the site level and the mast was downwind of more than one source).

**Section 2.4.3**

The GPIM was evaluated under six scenarios (two equations and three different dispersion coefficients generations) using single-release single-point emissions to test when the model works best (Supplementary Information Section 2a: Equation 7 and 8). Dispersion coefficients were generated based on (1) high frequency sonic anemometer data at ~ 10 Hz, (2) EPA point-source dispersion coefficients (US EPA, 2013), and (3) 1 Hz sonic anemometer data. The scenario with the slope closest to 1, and highest R2 across averaging durations, and wind sector ranges was selected and used for multi-release single-point emissions quantification.

**Section 3.4**

**3.4 Models Comparison - Subset Data**

Using a subset of the data (SRSP), filtered by 15-minute intervals within a 10-degree wind sector range where each model provided an emission estimate, the bLs model exhibited the best performance, with its linear regression closely aligned with the 1:1 line (Figure 10). The slope of the regression line for the GPIM was 1.6, indicating an overestimation, while the bLs had a slope of 0.95, suggesting high accuracy. In contrast, the EC model produced slopes of 0.08 and 0.10 when using the Kormann and Meixner (2001) and Kljun et al. (2015) footprint parameterizations, respectively, indicating significant underestimation. When emission estimates were categorized by emission point, the GPIM notably overestimated emissions at locations 4W-22 and 4W-51 (identified in Figure 3), both situated approximately 10 m from the measurement location. The EC model consistently underestimated emissions across all sites, while the bLs model provided estimates closest to the expected values. The EC model produced negative emission rates associated with negative fluxes during periods of high non-stationarity (Supplementary Information, Section 2c. iv). These deviations from stationarity reflect intermittent plume capture,

where the EC system alternated between sampling emitting and non-emitting regions. Overall, these findings indicate that for source-receptor distances ranging from approximately 10 to 90 meters, the bLs model demonstrated the highest accuracy in quantifying emissions.

Additionally, it is stated that: "continuous monitoring requires deployment of multiple sensors which create limitations of cost ...". The term "continuous monitoring (CM)" is quite broad. Although the statement is true when multiple sensors are deployed at different points on a site to perform quantification and localization, the present framework of using effectively a single device (with vertically-separated inlets) in a fence-line role is clearly a different use of the CM term. It would be helpful for the reader if this aspect is clarified to remind the reader this investigation deals with a CM fence-line type system that is only one of the possibilities in which a CM system can be constructed and deployed.

**Authors response**

We agree that "continuous monitoring" encompasses a wide range of deployment strategies. In this study, the CM system consisted of a single sensor with an inlet in a fence-line configuration. To clarify this distinction, we have revised the manuscript to explicitly state that the results and limitations discussed apply specifically to this single-instrument fence-line CM approach, which differs from multi-sensor CM networks that may offer broader spatial coverage and source localization capabilities but with increased cost and complexity.

**Changes to Manuscript**

**Lines 496-502**

In this study, continuous monitoring was conducted using a single sensor with an inlet deployed at a fence-line distance. This system requires instrumentation capable of measuring a wide concentration range, as emissions from oil and gas sites can vary between 0 and 250 ppm (Supplementary Information Section 1). While continuous monitoring systems, comprising multiple sensors can offer enhanced spatial coverage and source localization, they also introduce higher costs. The limitations and findings reported here therefore apply specifically to this single-sensor fence-line continuous monitoring approach and may not be representative of all continuous monitoring frameworks.

**Comment 14:**

Please note that Ilonze et al. (2024) compared the results of the solution provided, most of which employed multiple sensors for quantification. It may lack the equivalency with the fence-line framework used in this study with the analysis of the solution providers' performance by Ilonze et al. (2024). My advice would be to withdraw this comparison or allaborate on the differences and their consequences.

**Authors response**

Thank you for the suggestion. We have retained the reference to Ilonze et al. (2024) but have clarified in the manuscript that the comparison is qualitative rather than direct. The differences in monitoring frameworks—multisensor vs. single-sensor fence-line setup—are now explicitly acknowledged, and we have discussed how these differences may influence quantification performance and interpretation.

**Changes to Manuscript**

**Lines 61-71**

Detection and localization of simulated fugitive emissions using this approach have been demonstrated successfully in controlled release studies. For example, Ilonze et al. (2024) reported a 90% probability of detection for emissions between 3.9 and 18.2 kg CH4 h-1 using multi-sensor and scanning/imaging systems. However, significant uncertainty in quantification remains; their study reported emissions being misestimated by a factor of 0.2 to 42 for releases between 0.1 and 1 kg CH4 h-1, and by a factor of 0.08 to 18 for emissions above 1 kg CH4 h-1. While informative, the methods in Ilonze et al. (2024) differ in keyways from those employed here—specifically, their use of multiple sensors and a distributed monitoring configuration as opposed to the single-instrument, fence-line-based framework used in our study—limiting direct comparison of quantification accuracy. This study will evaluate the quantification accuracy of the closed-path EC, Gaussian plume inverse model (GPIM), and the backward Lagrangian stochastic model (bLs) for oil and gas point source quantification using a single-instrument deployed at a fence line distance.

**Comment 16-17:**

I recommend clarification that the error range (40-60%) from the cited reference applies to operational emissions (not controlled releases) from a driving survey (i.e. not fence line motoring in the same sense of the word as continuous monitoring) of an entire basin that includes agriculture emissions among other things. It may be overgeneralization to use this reference to support the argument that GPIM is fundamentally limited, as appears to be the case right now.

**Authors response**

Thank you for your comment. We acknowledge that Riddick et al. (2022b) primarily reports on mobile survey-based quantification across a basin. However, the error range cited in our manuscript (40.7–60%) was derived from a controlled release experiment within that study, not from the basin-scale results. Specifically, the controlled release involved 10 replicate measurements of compressed natural gas released at 1.5 m above ground level and quantified using the same Gaussian plume inverse framework applied to mobile survey data. This provides a relevant and controlled reference point for understanding GPIM performance, even if the deployment mode differs from continuous fence-line monitoring. We have revised the manuscript to clarify this distinction and avoid overgeneralizing the results.

**Changes to Manuscript**

Lines 101-105

Riddick et al. (2022b) reported absolute uncertainties of between 40.7 and 60% in a controlled release experiment involving 10 replicate measurements of compressed natural gas (1.5 m release height), with concentrations measured using a mobile vehicle survey. While this differs from continuous fence-line deployment, it offers insight into the inherent uncertainty of the GPIM method in field conditions.

Second, here and elsewhere, the authors suggest that "quantification is complexed by interference from other neighboring sources". These statements may give the impression that quantification is impossible in such scenarios whereas it is clearly a question of the choices made in experimental design and the study objectives, both of which

are defined at the outset. Fence-line monitoring used in the way used here to model the relatively short controlled-release experiment is limited by the constraints imposed during the design phase rather than some larger theoretical limitation. I recommend that the explanation here and elsewhere in the manuscript should highlight this basic fact to ensure that the discussion is in good faith.

**Authors response**

The authors have modified the manuscript to address this concern regarding the framing of quantification challenges in the presence of neighboring sources. The revised text now appropriately distinguishes between theoretical limitations and practical constraints arising from experimental design.

**Changes to Manuscript**

**Lines 520-527**

For multiple emissions, even when the sensor is nominally downwind of a single source based on the average wind direction, quantification can be complicated by interference from neighboring sources. However, it is important to emphasize that such complexity is not a fundamental limitation of quantification itself, but rather a function of the experimental design and study objectives. For example, plume interference can often be minimized through strategic localization and optimization using multiple sensors—an approach that differs from the single-instrument setup used in this study. This study's design involves defining plumes based on wind sector ranges, as opposed to using multiple sensors to localize sources, and therefore does not replicate how various continuous monitoring solutions typically operate.

**Lines 541-551**

This discrepancy may be due to design-related challenges—specifically, interference from neighboring sources and the lack of distinct plume separation in complex flow conditions. Although the measurement point was nominally downwind of a single source, the real-world plume structure may not align with model assumptions. Additionally, the bLs implementation in WindTrax is designed for single-source scenarios and applying it in multi-source environments without adaptation can lead to inaccuracies. The GPIM and bLs methods are sensitive to background correction, which in this study was complicated by temporal overlap between release events and residual CH4 accumulation, particularly under stable atmospheric conditions. Although this is a controlled-release study, residual methane from prior emissions and the presence of multiple plumes can affect the CH4 concentration during a candidate event, challenging the assumptions used to define background and isolate a single-source plume using wind-sector-based criteria. These findings highlight the importance of aligning modeling assumptions with the experimental context rather than pointing to a fundamental limitation of the method itself.

**Comment 18:**

The authors state that this study shows the difficulty in defining the background. To my knowledge, one major point of pursuing controlled-release testing like the ones discussed here was to have test cases without interference from operational background sources. As such, this statement may be confusing to a reader.

**Authors response**

Indeed, one of the motivations for using controlled-release studies is to limit interference from external background sources. To clarify, our reference to the difficulty in defining the background pertains not to interference from external operational sources, but to challenges internal to the experimental design—specifically, the presence of residual CH4 from prior release events and the potential for multiple overlapping plumes during some test periods. While controlled-release removes background uncertainty from unrelated field operations, it does not fully eliminate the complexity of defining background in cases where emissions are frequent, closely spaced in time, or where meteorological conditions (e.g., stable stratification) limit dispersion. We have updated the manuscript (Lines 712–716) to reflect this clarification, emphasizing that the challenge stems from the temporal and spatial overlap of test releases, rather than from uncontrolled ambient sources.

Changes to Manuscript

Lines 545-551

The GPIM and bLs methods are sensitive to background correction, which in this study was complicated by temporal overlap between release events and residual CH4 accumulation, particularly under stable atmospheric conditions. Although this is a controlled-release study, residual methane from prior emissions and the presence of multiple plumes can affect the CH4 concentration during a candidate event, challenging the assumptions used to define background and isolate a single-source plume using wind-sector-based criteria.

Secondly, it is mentioned that "Gaussian model and the backward Lagrangian stochastic models are limited, as they can only quantify one source at a time; and interference from neighboring emissions affects the underlying principles of dispersion on which these models were developed". While it is true that the specific choice made in this paper restricts the Gaussian plume approach to a single source, there is numerous evidence of employing the Gaussian plume approach for multi-source problems and there are no limits imposed by the so-called "principles of dispersion" for such synchronous, constant-rate releases rates. As an example, a highly referenced article titled "The Mathematics of Atmospheric Dispersion Modeling" (2011) by JM Stockie provides examples of using the Gaussian plume model for multi-source problems.

**Authors response**

Thank you for the comment. We agree that the Gaussian plume and bLs models are not inherently limited to single-source applications and have been successfully applied to multi-source scenarios in the literature. Our original wording unintentionally implied a theoretical constraint, when the actual limitation stems from the specific model application choices in our study—namely, that each quantification event was treated under the assumption of a single dominant source upwind. This was an intentional simplification aligned with our controlled-release test design and quantification framework. To address this, we have revised the manuscript to clarify that the constraints discussed are due to implementation choices and experimental design, not to the underlying principles of dispersion modeling.

Changes to Manuscript Lines 560-566 Oil and gas point sources could either be single emissions or multiple emissions occurring concurrently. In this study's design, cases involving multiple emissions with more than one release point located upwind posed challenges for the specific Gaussian and backward Lagrangian stochastic (bLs) model implementations, which were applied assuming a single active source at a time. While these models can be extended to handle multi-source scenarios, the assumptions used here limited their ability to distinguish individual contributions when plumes overlapped. As a result, interference from neighboring emissions introduced ambiguity in model-observation alignment, particularly under complex wind conditions.

---

## Author Response (AR3)

Ms. Ref. No.: EGUSPHERE-2024-3161

Title: Evaluating the feasibility of using downwind methods to quantify point source oil and gas

emissions using continuous monitoring fence-line sensors

The Powerhouse Energy Campus

Colorado State University

430 North College Avenue

Fort Collins, CO 80524

E-mail: Mercy.Mbua@colostate.edu

August 9th, 2025

Dear Professor Presto,

We appreciate the time and effort you, the reviewers and community dedicated to providing

feedback on our manuscript. We have closely followed the suggestions and have revised the

manuscript accordingly. We believe the revisions have improved our manuscript and that it is

ready for publication. We have listed the original comments and our response in blue below for

our reply to comments. All figure numbers, tables, and lines refer to the updated manuscript.

Data for reviewers' is available at URL: https://doi.org/10.5061/dryad.hhmgqnkss.

We look forward to hearing from you.

Please find our detailed responses below.

Yours sincerely,

Mercy Mbua (corresponding author)

and co-authors: Stuart N. Riddick, Elijah Kiplimo, Kira B. Shonkwiler, Anna Hodshire and

Daniel Zimmerle

**Reviewer Comments**

Review of revision 2 egusphere-2024-3161 "Evaluating the accuracy of downwind methods for quantifying point source emissions", Mbua et al. (Reviewer Comments by E. Thoma and S. Ludwig)

The authors were responsive to reviewer comments, offering a significantly revised manuscript with improved results visibility and supporting detail. The authors added critical QA/QC information on eddy covariance (EC) and decided to remove the aerodynamic flux gradient analysis. The authors modified the Gaussian plume inverse method (GPIM) approach, and backwards Lagrangian stochastic (bLs) analysis (based on WindTrax). Additional details were added for these emission calculation approaches.

**Authors response**

Thank you for reviewing our manuscript and for the constructive feedback, which has greatly helped us improve its clarity and rigor.

There remains one major set of concerns with the revised manuscript relating to the EC analysis. As a first point, the EC results changed significantly from manuscript version 2. The reason for this is not clear and the authors are encouraged to double check the analysis and identify the root cause for this difference.

**Authors response**

Thank you for the comment. Yes, the EC results changed from the previous version because we initially used *absolute flux* rather than *positive flux* to calculate emissions. As a result, large negative fluxes were incorrectly represented as large emissions in our earlier analysis.

Secondly, version 2 of the manuscript evaluated model performance using the bootstrapped mean, whereas version 3 uses a linear function. The bootstrapped mean in version 2 was disproportionately influenced by large fluxes (and thus large emissions), which gave a misleading impression of strong EC performance. Switching to the linear function in version 3 provided a more accurate and transparent assessment of the results.

We have carefully reviewed the analysis and confirm that the EC results presented in the current manuscript are final.

Assuming the current version of the EC analysis is final, the major concern centers on the strength of conclusions on the performance of the EC approach that can be drawn from this study. The authors have progressed their EC footprint and QA/QC analysis significantly from the

original manuscript. However, new supporting information on ogive and cospectra departs significantly from expected form. These results indicate that EC analysis is likely not possible with these data and in fact illustrate "textbook examples" of issues illuminated by these QA/QC checks. This general issue with the EC analysis is further evidenced by the presence of large negative fluxes (which indicate issues in the EC data collection).

When examining both ogives and cospectra as a part of the QA/QC process, there is both a qualitative shape expected and quantitative metrics of slopes (for portions of the cospectra) and sigmoidal parameters (for the ogives) when good EC data are collected. Some deviations from the ideal form are expected. For example, especially in closed-path eddy covariance, there is often tube attenuation or increased lag that results in poorer performance with data at the highest frequencies. This is seen as a slightly steeper slope than ideal in the cospectra shape at high frequencies and is compensated for during the transfer function calculations when processing data into fluxes. However, even in closed-path EC or EC sampled at 5 Hz rather than 10 Hz, the cospectra still closely follow the ideal shape (especially when examining the cospectra of sonic temperature, which should not suffer any of the issues of the closed path gas sampling system), and slope changes at high frequencies are well modeled by the transfer functions. When the shape of cospectra deviate significantly from the ideal curve (as is the case here), it is an indication the data were not collected properly in a way that can be used for eddy covariance, with causes that include obstructions, mis-aligned time series, too slow system response time, among other issues with the instrumentation as seen here.

Similarly, the ogive analysis should follow a characteristic shape, a sigmoid curve plateauing at the y-axis at 1 and also at 0. The ogive analysis is used to indicate if an appropriate averaging interval was used, as those that are too short will not sufficiently plateau at 1. Furthermore, those ogives which do not follow a sigmoid shape at all indicate issues in data collection. Even accounting for the log-scale y-axis in the authors' ogive figures, they do not follow an acceptable shape, and all ogives here would indicate issues leading to removing the data during QA/QC. I am including examples of the appropriate expected shapes of cospectra and ogives as described in the textbook "Eddy Covariance Method" by George Burba, section 5.1 "Quality Control of Eddy Covariance Flux Data". This chapter provides several examples of how the shapes of cospectra can be used to diagnose issues with the instrumentation and data collection (such as is the case here) that invalidate the EC method.

From the book "Eddy Covariance Method" by George Burba, section 5.1 Quality Control of EC data; Cospectra Analysis. This figure depicts both the ideal cospectra and expected slope deviations at high frequencies for certain gases and systems.

From the book "Eddy Covariance Method" by George Burba, section 5.1 Quality Control of EC data; Ogive Optimization. This figure depicts the expected ogive shape for observations at a site with an optimized averaging interval of 60-minutes.

**Authors' response**

Thank you for your comment. We acknowledge there were issues with the data collection system that invalidates EC analysis and conclusions. To clarify this to our readers, we have added the section below to the manuscript.

**Changes to the Manuscript**

Section 3.6

**3.6 Eddy Covariance Quality Assurance and Control**

Evaluation of the EC data revealed quality assurance and control issues that compromised both the analysis and the conclusions drawn from the EC results. The flux data were flagged as "2" (low quality) according to the 0-1-2 quality classification system of Mauder and Foken (2004), indicating that the data were not suitable for EC analysis. In EC quality assessment, both the qualitative shape of the cospectra and the quantitative slopes of selected portions are examined to determine if the data meet accepted standards. In this study, the cospectra deviated significantly from the ideal shape, indicating problems in data collection and pre-processing. Possible causes include obstructions in the testing area, misalignment between CH4 and sonic anemometer time series (due to the absence of a reliable method for alignment), slow response time of the gas analyzer, increased lag from the 3 m inlet tubing, and inconsistent sampling frequency. Similarly, ogive analysis—used to evaluate whether the averaging time is sufficient—showed that the ogive curves did not follow the characteristic sigmoidal shape (plateauing at the y-axis and at zero). Although the ogive shapes were similar across all averaging intervals, none plateaued sufficiently, further indicating data collection issues that invalidate the EC method for this study. For clarity and to guide future studies, Burba (2013) provides examples of ideal cospectra and ogive shapes illustrating how these tools can be used to diagnose instrumentation and data collection problems.

The authors now acknowledge the limitations in design of the EC data acquisition system for this study and attempt caveat in numerous places. They also point to non-stationarity in the data as part of the issue with the EC measurement.

However, if the EC results are deemed invalid, then these caveats are insufficient and conclusions around EC performance are not supported. The authors should either remove the EC analysis or suitably modify description to further clarify the severity of the issues for the reader. With little further work, the authors may choose to take this opportunity to illustrate some basic aspects of QA/QC assessment of EC data for this application. The information would be beneficial to the oil and gas/leak detection community (that largely consists of non-EC experts) and would assist

future efforts to assess EC for this application.

Here is one example of unsupported conclusions from the abstract.

Ln 17 "Generally, the closed-path EC system used in this study proved generally unreliable and largely underestimated emissions, primarily due to non-stationarity and study limitations associated with using a non-standard setup. In comparison, the Gaussian Plume Inverse Method (GPIM) consistently outperformed the EC system for both single-release and multi-release single-point emissions."

This is an inappropriate indictment of the EC methodology. If your primary QA/QC data indicate that the attempt at the application of the EC method was not successful (for whatever reason), then it is not possible to draw this conclusion. If the EC analysis is to remain in this manuscript, the description needs to be recast as an attempt at EC that failed. This would render the presented comparisons to other methods invalid. The issues with the method application were detected and reasons for these issues are presented here as lessons learned. Future attempts at exploring EC for this application will benefit from the information in this paper.

**Authors' response**

Thank you for your comment. We acknowledge that issues with the EC data collection system invalidate the results and the conclusions previously drawn. However, we have chosen to retain the EC analysis in the manuscript, reframing it as a "lessons learned" case study. We believe this provides valuable guidance for future studies by documenting the challenges encountered, the diagnostic tools applied, and the indicators of compromised data quality.

**Changes to the Manuscript**

Abstract

Lines 17-20

This study's EC attempt was unsuccessful due to data collection and instrumentation issues, resulting in invalid outputs characterized by underestimated emissions, large negative fluxes, and cospectra/ogives that deviated from their ideal shapes. Consequently, the EC results could not be compared with the GPIM or the bLS models.

**Section 4.1**

**Lines 507-518**

As a result, the conclusions drawn from the EC data are invalid and not comparable to the other tested models are constrained.

This study identified data collection and instrumentation issues that future work can address to enable successful EC application. Based on flagged low-quality data, non-ideal cospectra and ogive shapes, and the presence of large negative fluxes, the dataset was deemed unsuitable for EC analysis. The primary causes of the unsuccessful application were: (1) the CH4 analyzer was not designed for EC measurements, exhibiting slow response time, low pump flow rate, and inconsistent sampling frequency; (2) the 3 m inlet tubing length for the closed-path analyzer caused signal attenuation and increased lag; (3) the sonic anemometer and CH4 analyzer data were not synchronously logged, preventing accurate time-series alignment; (4) the EC system was installed near obstacles that disrupted smooth eddy formation; and (5) ogive plots suggested that the maximum 30-minute averaging interval used in this study may have been insufficient. We recommend further EC testing with these issues corrected to properly evaluate its application in continuous oil and gas monitoring.

**Lines 569-582**

Oil and gas point sources could either be single emissions or multiple emissions occurring concurrently. In this study's design, cases involving multiple emissions with more than one release point located upwind posed challenges for the specific Gaussian and backward Lagrangian stochastic (bLs) model implementations, which were applied assuming a single active source at a time. While these models can be extended to handle multi-source scenarios, the assumptions used here limited their ability to distinguish individual contributions when plumes overlapped. As a result, interference from neighboring emissions introduced ambiguity in model-observation alignment, particularly under complex wind conditions. Closed-path eddy covariance was generally unreliable in this study due to data-collection and instrumentation issues non-stationarity and limitations associated with using a non-standard EC system. This resulted in invalid EC results that could not be compared with the GPIM and the bLs models. In contrast, the Gaussian Plume Inverse Method (GPIM) outperformed the non-standard EC system for both single-release and multi-release single-point emissions. The backward Lagrangian stochastic (bLs) method was the most accurate for single-release single-point emissions but was less accurate than the GPIM under multi-release conditions. For both GPIM and bLs, 15-minute averaging with a narrow wind-sector (5°) yielded the best performance. While EC results in this

study were limited by system constraints, future work is recommended using standard EC instruments and further optimizing GPIM and bLs models—particularly for complex multi-release scenarios—to improve accuracy and reduce uncertainties.